# TAP: Two-Stage Adaptive Personalization of Multi-Task and Multi-Modal Foundation Models in Federated Learning

## Abstract

Federated Learning (FL), despite demonstrating impressive capabilities in the training of multiple models in a distributed manner, has been shown to produce a final model not necessarily well-suited to the needs of each client. While extensive work has been conducted on how to create tailored personalized models, called Personalized Federated Learning (PFL), less attention has been given to personalization via fine-tuning of foundation models with multi-task and multi-modal properties. Moreover, there exists a lack of understanding in the literature on how to fine-tune and personalize such models in a setting that is heterogeneous across clients not only in data, but also in tasks and modalities. To address this gap in the literature, we propose TAP (Two-Stage Adaptive Personalization), which (i) leverages mismatched model architectures between the clients and server to selectively conduct replacement operations when it benefits a client's local tasks and (ii) engages in post-FL knowledge distillation for capturing beneficial general knowledge without compromising personalization. We also introduce the first convergence analysis of the server model under its modality-task pair architecture, and demonstrate that as the number of modality-task pairs increases, its ability to cater to all tasks suffers. Through extensive experiments, we demonstrate the effectiveness of our proposed algorithm across a variety of datasets and tasks in comparison to a multitude of baselines.

## 1 Introduction

Federated Learning (FL) is a distributed machine learning paradigm that has garnered significant attention in recent years due to its ability to collaboratively train a model without the need to share potentially sensitive data. Through a network of clients, each client hosts its own model, which are trained locally and transmitted to a central server for aggregation (Konečný et al., 2016; McMahan et al., 2017). In this vein, recent attention has been given to the question of how to deploy foundation models in a federated setting, with large language models (LLMs) (Fang et al., 2025; Ye et al., 2024) being the most popular application.

However, due to the collaborative nature of FL, the final model may not be particularly well-suited to each local client. To tackle this challenge, **personalized FL (PFL)** has been explored (Tan et al., 2022; Deng et al., 2020), seeking to allow the server to train a global collaborative model while simultaneously allowing each local client to train a model tailored to its own local data.

**Challenges:** Despite many promising approaches to PFL, a majority of explored applications are limited to a uni-modal and uni-task scenario, where the clients and server share the same model architecture, task, and modality. Moreover, the application of fine-tuning larger foundation models to a PFL setting is an emerging area of research, with a limiting assumption being that while potentially dealing with multi-modal and/or multi-task models, they still share a unified architecture amongst all clients and the server, enabling easy aggregation protocols. In real-life scenarios, however, it will often be the case that each client's modalities and tasks will differ, resulting in differing model architectures and necessitating a need for the learning process to account for these differences. For example, in a healthcare setting, differing institutions (e.g., hospitals, clinics, etc.) may collect differing modalities (e.g., image scans or text reports) and pursue distinct tasks such as diagnosis

prediction or report generation. In industrial IoT, factories may have differing sensor systems (e.g. acoustic vs. temperature measurement sensors) that optimize for differing priorities such as safety or defective product detection. Therefore, a crucial question will be how to personalize each client's local model when models returned from the server are *heterogeneous in both tasks and modalities.* While an existing work in this domain exists via a Mixture of Experts (MoE) based modality and task routing mechanism (Chen & Zhang, 2024), its approach largely relies on the model returned from the server via its MoE-based architecture, leaving out potential further personalization that could be obtained from local insights. Moreover, it does not consider how to utilize this architecture for fine-tuning with already existing pre-trained foundation models.

## 1.1 CONTRIBUTIONS

Motivated by these challenges and gaps in the literature, we develop a personalized FL methodology to allow for personalization of foundation models that are heterogeneous in tasks and modalities. Our methodology allows for the server to still learn a generalizable model capable of handling all tasks and modalities while the local client's model is adapted towards its own tasks. The approach follows a two stage process. Firstly, during FL training, each client will hold a personalized model *in addition to the model transmitted to the server*, and will replace a subset of its personalized model when it receives indication that the returned model from the server would benefit the personal model on a specific task, permitting each client to effectively pick-and-choose beneficial modality-task pairings with limited interference among pairs under multi-modal, multi-task conditions. Secondly, after FL communication ends, we engage in knowledge distillation, as done in many existing PFL works (Chen et al., 2024; Jiang et al., 2020), by which we utilize the returned model from the server as a teacher, serving as a means to incorporate potentially unseen and beneficial insights from the teacher model without affecting personalization. While not tailored for personalization, its learned representations can still provide generalizable knowledge presented from other modality-task pairings without engaging in an aggressive operation such as replacement. Therefore, our key contributions are outlined as follows:

- We propose a two-stage adaptive personalization (TAP) algorithm, which seeks to leverage parameters engaged in the vanilla FL protocol when beneficial to a client's local task. Utilizing client defined margin hyperparameters, each client can effectively engage in replacement when the model engaged in the FL process offers significant benefit to the personalization process. Afterwards, we utilize a knowledge distillation-based post-training process, tuning the personalized model by distilling knowledge from representations of the FL-engaged model while maintaining high personalization.

- We provide the first convergence analysis of the server model under its modality-task based architecture. Although the server model itself is not personalized, this analysis sheds light on how its convergence degrades with the addition of more modality-task pairs, motivating the necessity for designing personalized FL methods under this setup.

- We conduct extensive experiments across a wide variety of common datasets encompassing a diverse array of tasks, demonstrating the superiority of our proposed method in comparison to a multitude of relevant baselines. In other words, we show that our method is better suited to personalization when considering the additional complexity of heterogeneity in modalities and tasks across clients.

## 2 RELATED WORK

**Foundation Models:** Foundation Models, as outlined by Bommasani et al. (2021), are models trained on large amounts of data that can be fine-tuned and adapted to a range of downstream tasks (Lian et al., 2024). In this regard, a large variety of foundation models have been developed and considered, such as BERT (Koroteev, 2021), DALL-E (Marcus et al., 2022), and the highly-popularized GPT family (Achiam et al., 2023). These base models are often considered *pre-trained*, and fine-tuning of such models have been shown to offer promise across a variety of applications, such as medical imaging (Zhang et al., 2025) and sentiment analysis (Zhang et al., 2023).

**Personalized Federated Learning (PFL):** With Personalized Federated Learning (PFL), the goal is for clients to train models that are personalized towards their own data while also collaboratively

training with other clients via the vanilla FL training process. In PFL, personalization usually falls into two broad categories: (i) personalized fine-tuning via the global model (Kairouz et al., 2021; Mansour et al., 2020; Fallah et al., 2020) and (ii) training of individual personalized models, separated from the typical FL learning process (Tan et al., 2022; Ghuhan et al., 2019). For the second category, while the personalized model is not presented to the server during training, it is common to utilize the global model as an informative basis for personalization (Chen et al., 2024). In the vein of PFL with foundation models, while existing works on dealing with multi-modal (Luo et al., 2025) and/or multi-task PFL exist (Chen et al., 2024; Long et al., 2024), these works are still limited in that either their tasks and/or modalities are consistent across all clients and assume a common model architecture, which TAP does not require. While Lu et al. (2024) explores heterogeneous model architectures in a multi-task setting, it is specifically designed for image inputs and therefore is uni-modal, unlike TAP, which deals with both heterogeneous tasks and modalities. To deal with this limitation, Chen & Zhang (2024) propose to have separate encoders and decoders for differing modalities and tasks, with a shared transformer backbone utilizing a Mixture of Experts (MoE) (Yuksel et al., 2012) structure to route inputs based off modality-task pairs. However, while a disentanglement axillary loss is introduced to potentially create more separable latent spaces between pairs, this approach can still result in entanglement of local models that share common subsets of the server model during aggregation. By contrast, TAP seeks to decouple itself as much as possible from the FL training process by only interacting with the server-engaged parameters when they signal that they can benefit the personalization process, avoiding the issue of losing the granularity of specific tasks during aggregation. Similar to Chen & Zhang (2024), Panchal et al. (2022) also explores using a routing mechanism for personalization, but the approach is uni-modal and uni-task without consideration for foundation models. Moreover, it necessitates the training of additional routing parameters, which TAP does not require for replacement operations.

**Parameter Efficient Fine-tuning (PEFT):** In fine-tuning of foundation models, considerable attention has been given to methodologies that are efficient in that only a small number of parameters are updated. These suite of techniques fall into the realm of parameter efficient fine-tuning (PEFT), and are particularly well-suited for FL scenarios (Wang et al., 2019), due to the limited computational resources of local clients. Well-known PEFT algorithms include prefix tuning (Li & Liang, 2021), prompt tuning (Lester et al., 2021), and LoRA (Hu et al., 2022), and are often trained on a small subset of the model parameters, significantly reducing computational resource requirements, making training on clients feasible. LoRA remains the most popular choice that a majority of works and frameworks adopt due to its simplicity, updating two low-rank matrices instead of the original model parameters, which are added on top of the original weight matrices.

**Knowledge Distillation (KD):** Knowledge distillation (KD) is a suite of techniques designed to distill useful knowledge from one model (known as the teacher) to another (known as the student). The teacher, which has been trained on data that would be beneficial to whatever task(s) the student seeks to optimize, allows the student to utilize its logits to try to minimize the KL-divergence (Kullback & Leibler, 1951) between them (Gou et al., 2021). Existing works have consistently demonstrated that KD offers significant benefits in quicker and more reliable training in a wide range of tasks, ranging from classification (Phuong & Lampert, 2019) to image generation (Cui et al., 2023). In the realm of FL, it has been shown to be a powerful technique in dealing with the issue non-i.i.d. data distributions between clients (Hsieh et al., 2020; Li et al., 2020; Lee et al., 2024) by engaging in KD with the server's logits on a per-label basis (Jeong et al., 2018). Moreover, KD has been shown to act as an effective regularizer in PFL, whereby the global model's logits prevent the personalized model from overfitting on the client's data (Chen et al., 2024).

**Mixture of Experts (MoE):** Mixture of Experts (MoE) are a class of architectures that seek to specialize models by introducing "expert" sub-networks with routing mechanisms to activate certain experts over others depending on the input. A common example is the top-$k$ selection (Zhou et al., 2022), whereby the top-$k$ experts selected by the router are utilized. Existing work has shown that MoEs can be utilized for multi-task objectives (Ma et al., 2018). Moreover, MoEs have been developed for multi-task (Chen et al., 2023; Fan et al., 2022) and multi-modal (Cao et al., 2023) foundation models, with some considering both dimensions (Wu et al., 2025). However, these works focus on a centralized setting and offer no straightforward mechanism to personalize specific modality-task pairs that vary across clients when naively extended to a federated environment. While our method adopts the MoE components from Chen & Zhang (2024), this choice is architectural rather than conceptual as TAP itself does not target MoE-specific routing or expert selection strategies.

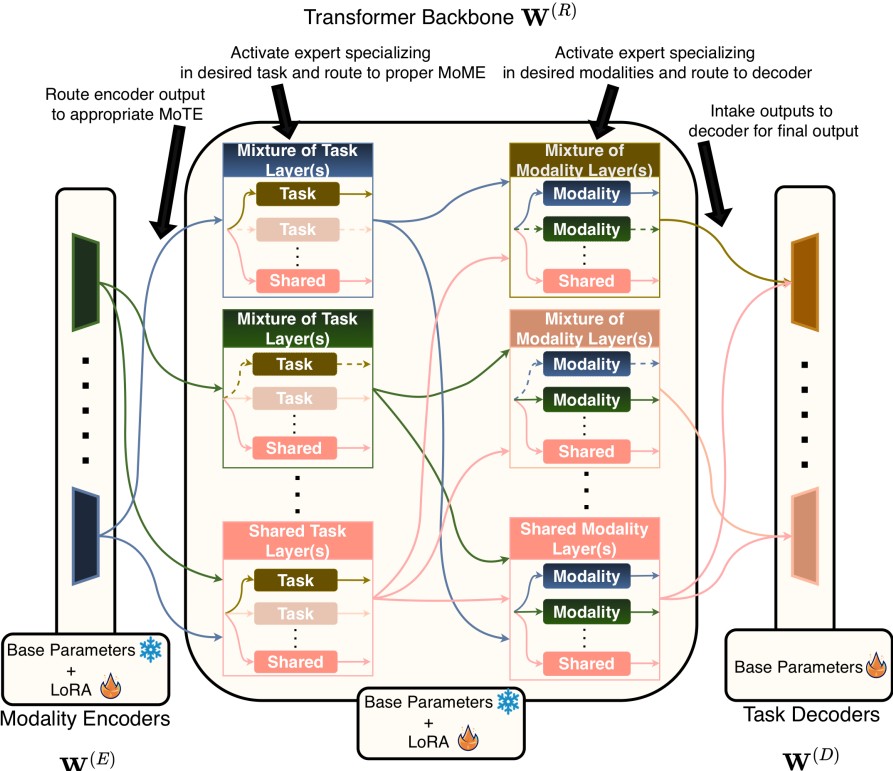

Figure 1: Architecture of server model $\mathbf{W}$. Each local client $c_i$ has a subset of $\mathbf{W}$ consisting of encoders, transformer layers, decoders, and experts relevant to its set of modalities and tasks.

## 3 PROPOSED METHODOLOGY

### 3.1 PERSONALIZED FEDERATED LEARNING IN MULTI-TASK, MULTI-MODALITY SETUP

We consider a multi-modal and multi-task federated learning setup with $K$ clients in set $c_i \in \mathcal{C}$ and a server $S$, with $\mathcal{C}_t \subseteq \mathcal{C}$ clients selected for aggregation each communication round $t$. Adopting the architecture from Chen & Zhang (2024), the server model's parameter vector $\mathbf{W} \in \mathbb{R}^{d \times 1}$ holds all modality encoders $\mathbf{W}^{(E)} \in \mathbb{R}^{d^{(E)} \times 1}$ and task decoders $\mathbf{W}^{(D)} \in \mathbb{R}^{d^{(D)} \times 1}$. Therefore, the server model is designed to handle all modalities $\mathcal{M}$ and tasks $\mathcal{O}$, i.e., $\mathcal{M} = \bigcup_{c_i \in \mathcal{C}} \mathcal{M}_i$ and $\mathcal{O} = \bigcup_{c_i \in \mathcal{C}} \mathcal{O}_i$, where $\mathcal{M}_i$ and $\mathcal{O}_i$ are the modalities and tasks pertaining to client $c_i$ respectively. The transformer backbone $\mathbf{W}^{(R)} \in \mathbb{R}^{d^{(R)} \times 1}$ sits between the encoders and decoders. Additional detailed specifics on the server model layout can be found in Appendix A.2.

Besides the general architecture, the transformer backbone and encoder parameters are pre-trained and frozen, with it being fine-tuned with LoRA low-rank matrices $\mathbf{A}$ and $\mathbf{B}$. This means only the decoder is fully trained. Each client $c_i$ holds a subset of the server model, consisting of modality encoders $\mathbf{W}_{[i]}^{(E)} \in \mathbb{R}^{d_i^{(E)} \times 1}$, task decoders $\mathbf{W}_{[i]}^{(D)} \in \mathbb{R}^{d_i^{(D)} \times 1}$, transformer backbone $\mathbf{W}_{[i]}^{(R)}$, and LoRA matrices $\mathbf{A}_i$ and $\mathbf{B}_i$ based off the set of modalities $\mathcal{M}_i$ and tasks $\mathcal{O}_i$ that client $c_i$ is responsible for. The combined local model is therefore defined as $\mathbf{W}_{[i]} = \mathbf{W}_{[i]}^{(E)} \cup \mathbf{W}_{[i]}^{(D)} \cup \mathbf{W}_{[i]}^{(R)} \cup \mathbf{A}_i \cup \mathbf{B}_i$ with a learning rate of $\eta_t$. We define the frozen and trainable components of the model for each client $c_i$ as $\widehat{\mathbf{W}}_{[i]} = \mathbf{W}_{[i]}^{(E)} \cup \mathbf{W}_{[i]}^{(R)}$ and $\widetilde{\mathbf{W}}_{[i]} = \mathbf{A}_i \cup \mathbf{B}_i \cup \mathbf{W}_{[i]}^{(D)}$ respectively. For the global model, it is $\widehat{\mathbf{W}} = \mathbf{W}^{(E)} \cup \mathbf{W}^{(R)}$ and $\widetilde{\mathbf{W}} = \mathbf{A} \cup \mathbf{B} \cup \mathbf{W}^{(D)}$. Fig. 1 presents the multi-modal, multi-task server model architecture along with the depiction of the LoRA fine-tuned and fully trained components.

For training, clients will train their local models for $\tau$ minibatch iterations, and then broadcast their parameters to the server for aggregation. When the server transmits parameters back to each client

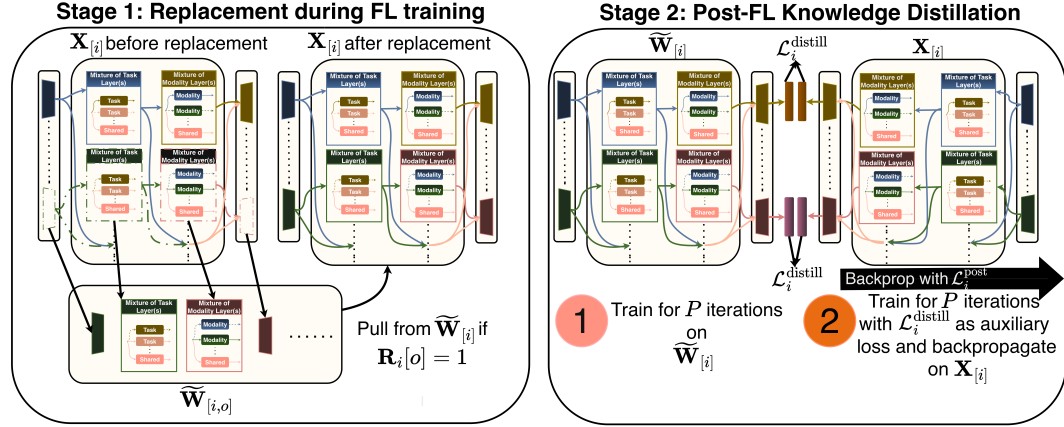

Figure 2: The two-step process of the proposed TAP algorithm. Firstly, during FL training, at each time step, $\mathbf{X}_{[i]}$ will load in parameters from $\widetilde{\mathbf{W}}_{[i,o]}$ when $\mathbf{R}_i[o] = 1$. After FL, using $\widetilde{\mathbf{W}}_{[i]}$ as a teacher, $\mathbf{X}_{[i]}$ will engage in knowledge distillation (KD).

after aggregation, it will only send the decoders and LoRA parameters that are relevant to each client based off their tasks and modalities. Utilizing this structure, in traditional federated learning, the objective is find parameters that minimize the global objective, which can be expressed as

$$\min_{\widetilde{\mathbf{W}}} \left[ f\left(\widetilde{\mathbf{W}}\right) := \frac{1}{|\mathcal{D}|} \sum_{i=1}^{K} |\mathcal{D}_i| \cdot g_i\left(\widetilde{\mathbf{W}}\right) \right], \quad (1)$$

where $\mathcal{D}$ and $\mathcal{D}_i \subseteq \mathcal{D}$ denote the full dataset across all clients and a local dataset on client $c_i$ respectively. We say that the server model trainable parameter vector $\widetilde{\mathbf{W}}$ can be partitioned into disjoint blocks via $\mathcal{B} = \{\mathcal{B}_0, \dots, \mathcal{B}_R\}$. Then, for notation purposes, we say there are projection operators $P_{\mathcal{B}_r}$ (linear) that zero out coordinates outside of block $\mathcal{B}_r$, so $\widetilde{\mathbf{W}}_{[r]} = P_{\mathcal{B}_r}\widetilde{\mathbf{W}}$ and $\widetilde{\mathbf{W}} = \sum_{r=0}^{R} \widetilde{\mathbf{W}}_{[r]}$. Each client $c_i$ owns a subset of blocks $\mathcal{B}_i \subseteq \mathcal{B}$, and since the client's local loss only depends on these blocks, local loss for client $c_i$ is $g_i\left(\widetilde{\mathbf{W}}\right) = f_i\left(P_{\mathcal{B}_i}\widetilde{\mathbf{W}}\right) = f_i\left(\widetilde{\mathbf{W}}_{[i]}\right)$. Outside of these considerations, when dealing with multi-task scenarios, it is also necessary to consider the loss of multiple differing tasks, resulting in

$$g_i\left(\widetilde{\mathbf{W}}\right) = \mathbb{E}_{\mathcal{H}_i \sim \mathcal{D}_i} \left[ \sum_{o \in \mathcal{O}_i} \lambda_o \cdot \ell_{i,o}\left(\mathcal{H}_{i,o}; \widetilde{\mathbf{W}}_{[i,o]}\right) \right], \quad (2)$$

where $\ell_{i,o}$, $\mathcal{H}_{i,o} \subseteq \mathcal{H}_i$, and $\lambda_o$ denotes the sample loss of $c_i$ on task $o$, a subset of minibatch $\mathcal{H}_i$ with samples pertaining to task $o$, and the weighting given to the loss of task $o$ respectively. $\widetilde{\mathbf{W}}_{[i,o]} = P_{\mathcal{B}_{i,o}}\widetilde{\mathbf{W}}$, with $\mathcal{B}_{i,o} \subseteq \mathcal{B}_i$ being a subset of $\mathcal{B}_i$ pretaining to parts relevant to task $o$. However, unlike equation 1, in PFL, the endpoint is for each client to a achieve low local loss, i.e., a low value on $g_i(\cdot)$. Due to heterogeneous model architectures amongst clients in the current setup, aggregation is performed by utilizing FedAvg (McMahan et al., 2017) on each component individually, combining common components based off the clients chosen for a specific aggregation round. Therefore, it is a possibility that the entirety of $\widetilde{\mathbf{W}}$ is not updated within a single communication round.

## 3.2 ADAPTIVE REPLACEMENT FOR PERSONALIZATION

Since the goal of each client $c_i$ is to maximize personalization of its multi-modal and multi-task model towards its own local dataset, relying solely on server parameters $\widetilde{\mathbf{W}}$ is challenging due to fact that $\widetilde{\mathbf{W}}$ is seeking to optimize all tasks $\mathcal{O}$, which utilizes all modalities $\mathcal{M}$. Therefore, it is desirable to limit interaction with $\widetilde{\mathbf{W}}_{[i]}$ (a subset of $\widetilde{\mathbf{W}}$ for client $c_i$), as optimizing for tasks not in $\mathcal{O}_i$ could interfere with personalization. For this reason, on top of $\widehat{\mathbf{W}}_{[i]}$, which engages in the vanilla FL

training protocol, we utilize local personalized parameters $\mathbf{X}_{[i]}$, which follows the same architecture as $\widetilde{\mathbf{W}}_{[i]}$, but does not participate in the transmitting and receiving of parameters from the server, and trains only on the local dataset $\mathcal{D}_i$. However, it could still be the case that the returned server parameters has useful information that could benefit the performance of $\mathbf{X}_{[i]}$, especially early on in the training process, when the most beneficial aspects of a task are learned (Frankle et al., 2020). Therefore, each client will keep track of two types of values–the average local training loss of $\widetilde{\mathbf{W}}_{[i]}$ and $\mathbf{X}_{[i]}$ for each task that was seen over the course of local training for that round $t$ ($\mathcal{O}_{t,i}^{\text{seen}} \subseteq \mathcal{O}_i$), which are defined as $\ell_{i,o}^{(l)}$ and $\ell_{i,o}^{(p)}$ respectively, where $o$ is a singular task. These values are used to update client $c_i$'s history values, defined as $h_{i,o}^{(l)}$ and $h_{i,o}^{(p)}$ via $h_{i,o}^{(l)} = \ell_{i,o}^{(l)}$ and $h_{i,o}^{(p)} = \ell_{i,o}^{(p)}$. Then, client $c_i$ will introduce margin hyperparameters $m_{i,o}$ for each task in $\mathcal{O}_i$, and will set a value to an entry of indicator vector $\mathbf{R}_i \in \mathbb{R}^{1 \times |\mathcal{O}_i|}$ via

$$\mathbf{R}_i[o] = \begin{cases} 1 & \texttt{if } h_{i,o}^{(l)} + m_{i,o} < h_{i,o}^{(p)} \\ 0 & \texttt{otherwise.} \end{cases} \tag{3}$$

Intuitively, $\mathbf{R}_i[o]$ is an indicator that $\widetilde{\mathbf{W}}_{[i]}$ at timestep $t$ achieves superior performance compared to $\mathbf{X}_{[i]}$ on task $o$ by at least a margin of $m_{i,o}$, and therefore can offer benefit to $\mathbf{X}_{[i]}$. In addition, through equation 3, we ensure $\mathbf{X}_{[i]}$'s training is no worse than $\widetilde{\mathbf{W}}_{[i]}$'s on any task $o$, preventing replacements on tasks that do not require it. For these reasons, when $\mathbf{R}_i[o] = 1$, client $c_i$ will replace parameters of $\mathbf{X}_{[i]}$ responsible for task $o$ with parameters from $\widetilde{\mathbf{W}}_{[i]}$, conducting the replacement operation (Stage 1 of Fig. 2) via

$$\mathbf{X}_{[i,o]} = (1 - \mathbf{R}_i[o])\mathbf{X}_{[i,o]} + \mathbf{R}_i[o]\widetilde{\mathbf{W}}_{[i,o]}, \tag{4}$$

where $\mathbf{X}_{[i,o]} \subseteq \mathbf{X}_{[i]}$ is the subset of $\mathbf{X}_{[i]}$ responsible for task $o$. In this way, the replacement operation can selectively extract the desired parameters while minimizing interference from other modality-task pairings, thereby enabling effective multi-modal and multi-task personalization, with input from $\widetilde{\mathbf{W}}_{[i]}$ incorporated when needed.

**Remark:** We note that the replacement process, which occurs in conjunction with the FL training protocol, does not induce additional communication costs, as the personalized parameters $\mathbf{X}_{[i]}$ *are not transmitted to server $S$*. Moreover, $\mathbf{X}_{[i]}$ can be trained on $c_i$ in parallel while $\widetilde{\mathbf{W}}_{[i]}$ is being aggregated at $S$, enabling effective usage of both local and server resources simultaneously without introducing additional idle time.

### 3.3 POST-FL KNOWLEDGE DISTILLATION

After FL training has been completed, we engage in knowledge distillation (KD) (Gou et al., 2021), whereby a teacher model seeks to distill knowledge to a student model via an auxiliary KL-divergence (Kullback & Leibler, 1951) loss. Firstly, the final parameters $\widetilde{\mathbf{W}}_{[i]}$ that was returned from the server $S$ will be trained locally for $P$ mini-batch iterations, and will then be used as a teacher model for $\mathbf{X}_{[i]}$, who will then also train for $P$ iterations utilizing KD. The intuition is that $\widetilde{\mathbf{W}}_{[i]}$ now benefits from both the FL process (which captures generalized knowledge) and some specialization on local data, free from conflicts introduced by other clients' distributions. The FL-trained base may help learn generalizable representations present across differing tasks, which a purely personalized model would otherwise not have access to. Unlike more aggressive operations such as replacement, post-FL KD enables transfer of beneficial knowledge while preserving each client's personalized nature. Existing work (Phuong & Lampert, 2019) theoretically demonstrates this to be the case, whereby the student is pulled towards the teacher but in general will not coincide completely with the teacher's logits. Overall, the distillation loss can be defined as

$$\mathcal{L}_i^{\text{distill}}(\mathbf{z}_{i,o}^{\text{teacher}}, \mathbf{z}_{i,o}^{\text{student}}) =$$

$$\widetilde{\tau}^2 \cdot \text{KL}\left( \frac{\exp\left(\mathbf{z}_{i,o}^{\text{teacher}}/\widetilde{\tau}\right)}{\sum_{j=1}^{|\mathbf{z}_{i,o}^{\text{teacher}}|} \exp\left(z_{i,o,j}^{\text{teacher}}/\widetilde{\tau}\right)} \;\middle\|\; \left( \frac{\mathbf{z}_{i,o}^{\text{student}}}{\widetilde{\tau}} - \log \sum_{k=1}^{|\mathbf{z}_{i,o}^{\text{student}}|} \exp\left(z_{i,o,k}^{\text{student}}/\widetilde{\tau}\right) \right) \right), \tag{5}$$

where $\mathbf{z}_{i,o}^{\text{teacher}}$ and $\mathbf{z}_{i,o}^{\text{student}}$ are the output logits of the local and personalized model for task $o$, with $\widetilde{\tau}$ as the temperature. Utilizing equation 5, $\mathbf{X}_{[i]}$ trains in the post-FL phase (Stage 2 of Fig. 2) via

$$\mathcal{L}_i^{\text{post}}\left(\mathcal{H}_i, \mathbf{z}_{i,o}^{\text{teacher}}, \mathbf{z}_{i,o}^{\text{student}}\right) = \sum_{o \in \mathcal{O}_i'} \lambda_o \cdot \ell_{i,o}(\mathcal{H}_{i,o}) + \sum_{o \in \mathcal{O}_i'} \beta_o \cdot \mathcal{L}_i^{\text{distill}}\left(\mathbf{z}_{i,o}^{\text{teacher}}, \mathbf{z}_{i,o}^{\text{student}}\right), \quad (6)$$

where $\beta_o$ and $\mathcal{O}_i' \subseteq \mathcal{O}_i$ is the weight given to the KD loss for task $o$ and the set of tasks present in minibatch $\mathcal{H}_i$ respectively. Pseudocode for the entirety of the TAP algorithm can be found in Appendix A.1.

# 4 CONVERGENCE ANALYSIS

In this following section, we analyze the convergence of the server model under its modality-task pair architecture. Firstly, the following assumptions common throughout literature (Bottou et al., 2018; Li et al., 2020; Fang et al., 2022; Lee et al., 2025) are made:

**Assumption 1** $g_i(\widetilde{\mathbf{W}})$ *is differentiable and L-smooth, i.e., there exists a positive constant L such that* $\|\nabla g_i(\widetilde{\mathbf{W}}_1) - \nabla g_i(\widetilde{\mathbf{W}}_2)\| \leq L\|\widetilde{\mathbf{W}}_1 - \widetilde{\mathbf{W}}_2\|$ *for any* $\widetilde{\mathbf{W}}_1$ *and* $\widetilde{\mathbf{W}}_2$.

**Assumption 2** *The minibatch* $\mathcal{H}_i$ *loss gradient for client* $c_i$, *denoted as* $\nabla\widetilde{\ell}_i\left(\mathcal{H}_i; \widetilde{\mathbf{W}}_{[i]}\right) = \sum_{o \in \mathcal{O}_i} \lambda_o \cdot \ell_{i,o}\left(\mathcal{H}_{i,o}; \widetilde{\mathbf{W}}_{[i,o]}\right)$ *is an unbiased estimate of* $\nabla g_i(\widetilde{\mathbf{W}})$, *i.e.,* $\mathbb{E}_{\mathcal{H}_i \sim \mathcal{D}_i}\left[\widetilde{\ell}_i\left(\mathcal{H}_i; \widetilde{\mathbf{W}}_{[i]}\right)\right] = \nabla g_i(\widetilde{\mathbf{W}})$, *and with its variance also being uniformly bounded, i.e.,* $\mathbb{E}\left\|\widetilde{\ell}_i\left(\mathcal{H}_i; \widetilde{\mathbf{W}}_{[i]}\right) - \nabla g_i(\widetilde{\mathbf{W}})\right\|^2 \leq \sigma^2$.

**Assumption 3** *For every block* $\mathcal{B}_r$ *and parameters* $\widetilde{\mathbf{W}}$, $\frac{1}{K_r}\sum_{i:\mathcal{B}_r \in \mathcal{B}_i}\|\nabla g_i(\widetilde{\mathbf{W}}_{[r]}) - \nabla f(\widetilde{\mathbf{W}}_{[r]})\|^2 \leq \zeta_r^2$, *i.e., the average squared deviation of owners' gradients in block* $\mathcal{B}_r$ *is at most* $\zeta_r^2$, *where* $K_r$ *is the number of clients having block* $\mathcal{B}_r$.

**Theorem 1** *Suppose Assumptions 1-3 hold. Then the iterates generated by component-based Fe-dAvg with full client participation and learning rate* $\eta_t \leq \min\left\{\frac{1}{48L\tau}, \frac{1}{\sqrt{8}L\tau}, \left(\frac{1}{96L^3\tau^3}\right)^{\frac{1}{3}}\right\}$ *satisfies:*

$$\frac{1}{\sum_{t=0}^{T-1}\eta_t}\sum_{t=0}^{T-1}\eta_t\mathbb{E}\|\nabla f(\widetilde{\mathbf{W}}_t)\|^2 \leq \frac{8\left(f(\widetilde{\mathbf{W}}_0) - f(\widetilde{\mathbf{W}}_T)\right)}{\sum_{t=0}^{T-1}\eta_t} + 16\left(Z + \frac{R+1}{3}\sigma^2\right)\frac{\tau^3 L^2}{\sum_{t=0}^{T-1}\eta_t}\sum_{t=0}^{T-1}\eta_t^3$$

$$+ 48L\tau^2 Z \frac{1}{\sum_{t=0}^{T-1}\eta_t}\sum_{t=0}^{T-1}\eta_t^2 + 16L\tau\sigma^2 C_K \frac{1}{\sum_{t=0}^{T-1}\eta_t}\sum_{t=0}^{T-1}\eta_t^2,$$

$$(7)$$

where $\widetilde{\mathbf{W}}_0$ and $\widetilde{\mathbf{W}}_T$ are the global parameters at global timestep $0$ and $T$ respectively. $\tau$ is the number of local iterations, $R$ is for indexing over the blocks, $Z := \sum_{r=0}^{R}\zeta_r^2$, and $C_K := \sum_{r=0}^{R}\frac{1}{K_r}$. With a diminishing step size of $\eta_t = \frac{\alpha}{t+1}$, $\alpha = \min\left\{\frac{1}{48L\tau}, \frac{1}{\sqrt{8}L\tau}, \left(\frac{1}{96L^3\tau^3}\right)^{\frac{1}{3}}\right\}$, $\lim_{T\to\infty}\frac{1}{\sum_{t=0}^{T-1}\eta_t} \to 0$, $\lim_{T\to\infty}\frac{1}{\sum_{t=0}^{T-1}\eta_t}\sum_{t=0}^{T-1}\eta_t^2 \to 0$, and $\lim_{T\to\infty}\frac{1}{\sum_{t=0}^{T-1}\eta_t}\sum_{t=0}^{T-1}\eta_t^3 \to 0$. Hence, the RHS of equation 7 goes to $0$ as $T$ increases to infinity. If there are more disjoint blocks in $\mathcal{B}$ to consider, and therefore more modality-task pairs, then we note that the bound increases in the last three terms of the RHS. This means that if the number of modality-task pairs increases, the ability of the server to cater well to all clients decreases, which TAP seeks to alleviate through its two-stage personalization process. Derivation of the theorem can be found in Appendix B.2.

# 5 EXPERIMENTAL RESULTS

## 5.1 SETUP

**Model Architecture:** To evaluate the effectiveness of the proposed setup, we consider a server model with text and image input modalities, with tasks relating to image generation, text generation,

image classification, and text classification. We evaluate the method on two pre-trained foundation models dealing with image and text modalities: FLAVA (Singh et al., 2022) and ViLT (Kim et al., 2021). For FLAVA, the modality encoders are the pre-trained image and text encoders, with ViLT being the module of linear patch projections and word embeddings respectively. In terms of the transformer backbone, due to existing work suggesting that the pruning of later layers of a pre-trained foundation model (Sajjad et al., 2023) can be conducted while maintaining a majority of its performance, we load only the first two layers of the multi-modal encoder of FLAVA into $\mathbf{W}^{(R)}$. For ViLT, the first eight layers are utilized. Moreover, within the backbone, similar to existing work (Wu et al., 2022), we load the pre-trained FFN weights at each layer as the frozen pre-trained weights of the Mixture of Experts (MoE). As outlined in Sec. 3.1, the encoders and transformer backbone are fine-tuned with LoRA, with specifics outlined in Appendix D.2. For the decoders, the generation heads are shared for all generation tasks of a particular modality (decoder specifics outlined in Appendix D.1). In all experiments, the AdamW optimizer is used (Loshchilov & Hutter, 2019).

**Datasets:** We adopt the usage of eight common datasets for FLAVA and six for ViLT, spread across 30 clients. We utilize tiny-Imagenet (Le & Yang, 2015) and CIFAR-100 (Krizhevsky, 2009) for image classification, Fashion-MNIST (FMNIST) (Xiao et al., 2017) and Caltech-256 (Griffin et al., 2007) for image generation reconstruction, AG News for text classification (Zhang et al., 2015), and Massive Multitask Language Understanding (MMLU) (Hendrycks et al., 2021a;b) (Professional Law, Professional Medicine, and Moral Scenario subsets), the small variant of VQAv2 (Visual Question Answering v2) (Goyal et al., 2017), and CommonGen (Lin et al., 2020) for text generation. For measuring the quality of the generated text, we use commonly utilized metrics of BERTSCore (BS) (Zhang et al., 2020) and METEOR (Banerjee & Lavie, 2005). While FLAVA uses all 8 datasets, we exclude the usage of tiny-ImageNet and CIFAR-100 for ViLT, as ViLT's pre-trained model is not well-suited for such tasks.

**Baselines:** To benchmark the performance of the TAP algorithm, we compare against the following baselines: local, FedAvg (McMahan et al., 2017), FedAvg + Post-train, Per-FedAvg (Fallah et al., 2020), Per-FedAvg + Postrain, DisentAFL (Chen & Zhang, 2024), DisentAFL + Post-train, Fed-DAT (Chen et al., 2024), and FedDAT + Post-train. For local, no FL takes place, with each client training its own model locally without any interaction from the server. With FedAvg, as mentioned in Sec. 3.1, server $S$ aggregates common parameter components from the subset of clients chosen at round $t$ and returns the parameters relevant to each client's modality and tasks back down to all clients $c_i \in \mathcal{C}$. Per-FedAvg engages in an inner and outer step for updating the model, to find an initial shared point that can help clients adapt quickly to their own data. DisentAFL, besides utilizing the model architectures of server $S$ and clients $c_i \in \mathcal{C}$ as outlined in Fig. 1 and Sec. 3.1, also introduces an auxiliary disentanglement loss to local training, whereby it attempts to make differing modality-task pairs in the latent space orthogonal to one another. FedDAT introduces adapters near the FNNs of a pre-trained model, whose structure includes a global and personalized component. Lastly, we also incorporate post-training baselines for FedAvg, DisentAFL, Per-FedAvg, and FedDAT, where $\widetilde{\mathbf{W}}_{[i]}$ is trained for $P$ iterations on local dataset $\mathcal{D}_i$ after FL training.

**Hardware and Hyperparameters:** For all results, experiments were conducted on a server with a cluster of four NVIDIA A100-40GB GPUs. For the weighting of each task's loss during a minibatch iteration, we weight each task's weight $\lambda_o$ based on the number of samples related to task $o$ within a batch, i.e., $\frac{|\mathcal{H}_{i,o}|}{|\mathcal{H}_i|}$. For other hyperparameters, explicit details are outlined in Appendix D.2.

## 5.2 Results

**Performances in Comparison to Baselines:** Firstly, we assess the effectiveness of TAP in comparison to the baselines outlined in Sec. 5.1, with results presented in Tables 1 and 2 for image-related tasks and Table 3 for text-related tasks. Unless stated otherwise, results outlined in bold signify the highest performing method, with the second best underlined. Any portions indicated with — means the method diverged and failed the converge. Based off the results from Tables 1, 2, and 3, we firstly note the superiority of the proposed TAP methodology across a vast majority of the tasks evaluated, with the highest average accuracy and generation scores for image classification and text generation tasks on FLAVA, with image generation and text generation for ViLT. For tasks where performance of TAP lags behind the baselines, the proposed TAP algorithm still retains high performance and

Table 1: Performance comparison for ViLT on image datasets. — indicates that the algorithm has failed to converge for that specific task.

| Model | Method | FMNIST MSE (↓) | Caltech-256 MSE (↓) | Avg. Gen. MSE (↓) |
|---|---|---|---|---|
| ViLT | Local | 0.6267 ± 0.0379 | 0.6518 ± 0.1302 | 0.6368 |
| | FedAvg (McMahan et al., 2017) | 0.6128 ± 0.0822 | 0.5011 ± 0.1496 | 0.5681 |
| | FedAvg + Post-train | _0.5502 ± 0.0454_ | _0.4099 ± 0.1168_ | _0.4941_ |
| | Per-FedAvg (Fallah et al., 2020) | — | — | — |
| | Per-FedAvg + Post-train | — | — | — |
| | DisentAFL (Chen & Zhang, 2024) | 0.7925 ± 0.0444 | 0.5137 ± 0.0787 | 0.6810 |
| | DisentAFL + Post-train | 0.6333 ± 0.0423 | 0.4347 ± 0.0651 | 0.5539 |
| | FedDAT (Chen et al., 2024) | 1.0326 ± 0.1585 | 1.5726 ± 0.1497 | 1.2576 |
| | FedDAT + Post-train | 64.4282 ± 360.6894 | 1.5497 ± 0.1876 | 38.2288 |
| | **TAP (Ours)** | **0.5467 ± 0.0409** | **0.3949 ± 0.1148** | **0.4860** |

Table 2: Performance comparison for FLAVA on image datasets.

| Model | Method | Tiny-ImageNet Acc (↑) | CIFAR-100 Acc (↑) | FMNIST MSE (↓) | Caltech-256 MSE (↓) | Avg. Class. Acc (↑) | Avg. Gen. MSE (↓) |
|---|---|---|---|---|---|---|---|
| FLAVA | Local | 18.78 ± 12.50 | 23.26 ± 13.13 | 0.6209 ± 0.0230 | 0.6331 ± 0.0752 | 21.02 | 0.6290 |
| | FedAvg (McMahan et al., 2017) | 33.39 ± 5.26 | 46.93 ± 7.41 | 0.6292 ± 0.0304 | 0.4915 ± 0.0650 | 40.16 | 0.5374 |
| | FedAvg + Post-train | 42.27 ± 6.79 | _53.67 ± 5.46_ | 0.5651 ± 0.0064 | 0.4345 ± 0.0141 | _47.97_ | 0.478 |
| | Per-FedAvg (Fallah et al., 2020) | 27.75 ± 7.34 | 43.19 ± 8.42 | 0.7404 ± 0.0833 | 0.6390 ± 0.1000 | 35.47 | 0.6728 |
| | Per-FedAvg + Post-train | 41.27 ± 7.06 | 53.36 ± 5.01 | 0.5796 ± 0.0105 | _0.4051 ± 0.0244_ | 47.31 | _0.4633_ |
| | DisentAFL (Chen & Zhang, 2024) | 36.99 ± 11.12 | 0.85 ± 0.22 | 0.9036 ± 0.0179 | 0.5890 ± 0.0203 | 18.92 | 0.6939 |
| | DisentAFL + Post-train | _45.53 ± 8.57_ | 4.15 ± 0.52 | 0.7714 ± 0.0144 | 0.5496 ± 0.0253 | 24.84 | 0.6235 |
| | FedDAT (Chen et al., 2024) | 16.54 ± 9.04 | 30.53 ± 12.74 | 0.5890 ± 0.0639 | 0.4164 ± 0.0733 | 23.53 | 0.4787 |
| | FedDAT + Post-train | 26.15 ± 13.96 | 39.63 ± 13.41 | **0.5491 ± 0.0026** | **0.3718 ± 0.0176** | 32.89 | **0.4309** |
| | **TAP (Ours)** | **47.06 ± 6.80** | **56.26 ± 5.44** | _0.5572 ± 0.0083_ | 0.4252 ± 0.0258 | **51.66** | 0.4692 |

Table 3: Performance comparison across FLAVA and ViLT on text datasets.

| Model | Method | AG News Acc (↑) | MMLU BS (↑) | MMLU METEOR (↑) | VQA BS (↑) | VQA METEOR (↑) | CommonGen BS (↑) | CommonGen METEOR (↑) | Avg. Gen. BS (↑) | Avg. Gen. METEOR (↑) |
|---|---|---|---|---|---|---|---|---|---|---|
| FLAVA | Local | 90.26 ± 0.55 | _41.63 ± 1.26_ | **20.89 ± 2.83** | 47.72 ± 1.69 | 10.44 ± 2.37 | 30.62 ± 1.52 | 13.17 ± 2.68 | 39.66 | 13.62 |
| | FedAvg (McMahan et al., 2017) | 92.88 ± 0.22 | 37.83 ± 2.25 | 7.71 ± 1.76 | 37.68 ± 6.88 | 10.84 ± 4.33 | 29.70 ± 4.05 | 12.46 ± 6.72 | 34.52 | 10.86 |
| | FedAvg + Post-train | _92.97 ± 0.26_ | 36.88 ± 1.97 | 8.66 ± 1.24 | 41.62 ± 4.56 | 12.85 ± 2.87 | 31.48 ± 1.84 | 11.92 ± 1.68 | 36.62 | 11.64 |
| | Per-FedAvg (Fallah et al., 2020) | 92.82 ± 0.20 | 39.74 ± 2.14 | 9.17 ± 1.27 | 41.79 ± 10.81 | 18.06 ± 11.09 | 31.06 ± 3.68 | 10.53 ± 5.05 | 37.09 | 13.27 |
| | Per-FedAvg + Post-train | **92.99 ± 0.28** | 39.40 ± 3.84 | 10.92 ± 4.06 | 46.70 ± 8.42 | 22.09 ± 8.38 | 31.83 ± 2.90 | 13.76 ± 5.21 | 39.29 | 16.53 |
| | DisentAFL (Chen & Zhang, 2024) | 92.76 ± 0.28 | 40.10 ± 0.77 | 19.54 ± 0.89 | 30.77 ± 0.64 | 5.46 ± 0.11 | 26.67 ± 0.48 | 7.60 ± 0.38 | 30.99 | 9.13 |
| | DisentAFL + Post-train | 92.69 ± 0.30 | 40.37 ± 0.37 | _20.39 ± 1.64_ | 34.99 ± 0.74 | 7.68 ± 0.77 | 27.86 ± 1.93 | 8.51 ± 0.33 | 33.21 | 10.56 |
| | FedDAT (Chen et al., 2024) | 92.54 ± 0.72 | 41.01 ± 2.63 | 6.83 ± 1.65 | 50.72 ± 17.55 | 28.30 ± 22.72 | _37.02 ± 5.32_ | 13.74 ± 11.69 | 43.30 | 18.18 |
| | FedDAT + Post-train | 92.81 ± 0.36 | 40.29 ± 2.86 | 11.13 ± 6.60 | _57.96 ± 14.97_ | _39.32 ± 24.72_ | **40.25 ± 1.99** | **17.91 ± 7.97** | _47.34_ | _25.12_ |
| | **TAP (Ours)** | 92.66 ± 0.34 | **45.81 ± 4.32** | 19.80 ± 3.43 | **69.38 ± 7.47** | **47.60 ± 14.92** | 34.01 ± 2.96 | _17.17 ± 4.56_ | **50.52** | **29.87** |
| ViLT | Local | 54.98 ± 15.88 | 55.03 ± 1.99 | 40.59 ± 2.65 | 55.68 ± 4.04 | 40.90 ± 11.61 | _40.79 ± 1.19_ | **25.63 ± 1.53** | 49.60 | _34.73_ |
| | FedAvg (McMahan et al., 2017) | 70.61 ± 4.00 | 41.89 ± 1.45 | 6.79 ± 1.40 | 49.01 ± 11.17 | 18.25 ± 8.81 | 37.06 ± 4.53 | 14.03 ± 9.69 | 42.80 | 14.27 |
| | FedAvg + Post-train | 77.71 ± 2.35 | 42.87 ± 2.53 | 9.60 ± 4.64 | 60.42 ± 11.15 | 34.70 ± 14.45 | 39.22 ± 3.43 | 17.19 ± 9.56 | 48.43 | 22.68 |
| | Per-FedAvg (Fallah et al., 2020) | 25.00 ± 0.05 | 44.61 ± 3.40 | 8.26 ± 1.66 | 49.30 ± 10.46 | 29.07 ± 17.48 | 31.98 ± 2.27 | 4.51 ± 2.97 | 41.43 | 15.08 |
| | Per-FedAvg + Post-train | 25.00 ± 0.05 | 44.81 ± 4.60 | 22.82 ± 10.01 | **75.01 ± 7.22** | **65.48 ± 3.27** | 31.98 ± 2.27 | 4.51 ± 2.97 | _51.76_ | 32.56 |
| | DisentAFL (Chen & Zhang, 2024) | 65.50 ± 9.35 | 54.83 ± 1.22 | 40.25 ± 1.95 | 42.01 ± 0.50 | 21.72 ± 0.85 | 33.37 ± 0.51 | 9.69 ± 0.13 | 41.12 | 20.62 |
| | DisentAFL + Post-train | 74.69 ± 4.02 | _55.31 ± 0.94_ | _40.64 ± 1.42_ | 48.94 ± 3.06 | 31.53 ± 10.13 | 34.94 ± 2.87 | 11.92 ± 1.33 | 44.61 | 25.51 |
| | FedDAT (Chen et al., 2024) | 26.62 ± 0.96 | 43.10 ± 1.88 | 8.92 ± 2.62 | 52.04 ± 16.53 | 29.67 ± 22.28 | 37.92 ± 3.64 | 15.93 ± 9.55 | 44.61 | 20.02 |
| | FedDAT + Post-train | 27.07 ± 0.97 | 43.72 ± 3.72 | 9.71 ± 7.71 | 56.53 ± 14.86 | 32.61 ± 22.48 | 39.77 ± 2.37 | 17.06 ± 8.88 | 47.27 | 22.21 |
| | **TAP (Ours)** | _77.44 ± 2.63_ | **56.43 ± 3.12** | **40.79 ± 1.61** | _72.99 ± 7.47_ | _62.52 ± 9.09_ | **41.46 ± 1.45** | _25.35 ± 1.85_ | **57.07** | **43.31** |

remains close with the best baseline (e.g., FMNIST on FLAVA with 0.5491 (FedDAT + Post-train) vs. 0.5572 (TAP) or AG News on ViLT with 77.71 (FedAvg + Post-train) vs. 77.44 (TAP)), with it often being the next best option available (noted through underline). Moreover, we note that TAP is more consistent in performance across all tasks, whereas some baselines seem to excel at specific tasks while suffering on others (e.g., image generation vs. image classification on FLAVA with Fed-DAT or AG News vs. MMLU METEOR score on ViLT for FedAvg + Post-train). This is especially prevalent with Per-FedAvg on ViLT, where it exhibits impressive metrics on VQA but collapses on image generation, failing to produce a coherent result due to potential fluctuation and instability in the meta-learning process that Per-FedAvg adopts (Lan et al., 2023). An analysis of the statistical significance of the above results can also be found in Appendix C.5.

**Ablation on KD:** In this section, we explore the impact of utilizing knowledge distillation (KD) from equation 6 in Tables 4, 5, and 6. Firstly, we note that while KD under-performs no distillation taking place on image-generation tasks, the performance difference is minimal (e.g., 0.3940 vs. 0.3949 on Caltech-256 and 0.5466 vs. 0.5467 on FMNIST for ViLT). Moreover, this difference is made up for with performance gains across a vast majority of other tasks. In particular, we note that text generation tasks in Table 6 benefit the most from this process, as seen in VQA, which exhibits significantly higher BS and METEOR scores ranging from roughly 18 to 30 points of improvement

Table 4: Ablation study on the use of knowledge distillation (KD) in the post-training phase on image datasets for ViLT.

| Model | Method | FMNIST | Caltech-256 | Avg. Gen. |
|-------|--------|--------|-------------|-----------|
|       |        | MSE ($\downarrow$) | MSE ($\downarrow$) | MSE ($\downarrow$) |
| ViLT  | No KD  | **0.5466** $\pm$ 0.0410 | **0.3940** $\pm$ 0.1145 | **0.4856** |
|       | KD     | 0.5467 $\pm$ 0.0409 | 0.3949 $\pm$ 0.1148 | 0.4860 |

Table 5: Ablation study on the use of knowledge distillation (KD) in the post-training phase on image datasets for FLAVA.

| Model | Method | Tiny-ImageNet | CIFAR-100 | FMNIST | Caltech-256 | Avg. Class. | Avg. Gen. |
|-------|--------|---------------|-----------|--------|-------------|-------------|-----------|
|       |        | Acc ($\uparrow$) | Acc ($\uparrow$) | MSE ($\downarrow$) | MSE ($\downarrow$) | Acc ($\uparrow$) | MSE ($\downarrow$) |
| FLAVA | No KD | **47.06** $\pm$ 6.79 | 56.25 $\pm$ 5.47 | **0.5570** $\pm$ 0.0085 | **0.4247** $\pm$ 0.0260 | 51.65 | **0.4688** |
|       | KD    | **47.06** $\pm$ 6.80 | **56.26** $\pm$ 5.44 | 0.5572 $\pm$ 0.0083 | 0.4252 $\pm$ 0.0258 | **51.66** | 0.4692 |

Table 6: Ablation study on the use of knowledge distillation (KD) in the post-training phase on text datasets.

| Model | Method | AG News | MMLU | | VQA | | CommonGen | | Avg. Gen. | |
|-------|--------|---------|------|---|-----|---|-----------|---|-----------|---|
|       |        | Acc ($\uparrow$) | BS ($\uparrow$) | METEOR ($\uparrow$) | BS ($\uparrow$) | METEOR ($\uparrow$) | BS ($\uparrow$) | METEOR ($\uparrow$) | BS ($\uparrow$) | METEOR ($\uparrow$) |
| FLAVA | No KD | 92.65 $\pm$ 0.34 | 42.20 $\pm$ 1.32 | 15.79 $\pm$ 2.27 | 51.50 $\pm$ 3.30 | 16.82 $\pm$ 5.93 | 31.08 $\pm$ 1.31 | 14.93 $\pm$ 4.13 | 41.47 | 15.86 |
|       | KD    | **92.66** $\pm$ 0.34 | **45.81** $\pm$ 4.32 | **19.80** $\pm$ 3.83 | **69.38** $\pm$ 7.47 | **47.60** $\pm$ 14.92 | **34.01** $\pm$ 2.96 | **17.17** $\pm$ 4.56 | **50.52** | **29.87** |
| ViLT  | No KD | **77.47** $\pm$ 2.61 | 55.50 $\pm$ 1.00 | **40.94** $\pm$ 0.75 | 65.29 $\pm$ 9.50 | 52.14 $\pm$ 15.28 | 40.67 $\pm$ 2.07 | 25.18 $\pm$ 2.57 | 53.48 | 39.12 |
|       | KD    | 77.44 $\pm$ 2.63 | **56.43** $\pm$ 3.12 | 40.79 $\pm$ 1.61 | **72.99** $\pm$ 7.47 | **62.52** $\pm$ 9.09 | **41.46** $\pm$ 1.45 | **25.35** $\pm$ 1.85 | **57.07** | **43.31** |

for FLAVA and approximately 7 to 10 points for ViLT. Overall, we see an average increase of 3.5 to 9 points on BS and 4 to 14 points on METEOR across all text-generation based datasets. In regards to the difference in performance of image vs. text based tasks with KD, we believe this discrepancy to occur due to (i) the overconfidence of language models in comparison to vision models (Li et al., 2025) and (ii) the higher cross-client heterogeneity of text data. With text data, clients may differ in aspects such as vocabulary, topic, or writing style, meaning for text data, the server model captures a broader and more diverse distribution over words than any single client. This provides KD more breadth of knowledge to the client in comparison to image-based tasks.

**Additional Experiments:** Additional experimental results, ranging from further ablation studies on the margin hyperparameters, comparison of TAP against the baselines trained for $2P$ iterations, the number of replacements over FL training, and comparing the architecture in Fig. 1 against a standard transformer backbone can be found in Appendix C.

## 6 CONCLUSION AND LIMITATIONS

We introduced TAP, a novel two-step adaptive Personalized Federated Learning (PFL) algorithm that enables personalization of heterogeneous multi-modal and multi-task foundation models. Despite this additional complexity, TAP is capable of leveraging beneficial knowledge from the collaborative server model while maintaining high levels of personalization across clients. We provide comprehensive convergence analysis to motivate the insufficiency of the server model to cater to the needs of all clients, necessitating the need for PFL methods such as TAP. Through margin hyperparameters in the FL training period and a knowledge distillation (KD) based post-FL training period, we demonstrate that TAP enables superior personalization capabilities across a multitude of datasets, tasks, and model architectures. A potential limitation of our work is the reliance of the training loss to conduct replacements, which can produce issues should the training loss not be an accurate representation of the true difference in performance between the personalized and regular parameters on a specific task. Future work can explore differing methods of engaging in replacement to mitigate this pitfall.

## REPRODUCIBILITY STATEMENT

The manuscript provides the necessary information needed to reproduce the results, with general information found in Sec. 5.1 with specifics on hyperparameters, model architectures, and text templates found in Appendix D.1, D.2, and D.3. We also provide detailed pseudocode of the algorithm in Appendix A.1. Implementation code of the algorithm can be found in the supplementary material of the submission.

## LLM USAGE

We employed the GPT-5 version of ChatGPT to assist with improving the wording, rephrasing, and overall readability of the manuscript. In addition, it was used as an aid in the creation of tables throughout the manuscript. All algorithm development and experimental work were carried out solely by the authors.

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

# A  TECHNICAL DETAILS

## A.1  PSEUDOCODE OF TAP

Here, we give a detailed step-by-step pseudocode for the TAP algorithm. The for-loop encompassing global rounds 0 to $T - 1$ entails the FL training protocol and the replacement stage of the algorithm (Sec. 3.2). The last three lines of the algorithm delineate the knowledge-distillation (KD) process after FL communication ends (Sec. 3.3).

---

**Algorithm 1:** TAP Training Algorithm

---

**Input:** Clients $c_i \in \mathcal{C}$; Server $S$; datasets $\mathcal{D}_i \subseteq \mathcal{D}$; trainable parameters of local models $\widetilde{\mathbf{W}}_{[i]}$;
        personalized parameters $\mathbf{X}_{[i]}$; history values $h_{i,o}^{(l)}$ and $h_{i,o}^{(p)}$

**for** $t = 0$ **to** $T - 1$ **do**
    Sample subset of clients $c_i \in \mathcal{C}_t$
    **for** *client* $c_i \in \mathcal{C}_t$ **do in parallel**
        Train $\widetilde{\mathbf{W}}_{[i]}$ on $\mathcal{D}_i$ with learning rate $\eta_t$
        **for** *seen task* $o \in \mathcal{O}_{t,i}^{seen}$ **do**
            Update $\widetilde{\mathbf{W}}_{[i]}$'s history value: $h_{i,o}^{(l)} = \ell_{i,o}^{(l)}$
        Transmit $\widetilde{\mathbf{W}}_{[i]}$ from client $c_i$ to server $S$ for aggregation.
        **for** *task* $o \in \mathcal{O}_{t,i}^{seen}$ **do**
            Set $\mathbf{R}_i[o]$ to 0 or 1 based on equation 3
            Replacement on $\mathbf{X}_{[i,o]} \subseteq \mathbf{X}_{[i]}$ for task $o$ via equation 4.
        Reset all entries of $\mathbf{R}_i$ to 0.
        Train $\mathbf{X}_{[i]}$ on $\mathcal{D}_i$ with learning rate $\eta_t$
        **for** *seen task* $o \in \mathcal{O}_{t,i}^{seen}$ **do**
            Update $\mathbf{X}_{[i]}$'s history value: $h_{i,o}^{(p)} = \ell_{i,o}^{(p)}$
    Server $S$ broadcasts $\widetilde{\mathbf{W}}_{[i]}$ to all clients $c_i \in \mathcal{C}$
**for** *client* $c_i \in \mathcal{C}$ **do**
    Train $\widetilde{\mathbf{W}}_{[i]}$ for $P$ minibatch iterations on $\mathcal{D}_i$
    Train $\mathbf{X}_{[i]}$ for $P$ minibatch iterations on $\mathcal{D}_i$ utilizing loss from equation 6.

---

## A.2  SERVER MODEL LAYOUT

In this section, we present more detailed specifics on the layout of the model structure at server $S$ as seen in Fig. 1.

As discussed in Sec. 3.1, we adopt the architecture from Chen & Zhang (2024), with the server model's parameter vector $\mathbf{W} \in \mathbb{R}^{d \times 1}$ holding all modality encoders $\mathbf{W}^{(E)} \in \mathbb{R}^{d^{(E)} \times 1}$ and task decoders $\mathbf{W}^{(D)} \in \mathbb{R}^{d^{(D)} \times 1}$. The transformer backbone $\mathbf{W}^{(R)} \in \mathbb{R}^{d^{(R)} \times 1}$ sits between the encoders and decoders and consists of two stacks of layers–the first stack, of which there are $|\mathcal{M} + 1|$ of for each modality, is responsible for taking the outputs of a modality encoder and routing it as input to the proper layers responsible for that modality. Then, within those layers, the feed forward networks (FFN) are replaced with an MoE, which will activate the expert specialized for the task that the input data is seeking to solve. Then it is routed to the second stack of layers, of which there are $|\mathcal{O} + 1|$ of, where it selects the layers responsible for a certain task, utilizing an expert specialized for a certain modality. Then the output is combined and sent to the appropriate task decoder. There are $|\mathcal{M} + 1|$ and $|\mathcal{O} + 1|$ stacks because each side of the transformer has what are called "shared" layers, which are activated at all times, no matter the input. The stacks are called the Mixture of Modality Task Expert (MoTE) and Mixture of Modality Expert (MoME) layers respectively.

Table 7: Memory footprint of utilizing TAP vs. no usage of TAP and the wall clock time in seconds of the KD post-FL phase on average for each client.

| Model | Setting | Avg. Total Params. | Avg. % trainable | Avg. Memory | Avg. KD Training Time (sec.) |
|-------|---------|--------------------|------------------|-------------|------------------------------|
| **FLAVA** | No TAP | $310, 559, 996, 40$ | $10.3347\%$ | 1.1569 GB | — |
| | TAP | $347, 422, 994.40$ | $18.1086\%$ | 1.2943 GB | 202.65 |
| **ViLT** | No TAP | $585, 627, 611.80$ | $9.9788\%$ | 2.1816 GB | — |
| | TAP | $646, 296, 580.40$ | $18.0053\%$ | 2.4076 GB | 190.58 |

### A.3 MEMORY FOOTPRINT AND POST-FL TRAINING TIME

Due to the necessity of TAP requiring a post-FL knowledge distillation (Stage 2 of Fig. 2) phase and additional personalized parameters, we seek to characterize how these aspects affects the memory footprint and total extra training time per-client, which can be found in Table 7.

We firstly note that although TAP adds additional parameters, this expansion remains small relative to the full model size. For example, with FLAVA, the total parameter count increases from 310.6M to 347.4M, with ViLT being 585.6M to 646.3M. For TAP, this accounts for about 18% of total parameters, showing that TAP still remains faithful to the idea that fine-tuning foundation models should only involve a small number of the total parameters. Moreover, the memory overhead introduced by TAP is minimal. FLAVA's memory usage increases from 1.16GB to 1.29GB, and ViLT's from 2.18GB to 2.41GB, amounting to a roughly 10% increase. This indicates that the additional personalized parameters do not substantially affect the memory footprint.

In terms of average per-client knowledge distillation (KD) time, we note that the post-FL KD step adds only a modest amount of computation. The average per-client KD time is approximately 200 seconds, meaning it does not require each client to train for long durations after FL has completed.

## B ANALYSIS

### B.1 LEMMAS

As a preliminary, we introduce some basic lemmas, which will be utilized in the derivation of the theorem.

**Lemma 1** *If vectors $v_1, \ldots, v_N$ are such that $v_n$ only has non-zero values on a unique block of coordinates (e.g. $v_1$ only has non-zero for block 1, $v_2$ for block 2, and so on), then*

$$\left\|\sum_{n=1}^{N} v_n\right\|^2 = \sum_{n=1}^{N} \|v_n\|^2.$$

**Lemma 2** *Given vectors $v_1, \ldots, v_N$, the following is true:*

$$\left\|\sum_{n=1}^{N} v_n\right\|^2 \leq N \sum_{n=1}^{N} \|v_n\|^2.$$

### B.1.1 PROOF OF LEMMA 1

Given vectors $v_1, \ldots, v_N$, we have

$$\left\|\sum_{n=1}^{N} v_n\right\|^2 = \langle \sum_{n=1}^{N} v_n, \sum_{n'=1}^{N} v_{n'} \rangle = \sum_{n=1}^{N} \sum_{n'=1}^{N} \langle v_n, v_{n'} \rangle.$$

Due to orthogonality, $\langle v_n, v_{n'} \rangle = 0$ for $n \neq n'$, and with $n = n'$ being $\|v_n\|^2$, we have

$$\left\| \sum_{n=1}^{N} v_n \right\|^2 = \sum_{n=1}^{N} \|v_n\|^2,$$

which completes the proof.

### B.1.2 PROOF OF LEMMA 2

Given vectors $v_1, \ldots, v_N$, we have

$$\left\| \sum_{n=1}^{N} v_n \right\|^2 = \langle \sum_{n=1}^{N} v_n, \sum_{n'=1}^{N} v_{n'} \rangle = \sum_{n=1}^{N} \sum_{n'=1}^{N} \langle v_n, v_{n'} \rangle,$$

which can be decomposed and expressed via the following:

$$\sum_{n=1}^{N} \|v_n\|^2 + 2 \sum_{1 \leq n < n' \leq N} \langle v_n, v_{n'} \rangle.$$

Then, by Cauchy-Schwarz, where $\langle v_n, v_{n'} \rangle \leq \|v_n\| \|v_{n'}\|$, $n \neq n'$, we have

$$\left\| \sum_{n=1}^{N} v_n \right\|^2 \leq \sum_{n=1}^{N} \|v_n\|^2 + 2 \sum_{1 \leq n < n' \leq N} \|v_n\| \|v_{n'}\|,$$

where the RHS is equivalent to $\left( \sum_{n=1}^{N} \|v_n\| \right)^2$. Then by using Cauchy-Schwarz again, we obtain

$$\left\| \sum_{n=1}^{N} v_n \right\|^2 \leq \left( \sum_{n=1}^{N} \|v_n\| \right)^2 \leq \left( \sum_{n=1}^{N} 1^2 \right) \left( \sum_{n=1}^{N} \|v_n\|^2 \right),$$

which gives

$$\left\| \sum_{n=1}^{N} v_n \right\|^2 \leq N \left( \sum_{n=1}^{N} \|v_n\|^2 \right).$$

### B.2 PROOF OF THEOREM 1

Firstly, from Assumption 1, we have:

$$f(\widetilde{\mathbf{W}}_{t+1}) \leq f(\widetilde{\mathbf{W}}_t) + \langle \nabla f(\widetilde{\mathbf{W}}_t), \widetilde{\mathbf{W}}_{t+1} - \widetilde{\mathbf{W}}_t \rangle + \frac{L}{2} \|\widetilde{\mathbf{W}}_{t+1} - \widetilde{\mathbf{W}}_t\|^2. \tag{8}$$

Then, decompose the update by blocks and take expectation over round-$t$ randomness:

$$\mathbb{E}\left[ f(\widetilde{\mathbf{W}}_{t+1}) \right] \leq f(\widetilde{\mathbf{W}}_t) + \sum_{r=0}^{R} \mathbb{E}\langle \nabla f(\widetilde{\mathbf{W}}_{t,[r]}), \Delta_{t,[r]} \rangle + \frac{L}{2} \mathbb{E}\left\| \sum_{r=0}^{R} \Delta_{t,[r]} \right\|^2, \tag{9}$$

where $\Delta_{t,[r]} = -\frac{\eta_t}{K_r} \sum_{i:\mathcal{B}_r \in \mathcal{B}_i} \sum_{s=0}^{\tau-1} \nabla \tilde{\ell}\left( \mathcal{H}_i; P_{\mathcal{B}_r} \widetilde{\mathbf{W}}_{t,[i]}^{(s)} \right)$, with $\tau$ as the number of local training iterations. Therefore, $\widetilde{\mathbf{W}}_{t+1} = \widetilde{\mathbf{W}}_t + \sum_{r=0}^{R} \Delta_{t,[r]}$. Then, by Assumption 2, $\mathbb{E}\langle \nabla f(\widetilde{\mathbf{W}}_{t,[r]}), \Delta_{t,[r]} \rangle = -\eta_t \sum_{s=0}^{\tau-1} \mathbb{E}\langle \nabla f(\widetilde{\mathbf{W}}_{t,[r]}), \frac{1}{K_r} \sum_{i:\mathcal{B}_r \in \mathcal{B}_i} \nabla g_i(\widetilde{\mathbf{W}}_{t,[r]}^{(s)}) \rangle$. Next, with $-\langle a, b \rangle = \frac{1}{2}\|a-b\|^2 - \frac{1}{2}\|a\|^2 - \frac{1}{2}\|b\|^2$, where $a = \nabla f(\widetilde{\mathbf{W}}_{t,[r]})$ and $b = \frac{1}{K_r} \sum_{i:\mathcal{B}_r \in \mathcal{B}_i} \nabla g_i(\widetilde{\mathbf{W}}_{t,[r]}^{(s)})$, we can write the second term on the RHS as

$$
\mathbb{E}\left[f(\widetilde{\mathbf{W}}_{t+1})\right] \leq f(\widetilde{\mathbf{W}}_t) + \sum_{r=0}^{R} \frac{\eta_t}{2} \sum_{s=0}^{\tau-1} \mathbb{E}\left\|\nabla f(\widetilde{\mathbf{W}}_{t,[r]}) - \frac{1}{K_r} \sum_{i:\mathcal{B}_r \in \mathcal{B}_i} \nabla g_i(\widetilde{\mathbf{W}}_{t,[r]}^{(s)})\right\|^2
$$

$$
- \sum_{r=0}^{R} \frac{\eta_t \tau}{2} \|f(\widetilde{\mathbf{W}}_{t,[r]})\|^2 - \sum_{r=0}^{R} \frac{\eta_t}{2} \sum_{s=0}^{\tau-1} \mathbb{E}\left\|\frac{1}{K_r} \sum_{i:\mathcal{B}_r \in \mathcal{B}_i} \nabla g_i(\widetilde{\mathbf{W}}_{t,[r]}^{(s)})\right\|^2
$$

$$
+ \frac{L}{2} \mathbb{E}\left\|\sum_{r=0}^{R} \Delta_{t,[r]}\right\|^2. \tag{10}
$$

Then, discard the second to last term on the RHS. For the second term of the RHS, using the fact $\nabla f(\widetilde{\mathbf{W}}_{[r]}) = \frac{1}{K_r} \sum_{i:\mathcal{B}_r \in \mathcal{B}_i} \nabla g_i(\widetilde{\mathbf{W}}_{[r]})$, Jensen's inequality, and Assumption 1, we derive

$$
\mathbb{E}\left[f(\widetilde{\mathbf{W}}_{t+1})\right] \leq f(\widetilde{\mathbf{W}}_t) - \frac{\eta_t \tau}{2} \sum_{r=0}^{R} \|\nabla f(\widetilde{\mathbf{W}}_{t,[r]})\|^2 + \frac{\eta_t L^2}{2} \sum_{r=0}^{R} \sum_{s=0}^{\tau-1} \psi_{t,r}^{(s)} + \frac{L}{2} \mathbb{E}\left\|\sum_{r=0}^{R} \Delta_{t,[r]}\right\|^2, \tag{11}
$$

with $\psi_{t,r}^{(s)} := \frac{1}{K_r} \sum_{i:\mathcal{B}_r \in \mathcal{B}_i} \mathbb{E}\|\widetilde{\mathbf{W}}_{t,[i]}^{(s)} - \widetilde{\mathbf{W}}_{t,[r]}\|^2$. Then for the last term of the RHS, using the property of $\|u+v\|^2 \leq 2\|u\|^2 + 2\|v\|^2$ from Lemma 2, we can obtain

$$
\mathbb{E}\left[f(\widetilde{\mathbf{W}}_{t+1})\right] \leq f(\widetilde{\mathbf{W}}_t) - \frac{\eta_t \tau}{2} \sum_{r=0}^{R} \|\nabla f(\widetilde{\mathbf{W}}_{t,[r]})\|^2 + \frac{\eta_t L^2}{2} \sum_{r=0}^{R} \sum_{s=0}^{\tau-1} \psi_{t,r}^{(s)}
$$

$$
+ L \sum_{r=0}^{R=0} \frac{\eta_t^2}{K_r^2} \left(2\mathbb{E}\left\|\sum_{i:\mathcal{B}_r \in \mathcal{B}_i} \sum_{s=0}^{\tau-1} \left(\nabla\widetilde{\ell}(\cdot) - \nabla g_i(\widetilde{\mathbf{W}}_t^{(s)})\right)\right\|^2 + 2\mathbb{E}\left\|\sum_{i:\mathcal{B}_r \in \mathcal{B}_i} \sum_{s=0}^{\tau-1} \nabla g_i(\widetilde{\mathbf{W}}_t^{(s)})\right\|^2\right), \tag{12}
$$

where we abbreviate $\nabla\widetilde{\ell}(\cdot) = \nabla\widetilde{\ell}\left(\mathcal{H}_i; P_{\mathcal{B}_r} \widetilde{\mathbf{W}}_{t,[i]}^{(s)}\right)$. This can be further simplified with the variance term by Assumption 2 via

$$
\mathbb{E}\left[f(\widetilde{\mathbf{W}}_{t+1})\right] \leq f(\widetilde{\mathbf{W}}_t) - \frac{\eta_t \tau}{2} \sum_{r=0}^{R} \|\nabla f(\widetilde{\mathbf{W}}_{t,[r]})\|^2 + \frac{\eta_t L^2}{2} \sum_{r=0}^{R} \sum_{s=0}^{\tau-1} \psi_{t,r}^{(s)}
$$

$$
+ L \sum_{r=0}^{R=0} \frac{\eta_t^2}{K_r^2} \left(2K_r \tau \sigma^2 + 2\mathbb{E}\left\|\sum_{i:\mathcal{B}_r \in \mathcal{B}_i} \sum_{s=0}^{\tau-1} \nabla g_i(\widetilde{\mathbf{W}}_t^{(s)})\right\|^2\right). \tag{13}
$$

Moreover, via Lemma 2, we can further say

$$
\mathbb{E}\left[f(\widetilde{\mathbf{W}}_{t+1})\right] \leq f(\widetilde{\mathbf{W}}_t) - \frac{\eta_t \tau}{2} \sum_{r=0}^{R} \|\nabla f(\widetilde{\mathbf{W}}_{t,[r]})\|^2 + \frac{\eta_t L^2}{2} \sum_{r=0}^{R} \sum_{s=0}^{\tau-1} \psi_{t,r}^{(s)}
$$

$$
+ L \sum_{r=0}^{R=0} \frac{\eta_t^2}{K_r^2} \left(2K_r \tau \sigma^2 + 2K_r \tau \sum_{i:\mathcal{B}_r \in \mathcal{B}_i} \sum_{s=0}^{\tau-1} \mathbb{E}\|\nabla g_i(\widetilde{\mathbf{W}}_t^{(s)})\|^2\right) \tag{14}
$$

$$
\mathbb{E}\left[f(\widetilde{\mathbf{W}}_{t+1})\right] \leq f(\widetilde{\mathbf{W}}_t) - \frac{\eta_t \tau}{2} \sum_{r=0}^{R} \|\nabla f(\widetilde{\mathbf{W}}_{t,[r]})\|^2 + \frac{\eta_t L^2}{2} \sum_{r=0}^{R} \sum_{s=0}^{\tau-1} \psi_{t,r}^{(s)}
$$

$$
+ L \sum_{r=0}^{R=0} \frac{\eta_t^2}{K_r} \left(2\tau \sigma^2 + 2\tau \sum_{i:\mathcal{B}_r \in \mathcal{B}_i} \sum_{s=0}^{\tau-1} \mathbb{E}\|\nabla g_i(\widetilde{\mathbf{W}}_t^{(s)})\|^2\right). \tag{15}
$$

For the last term of the RHS, split the parentheses into

$$2L\eta_t^2\tau\sum_{r=0}^{R}\frac{\sigma^2}{K_r} + 2L\eta_t^2\tau\sum_{r=0}^{R}\sum_{s=0}^{\tau-1}\left(\frac{1}{K_r}\sum_{i:\mathcal{B}_r\in\mathcal{B}_i}\mathbb{E}\|\nabla g_i(\widetilde{\mathbf{W}}_t^{(s)})\|^2\right),$$

and then decompose

$$\nabla g_i(\widetilde{\mathbf{W}}_t^{(s)}) = \underbrace{\nabla f(\widetilde{\mathbf{W}}_{t,[r]})}_{u} + \underbrace{\left(\nabla g_i(\widetilde{\mathbf{W}}_{t,[r]}) - \nabla f(\widetilde{\mathbf{W}}_{t,[r]})\right)}_{v} + \underbrace{\left(\nabla g_i(\widetilde{\mathbf{W}}_t^{(s)}) - \nabla g_i(\widetilde{\mathbf{W}}_{t,[r]})\right)}_{w}.$$

Next, using $\|u+v+w\|^2 \leq 3(\|u\|^2+\|v\|^2+\|w\|^2)$ from Lemma 2 in conjunction with Assumptions 1 and 3, we have

$$\frac{1}{K_r}\sum_{i:\mathcal{B}_r\in\mathcal{B}_i}\mathbb{E}\|\nabla g_i(\widetilde{\mathbf{W}}_t^{(s)})\|^2 \leq 3\|\nabla f(\widetilde{\mathbf{W}}_{t,[r]})\|^2 + 3\zeta_r^2 + 3L^2\psi_{t,r}^{(s)}.$$

Then, summing over $s = 0,\ldots,\tau-1$, we get

$$\sum_{s=0}^{\tau-1}\frac{1}{K_r}\sum_{i:\mathcal{B}_r\in\mathcal{B}_i}\mathbb{E}\|\nabla g_i(\widetilde{\mathbf{W}}_t^{(s)})\|^2 \leq \tau(3\|\nabla f(\widetilde{\mathbf{W}}_{t,[r]})\|^2 + 3\zeta_r^2) + 3L^2\sum_{s=0}^{\tau-1}\psi_{t,r}^{(s)}.$$

Now plug the above back into equation 15 and use Lemma 1 to get

$$\begin{aligned}\mathbb{E}\left[f(\widetilde{\mathbf{W}}_{t+1})\right] \leq & f(\widetilde{\mathbf{W}}_t) - \left(\frac{\eta_t\tau}{2} - 6L\eta_t^2\tau^2\right)\|\nabla f(\widetilde{\mathbf{W}}_t)\|^2 + \left(\frac{\eta_t L^2}{2} + 6L^3\eta_t^2\tau\right)\sum_{r=0}^{R}\sum_{s=0}^{\tau-1}\psi_{t,r}^{(s)} \\ & + 6L\eta_t^2\tau^2\sum_{r=0}^{R}\zeta_r^2 + 2L\eta_t^2\tau\sum_{r=0}^{R}\frac{\sigma^2}{K_r}.\end{aligned} \quad (16)$$

Then, for bounding the summations with $\psi_{t,r}^{(s)}$, firstly note the fact that $\widetilde{\mathbf{W}}_{t,[i]}^{(s+1)} - \widetilde{\mathbf{W}}_{t,[r]} = \widetilde{\mathbf{W}}_{t,[i]}^{(s)} - \widetilde{\mathbf{W}}_{t,[r]} - \eta_t\left(\nabla g_i(\widetilde{\mathbf{W}}_t^{(s)}) + \boldsymbol{\xi}_{t,i}^{(s)}\right)$, where $\boldsymbol{\xi}_{t,i}^{(s)} := \nabla\widetilde{\ell}(\cdot) - \nabla g_i(\widetilde{\mathbf{W}}_t^{(s)})$, with zero mean and $\mathbb{E}\|\boldsymbol{\xi}_{t,i}^{(s)}\|^2 \leq \sigma^2$. Therefore,

$$\begin{aligned}\mathbb{E}\|\widetilde{\mathbf{W}}_{t,[i]}^{(s)} - \widetilde{\mathbf{W}}_{t,[r]}\|^2 \leq & \mathbb{E}\|\sum_{h=0}^{s-1}-\eta_t(\nabla g_i(\widetilde{\mathbf{W}}_t^{(h)}) + \boldsymbol{\xi}_{t,i}^{(h)})\|^2 \\ \leq & s\sum_{h=0}^{s-1}\eta_t^2\mathbb{E}\|\nabla g_i(\widetilde{\mathbf{W}}_t^{(h)}) + \boldsymbol{\xi}_{t,i}^{(h)}\|^2 \\ \leq & \eta_t^2 s\sum_{h=0}^{s-1}\mathbb{E}\|\nabla g_i(\widetilde{\mathbf{W}}_t^{(h)})\|^2 + \eta_t^2 s\sum_{h=0}^{s-1}\mathbb{E}\|\boldsymbol{\xi}_{t,i}^{(h)}\|^2.\end{aligned} \quad (17)$$

Averaging over clients and recalling the definition of $\psi_{t,r}^{(s)}$, we can derive

$$\psi_{t,r}^{(s)} \leq \eta_t^2 s\sum_{h=0}^{s-1}\left(\frac{1}{K_r}\sum_{i:\mathcal{B}_r\in\mathcal{B}_i}\mathbb{E}\|\nabla g_i(\widetilde{\mathbf{W}}_t^{(h)})\|^2\right) + \eta_t^2 s\sum_{h=0}^{s-1}\left(\frac{1}{K_r}\sum_{i:\mathcal{B}_r\in\mathcal{B}_i}\mathbb{E}\|\boldsymbol{\xi}_{t,i}^{(h)}\|^2\right)$$

$$\psi_{t,r}^{(s)} \leq \eta_t^2 s\sum_{h=0}^{s-1}\left(3\|\nabla f(\widetilde{\mathbf{W}}_{t,[r]})\|^2 + 3\zeta_r^2 + 3L^2\psi_{t,r}^{(h)}\right) + \eta_t^2 s^2\sigma^2.$$

Summing them from $s = 0$ to $\tau - 1$ gives rise to

$$\Psi_{t,r} \leq 3\eta_t^2 \sum_{s=0}^{\tau-1} s \sum_{h=0}^{s-1} \|\nabla f(\widetilde{\mathbf{W}}_{t,[r]})\|^2 + 3\eta_t^2 \sum_{s=0}^{\tau-1} s \sum_{h=0}^{s-1} \zeta_r^2 + 3L^2\eta_t^2 \sum_{s=0}^{\tau-1} s \sum_{h=0}^{s-1} \psi_{t,r}^{(h)} + \frac{1}{3}\eta_t^2\tau^3\sigma^2,$$

where we used the fact that $\sum_{s=0}^{\tau-1} s^2 \leq \frac{1}{3}H^3$. We can further simplify it as

$$\begin{aligned}
\Psi_{t,r} &\leq \eta_t^2\tau^3\|\nabla f(\widetilde{\mathbf{W}}_{t,[r]})\|^2 + 2\eta_t^2\tau^2 L^2 \sum_{s=0}^{\tau-1} \psi_{t,r}^{(s)} + \eta_t^2\tau^3\zeta_r^2 + \frac{1}{3}\eta_t^2\tau^3\sigma^2 \\
&\leq \eta_t^2\tau^3\|\nabla f(\widetilde{\mathbf{W}}_{t,[r]})\|^2 + 2\eta_t^2\tau^2 L^2 \Psi_{t,r} + \eta_t^2\tau^3\zeta_r^2 + \frac{1}{3}\eta_t^2\tau^3\sigma^2.
\end{aligned} \tag{18}$$

Reorganizing it gives rise to

$$(1 - 2\eta_t^2\tau^2 L^2)\Psi_{t,r} \leq \eta_t^2\tau^3\|\nabla f(\widetilde{\mathbf{W}}_{t,[r]})\|^2 + \eta_t^2\tau^3\zeta_r^2 + \frac{1}{3}\eta_t^2\tau^3\sigma^2.$$

Then, setting $\eta_t \leq \frac{1}{2L\tau}$ such that $1 - 2L^2\eta_t^2\tau^2 \geq \frac{1}{2}$, we have $\Psi_{t,r} \leq 2\eta_t^2\tau^3\left(\|\nabla f(\widetilde{\mathbf{W}}_{t,[r]})\|^2 + \zeta_r^2 + \frac{1}{3}\sigma^2\right)$. Summing over $r$ and utilizing Lemma 1, we derive

$$\sum_{r=0}^{R} \Psi_{t,r} \leq 2\eta_t^2\tau^3\left(\|\nabla f(\widetilde{\mathbf{W}}_t)\|^2 + \sum_{r=0}^{R} \zeta_r^2 + \frac{R+1}{3}\sigma^2\right),$$

which when plugged back into equation 16 gives rise to

$$\begin{aligned}
\mathbb{E}\left[f(\widetilde{\mathbf{W}}_{t+1})\right] \leq & f(\widetilde{\mathbf{W}}_t) - \left(\frac{\eta_t\tau}{2} - 6L\eta_t^2\tau^2\right)\|\nabla f(\widetilde{\mathbf{W}}_t)\|^2 \\
&+ 2\eta_t^2\tau^3\left(\frac{\eta_t L^2}{2} + 6L^3\eta_t^2\tau\right)\left(\|\nabla f(\widetilde{\mathbf{W}}_t)\|^2 + Z + \frac{R+1}{3}\sigma^2\right) \\
&+ 6L\eta_t^2\tau^2 Z + 2L\eta_t^2\tau\sigma^2 C_K,
\end{aligned} \tag{19}$$

where $Z := \sum_{r=0}^{R} \zeta_r^2$ and $C_K := \sum_{r=0}^{R} \frac{1}{K_r}$. Then group the terms in $\|\nabla f(\widetilde{\mathbf{W}}_t)\|^2$ to get $\eta\tau\left(\frac{1}{2} - 6L\eta_t\tau - L^2\eta_t^2\tau^2 - 12L^3\eta_t^3\tau^3\right)$. Next, by setting the learning rate as $\eta_t \leq \min\left\{\frac{1}{48L\tau}, \frac{1}{\sqrt{8}L\tau}, \left(\frac{1}{96L^3\tau^3}\right)^{\frac{1}{3}}\right\}$, we give rise to

$$\begin{aligned}
\frac{\eta_t}{8}\|\nabla f(\widetilde{\mathbf{W}}_t)\|^2 \leq & f(\widetilde{\mathbf{W}}_t) - \mathbb{E}\left[f(\widetilde{\mathbf{W}}_{t+1})\right] \\
&+ 2\eta_t^2\tau^3\left(\frac{\eta_t L^2}{2} + 6L^3\eta_t^2\tau\right)\left(Z + \frac{R+1}{3}\sigma^2\right) \\
&+ 6L\eta_t^2\tau^2 Z + 2L\eta_t^2\tau\sigma^2 C_K.
\end{aligned} \tag{20}$$

Moreover, enforcing $6L^3\eta_t^2\tau \leq \frac{\eta_t L^2}{2}$ (when $\eta_t \leq \frac{1}{12L\tau}$, which is weaker than $\eta_t \leq \frac{1}{48L\tau}$), we can further derive

$$\begin{aligned}
\frac{\eta_t}{8}\|\nabla f(\widetilde{\mathbf{W}}_t)\|^2 \leq & f(\widetilde{\mathbf{W}}_t) - \mathbb{E}\left[f(\widetilde{\mathbf{W}}_{t+1})\right] + 2\eta_t^2\tau^3\left(\eta_t L^2\right)\left(Z + \frac{R+1}{3}\sigma^2\right) \\
&+ 6L\eta_t^2\tau^2 Z + 2L\eta_t^2\tau\sigma^2 C_K.
\end{aligned} \tag{21}$$

Now summing from $t = 0$ to $T - 1$, we obtain

$$\frac{1}{\sum_{t=0}^{T-1} \eta_t} \sum_{t=0}^{T-1} \eta_t \mathbb{E}\|\nabla f(\widetilde{\mathbf{W}}_t)\|^2 \leq \frac{8\left(f(\widetilde{\mathbf{W}}_0) - f(\widetilde{\mathbf{W}}_T)\right)}{\sum_{t=0}^{T-1} \eta_t} + 16\left(Z + \frac{R+1}{3}\sigma^2\right) \frac{\tau^3 L^2}{\sum_{t=0}^{T-1} \eta_t} \sum_{t=0}^{T-1} \eta_t^3$$

$$+ 48L\tau^2 Z \frac{1}{\sum_{t=0}^{T-1} \eta_t} \sum_{t=0}^{T-1} \eta_t^2 + 16L\tau\sigma^2 C_K \frac{1}{\sum_{t=0}^{T-1} \eta_t} \sum_{t=0}^{T-1} \eta_t^2,$$

$$(22)$$

which completes the proof.

## C ADDITIONAL EXPERIMENTS

In this section, we present some experimental results in addition to those presented in Sec. 5.2. Similar to Sec. 5.2, unless stated otherwise, results outlined in bold signify the highest performing method, with the second best underlined.

### C.1 POST-FL TRAINING FOR $2 \cdot P$ ITERATIONS VS. TAP

Since the proposed TAP algorithm relies on $\widetilde{\mathbf{W}}_{T,[i]}$ training for $P$ mini-batch iterations after FL before also training $\mathbf{X}_{T,[i]}$ for $P$ iterations with $\widetilde{\mathbf{W}}_{T,[i]}$ as the teacher model, we seek to see if $\mathbf{X}_{T,[i]}$ trained via the KD-based post-FL process still achieves better performance than $\widetilde{\mathbf{W}}_{T,[i]}$ being trained for $2 \cdot P$ iterations. We do not include FedDAT (Chen et al., 2024) in this experiment, as its optimization procedure involves sequentially updating the aggregated and personal adapters, resulting in the same number of forward passes as required by TAP during post-FL training. For similar reasons, we also do not include Per-FedAvg (Fallah et al., 2020), as it must conduct two passes and obtain two differing gradients for an inner and outer update step.

Table 8: TAP vs. DisentAFL and FedAvg trained for $2P$ iterations on image tasks for ViLT.

| Model | Method | FMNIST | Caltech-256 | Avg. Gen. |
|---|---|---|---|---|
| | | MSE ($\downarrow$) | MSE ($\downarrow$) | MSE ($\downarrow$) |
| ViLT | FedAvg + $2 \cdot P$ | **0.5419** $\pm$ 0.0450 | 0.3971 $\pm$ 0.1172 | **0.4840** |
| | DisentAFL + $2 \cdot P$ | 0.5952 $\pm$ 0.0032 | 0.4061 $\pm$ 0.0048 | 0.5196 |
| | **TAP (Ours)** | 0.5467 $\pm$ 0.0409 | **0.3949** $\pm$ 0.1148 | 0.4860 |

Table 9: TAP vs. DisentAFL and FedAvg trained for $2P$ iterations on image tasks for FLAVA.

| Model | Method | Tiny-ImageNet | CIFAR-100 | FMNIST | Caltech-256 | Avg. Class. | Avg. Gen. |
|---|---|---|---|---|---|---|---|
| | | Acc ($\uparrow$) | Acc ($\uparrow$) | MSE ($\downarrow$) | MSE ($\downarrow$) | Acc ($\uparrow$) | MSE ($\downarrow$) |
| FLAVA | FedAvg + $2 \cdot P$ | 44.92 $\pm$ 7.11 | 55.35 $\pm$ 5.40 | 0.5595 $\pm$ 0.0059 | **0.4244** $\pm$ 0.0140 | 50.13 | 0.4694 |
| | DisentAFL + $2 \cdot P$ | 36.08 $\pm$ 0.14 | 8.45 $\pm$ 0.65 | 0.7247 $\pm$ 0.0010 | 0.5075 $\pm$ 0.0054 | 22.26 | 0.5799 |
| | **TAP (Ours)** | **47.06** $\pm$ 6.80 | **56.26** $\pm$ 5.44 | **0.5572** $\pm$ 0.0083 | 0.4252 $\pm$ 0.0258 | **51.66** | **0.4692** |

Table 10: TAP vs. DisentAFL and FedAvg trained for $2P$ iterations on text tasks.

| Model | Method | AG News | MMLU | | VQA | | CommonGen | | Avg. Gen. | |
|---|---|---|---|---|---|---|---|---|---|---|
| | | Acc ($\uparrow$) | BS ($\uparrow$) | METEOR ($\uparrow$) | BS ($\uparrow$) | METEOR ($\uparrow$) | BS ($\uparrow$) | METEOR ($\uparrow$) | BS ($\uparrow$) | METEOR ($\uparrow$) |
| FLAVA | FedAvg + $2 \cdot P$ | **92.99** $\pm$ 0.22 | 38.35 $\pm$ 1.26 | 10.57 $\pm$ 1.24 | 54.03 $\pm$ 12.31 | 37.80 $\pm$ 19.76 | 32.06 $\pm$ 1.91 | 13.62 $\pm$ 3.17 | 42.11 | 22.68 |
| | DisentAFL + $2 \cdot P$ | 92.63 $\pm$ 0.09 | 39.90 $\pm$ 0.65 | **21.23** $\pm$ 0.97 | 38.94 $\pm$ 0.33 | 23.07 $\pm$ 1.08 | 31.01 $\pm$ 0.24 | 14.30 $\pm$ 0.09 | 35.96 | 19.19 |
| | **TAP (Ours)** | 92.66 $\pm$ 0.34 | **45.81** $\pm$ 4.32 | 19.80 $\pm$ 3.83 | **69.38** $\pm$ 7.47 | **47.60** $\pm$ 14.92 | **34.01** $\pm$ 2.96 | **17.17** $\pm$ 4.56 | **50.52** | **29.87** |
| ViLT | FedAvg + $2 \cdot P$ | **78.07** $\pm$ 2.29 | 44.47 $\pm$ 3.09 | 11.94 $\pm$ 5.21 | 62.58 $\pm$ 13.70 | 43.61 $\pm$ 21.68 | 39.65 $\pm$ 3.06 | 18.73 $\pm$ 7.60 | 49.78 | 27.33 |
| | DisentAFL + $2 \cdot P$ | 76.54 $\pm$ 0.52 | 54.91 $\pm$ 0.35 | **40.26** $\pm$ 0.31 | 56.99 $\pm$ 0.68 | 56.65 $\pm$ 0.18 | 39.23 $\pm$ 0.71 | 24.68 $\pm$ 0.85 | 49.47 | 40.58 |
| | **TAP (Ours)** | 77.44 $\pm$ 2.63 | **56.43** $\pm$ 3.12 | 40.79 $\pm$ 1.61 | **72.99** $\pm$ 7.47 | **62.52** $\pm$ 9.09 | **41.46** $\pm$ 1.45 | **25.35** $\pm$ 1.85 | **57.07** | **43.31** |

Based off the results of Tables 8, 9, and 10, we see that across a majority of tasks, TAP still outperforms both FedAvg + $2 \cdot P$ and DisentAFL + $2 \cdot P$. In addition, we see that when either FedAvg +

$2 \cdot P$ or DisentAFL $+ 2 \cdot P$ is better in performance, TAP always remains close to the best performing baseline, as seen with examples of $0.4252$ (TAP) vs. $0.4244$ (FedAvg $+ 2 \cdot P$) for Caltech-256 on FLAVA and $0.5467$ (TAP) vs. $0.5419$ (FedAvg $+ 2 \cdot P$) for FMNIST on ViLT. This demonstrates that the final personalized model produced by TAP induces greater levels of personalization in comparison to merely devoting more iterations to $\widetilde{\mathbf{W}}_{T,[i]}$.

## C.2 Ablation on Margin Hyperparameters

Table 11: Ablation study on margin hyperparameters $m_{i,o}$ for image datasets on ViLT.

| Model | Margin | FMNIST | Caltech-256 | Avg. Gen. |
| --- | --- | --- | --- | --- |
| | | MSE ($\downarrow$) | MSE ($\downarrow$) | MSE ($\downarrow$) |
| **ViLT** | 0.01 / 0.04 | $0.5460 \pm 0.0413$ | $0.3948 \pm 0.1158$ | 0.4855 |
| | 0.02 / 0.05 | $0.5582 \pm 0.0436$ | $0.4035 \pm 0.1158$ | 0.4963 |
| | 0.05 / 0.1 | $0.5807 \pm 0.0338$ | $0.4281 \pm 0.1198$ | 0.5197 |

Table 12: Ablation study on margin hyperparameters $m_{i,o}$ for image datasets on FLAVA.

| Model | Margin | Tiny-ImageNet | CIFAR-100 | FMNIST | Caltech-256 | Avg. Class. | Avg. Gen. |
| --- | --- | --- | --- | --- | --- | --- | --- |
| | | Acc ($\uparrow$) | Acc ($\uparrow$) | MSE ($\downarrow$) | MSE ($\downarrow$) | Acc ($\uparrow$) | MSE ($\downarrow$) |
| **FLAVA** | 0.01 / 0.04 | $46.82 \pm 6.83$ | $55.46 \pm 5.67$ | $0.5587 \pm 0.0102$ | $0.4285 \pm 0.0296$ | 51.14 | 0.4719 |
| | 0.02 / 0.05 | $46.80 \pm 6.80$ | $55.34 \pm 5.63$ | $0.5636 \pm 0.0078$ | $0.4306 \pm 0.0324$ | 51.07 | 0.4749 |
| | 0.05 / 0.1 | $46.30 \pm 7.01$ | $54.74 \pm 6.34$ | $0.5788 \pm 0.0191$ | $0.4713 \pm 0.6555$ | 50.52 | 0.5071 |

Table 13: Ablation study on margin hyperparameters $m_{i,o}$ for text datasets.

| Model | Margin | AG News | MMLU | | VQA | | CommonGen | | Avg. Gen. | |
| --- | --- | --- | --- | --- | --- | --- | --- | --- | --- | --- |
| | | Acc ($\uparrow$) | BS ($\uparrow$) | METEOR ($\uparrow$) | BS ($\uparrow$) | METEOR ($\uparrow$) | BS ($\uparrow$) | METEOR ($\uparrow$) | BS ($\uparrow$) | METEOR ($\uparrow$) |
| **FLAVA** | 0.01 / 0.04 | $92.58 \pm 0.37$ | $45.81 \pm 4.32$ | $19.80 \pm 3.83$ | $69.38 \pm 7.47$ | $47.60 \pm 14.92$ | $34.98 \pm 2.88$ | $17.28 \pm 4.60$ | 50.91 | 29.91 |
| | 0.02 / 0.05 | $92.46 \pm 0.36$ | $45.81 \pm 4.32$ | $19.80 \pm 3.83$ | $69.38 \pm 7.47$ | $47.60 \pm 14.92$ | $35.98 \pm 3.26$ | $17.53 \pm 4.50$ | 51.31 | 30.01 |
| | 0.05 / 0.1 | $92.03 \pm 0.34$ | $45.81 \pm 4.32$ | $19.80 \pm 3.83$ | $69.38 \pm 7.47$ | $47.60 \pm 14.92$ | $37.11 \pm 2.28$ | $18.17 \pm 4.69$ | 51.76 | 30.27 |
| **ViLT** | 0.01 / 0.04 | $77.44 \pm 2.63$ | $56.43 \pm 3.12$ | $40.79 \pm 1.61$ | $72.99 \pm 7.47$ | $62.52 \pm 9.09$ | $41.72 \pm 1.43$ | $25.47 \pm 1.88$ | 57.17 | 43.46 |
| | 0.02 / 0.05 | $77.68 \pm 2.43$ | $56.43 \pm 3.12$ | $40.79 \pm 1.61$ | $72.99 \pm 7.47$ | $62.52 \pm 9.09$ | $41.68 \pm 1.05$ | $25.77 \pm 1.30$ | 57.15 | 43.47 |
| | 0.05 / 0.1 | $77.31 \pm 2.22$ | $56.43 \pm 3.12$ | $40.79 \pm 1.61$ | $72.99 \pm 7.47$ | $62.52 \pm 9.09$ | $41.83 \pm 0.91$ | $25.71 \pm 1.59$ | 57.22 | 43.45 |

Next, we consider an ablation study on how the margin hyperparameters $m_{i,o}$ influences the performance of TAP. Similar to the style outlined in Table 22, we consider two differing margin values for each setting depending on the task. For each column, the lower margin value (e.g., 0.01 in the 0.01 / 0.04 column) apply to tasks pertaining to image generation and task classification. The higher margin value is used for all other tasks. Results are given in Tables 11, 12, and 13.

Based off the results outlined in Tables 11 and 12, we see that image-aligned tasks benefit from having lower $m_{i,o}$ values. For example, the average classification and generation performance for $m_{i,o} = 0.01$ or $0.04$ is $51.14$ and $0.4719$ while it is $50.52$ and $0.5071$ when $m_{i,o} = 0.05$ or $0.1$ on FLAVA. We also note similar characteristics for image-related tasks on ViLT. This means for image-aligned tasks, being more restrictive and setting higher margin values could inhibit $\mathbf{X}_{[i]}$ from regularly taking advantage of the parameters of $\widetilde{\mathbf{W}}_{[i]}$.

In Table 13, which deals with text-aligned tasks, we note that VQA and MMLU maintains identical performance across all margins. In terms of other tasks, we note minimal changes in performance between differing explored margin values. Overall, this indicates that for text-aligned tasks, the TAP algorithm is not overly sensitive to the values set for $m_{i,o}$.

## C.3 Number of Replacements

In Fig. 3, we present the number of cumulative replacements taken place over FL training for each task. We note that for a majority of the tasks, most of the replacements take place in the earlier stages of training and then indicate a trend toward leveling off, signifying that reliance on the FL-trained model is most beneficial when the models are still focusing on the basic structure and characteristics

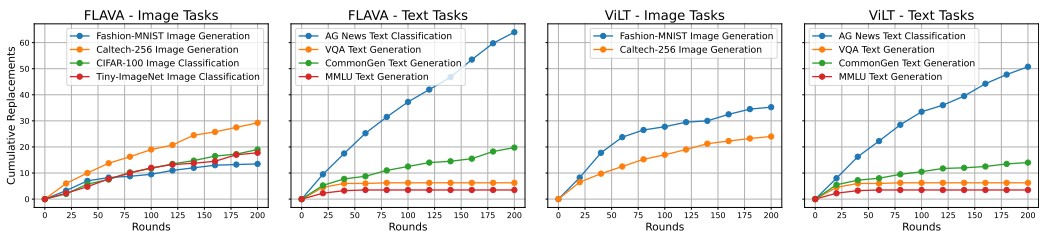

Figure 3: Number of replacements over FL training on FLAVA and ViLT models.

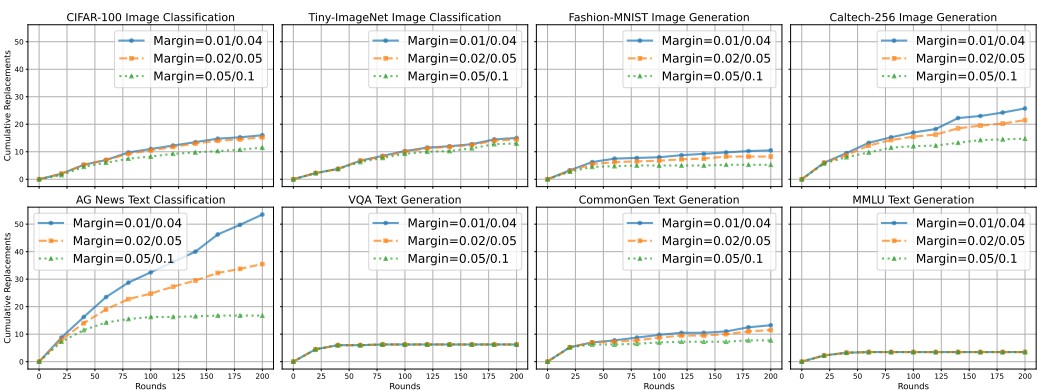

(a) Number of replacements for differing margin settings on FLAVA.

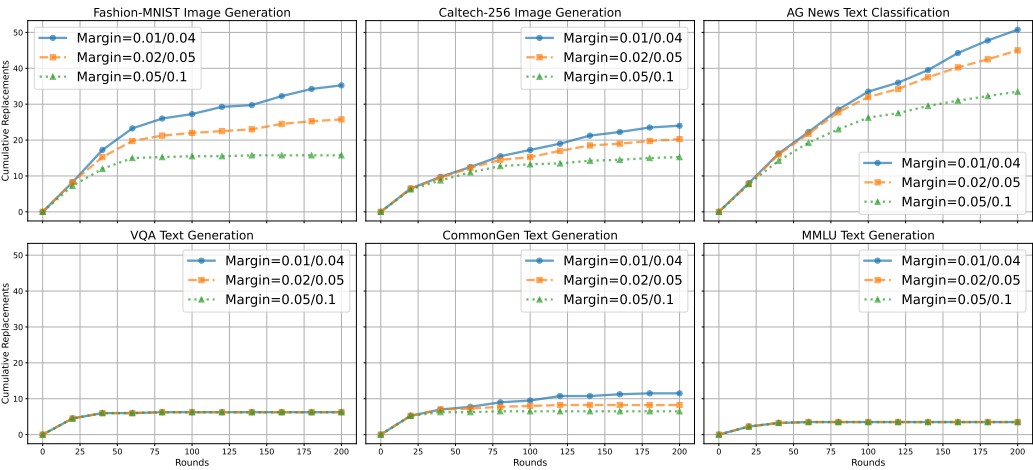

(b) Number of replacements for differing margin settings on ViLT.

Figure 4: Replacement comparisons for differing margins on each task across FLAVA and ViLT.

of the tasks (Frankle et al., 2020). Afterwards, the personalized models rely less on the model returned from the server, allowing them to better adjust to their local datasets. We note that for AG News, as outlined in Table 3, slightly under performs the baselines, which means more replacements take place in comparison with other tasks. This indicates that the TAP algorithm's margin-based method of replacement is an effective means of identifying when replacement is useful, as the $\widetilde{\mathbf{W}}_{[i]}$ model for the AG News task regularly exhibits beneficial information for $\mathbf{X}_{[i]}$.

Moreover, we also consider how replacements are affected by the differing margin values explored in Appendix C.2. Based on the results from Fig. 4a (FLAVA) and Fig. 4b (ViLT), we note that when we make the threshold for replacement higher, i.e., larger $m_{i,o}$ values, the number of cumulative replacements generally decreases and levels off quicker in comparison to smaller $m_{i,o}$ margins. For

example, with AG News, we see for margin values of either 0.01 or 0.04, a linear-like increase of cumulative replacements takes place; with $m_{i,o}$ as either 0.05 or 0.1, we observe lower total replacements ($\sim 50$ vs. $\sim 18$ replacements on FLAVA and $\sim 50$ vs. $\sim 30$ replacements on ViLT) and a noticeable leveling off around communication round 100. With VQA and MMLU, we note that the number of replacements remains unchanged across all settings, which corroborates with the results from Appendix C.2, where the BS and METEOR scores remain unchanged across differing margins. Overall, these outcomes indicate that in general, the margin hyperparameters play an important role in shaping $\mathbf{X}_{[i]}$'s interaction with FL-engaged $\widetilde{\mathbf{W}}_{[i]}$.

## C.4 TAP vs. Standard Transformer Architecture

Table 14: Standard transformer backbone with baselines vs. TAP on image datasets for ViLT.

| Model | Method | FMNIST | Caltech-256 | Avg. Gen. |
| --- | --- | --- | --- | --- |
| | | MSE ($\downarrow$) | MSE ($\downarrow$) | MSE ($\downarrow$) |
| ViLT | FedAvg | $0.6274 \pm 0.0243$ | $0.6192 \pm 0.0873$ | 0.6089 |
| | FedAvg + Post-train | $0.6097 \pm 0.0201$ | $0.5701 \pm 0.0816$ | 0.5939 |
| | Per-FedAvg | $0.7476 \pm 0.0726$ | $1.0180 \pm 0.2944$ | 0.8558 |
| | Per-FedAvg + Post-train | $0.6367 \pm 0.0310$ | $0.6436 \pm 0.0703$ | 0.6395 |
| | FedDAT | $0.5648 \pm 0.0188$ | $0.4332 \pm 0.0571$ | 0.5121 |
| | FedDAT + Post-train | $\underline{0.5612} \pm 0.0179$ | $\underline{0.4235} \pm 0.0566$ | $\underline{0.5061}$ |
| | **TAP (Ours)** | $\mathbf{0.5467} \pm 0.0409$ | $\mathbf{0.3949} \pm 0.1148$ | **0.4860** |

Table 15: Standard transformer backbone with baselines vs. TAP on image datasets for FLAVA.

| Model | Method | Tiny-ImageNet | CIFAR-100 | FMNIST | Caltech-256 | Avg. Class. | Avg. Gen. |
| --- | --- | --- | --- | --- | --- | --- | --- |
| | | Acc ($\uparrow$) | Acc ($\uparrow$) | MSE ($\downarrow$) | MSE ($\downarrow$) | Acc ($\uparrow$) | MSE ($\downarrow$) |
| FLAVA | FedAvg | $27.93 \pm 12.69$ | $35.25 \pm 12.57$ | $0.5996 \pm 0.0163$ | $0.6117 \pm 0.0555$ | 31.59 | 0.6077 |
| | FedAvg + Post-train | $\underline{29.55} \pm 12.49$ | $37.08 \pm 12.52$ | $0.5947 \pm 0.0150$ | $0.5959 \pm 0.0533$ | $\underline{33.32}$ | 0.5955 |
| | Per-FedAvg | $26.93 \pm 13.76$ | $35.10 \pm 12.45$ | $0.7397 \pm 0.0475$ | $0.8654 \pm 0.0932$ | 31.02 | 0.8204 |
| | Per-FedAvg + Post-train | $28.70 \pm 13.80$ | $\underline{37.59} \pm 11.93$ | $0.6081 \pm 0.0169$ | $0.6151 \pm 0.0420$ | 33.14 | 0.6128 |
| | FedDAT | $17.65 \pm 16.77$ | $30.45 \pm 18.06$ | $0.5815 \pm 0.0125$ | $0.5655 \pm 0.0530$ | 24.05 | 0.5709 |
| | FedDAT + Post-train | $19.11 \pm 17.29$ | $32.60 \pm 18.88$ | $\underline{0.5782} \pm 0.0122$ | $\underline{0.5550} \pm 0.0513$ | 25.85 | $\underline{0.5628}$ |
| | **TAP (Ours)** | $\mathbf{47.06} \pm 6.80$ | $\mathbf{56.26} \pm 5.44$ | $\mathbf{0.5572} \pm 0.0083$ | $\mathbf{0.4252} \pm 0.0258$ | **51.66** | **0.4692** |

Table 16: Standard transformer backbone with baselines vs. TAP on text datasets.

| Model | Method | AG News | MMLU | | VQA | | CommonGen | | Avg. Gen. | |
| --- | --- | --- | --- | --- | --- | --- | --- | --- | --- | --- |
| | | Acc ($\uparrow$) | BS ($\uparrow$) | METEOR ($\uparrow$) | BS ($\uparrow$) | METEOR ($\uparrow$) | BS ($\uparrow$) | METEOR ($\uparrow$) | BS ($\uparrow$) | METEOR ($\uparrow$) |
| FLAVA | FedAvg | $90.48 \pm 0.60$ | $41.67 \pm 1.24$ | $21.39 \pm 2.61$ | $48.32 \pm 1.91$ | $10.88 \pm 2.52$ | $30.37 \pm 1.65$ | $12.51 \pm 2.17$ | 39.81 | 13.63 |
| | FedAvg + Post-train | $90.68 \pm 0.54$ | $\underline{41.90} \pm 0.96$ | $\underline{21.68} \pm 2.37$ | $47.88 \pm 1.91$ | $10.48 \pm 2.24$ | $30.39 \pm 1.47$ | $13.17 \pm 2.97$ | 39.69 | 13.80 |
| | Per-FedAvg | $89.64 \pm 1.45$ | $41.46 \pm 1.99$ | $20.31 \pm 3.93$ | $47.12 \pm 2.49$ | $10.20 \pm 2.10$ | $30.60 \pm 1.48$ | $13.19 \pm 2.72$ | 39.38 | 13.42 |
| | Per-FedAvg + Post-train | $\underline{90.72} \pm 0.58$ | $41.87 \pm 1.14$ | $20.85 \pm 2.19$ | $47.31 \pm 2.24$ | $10.01 \pm 1.34$ | $30.42 \pm 1.30$ | $13.57 \pm 2.70$ | 39.47 | 13.60 |
| | FedDAT | $90.21 \pm 0.68$ | $41.63 \pm 1.05$ | $21.06 \pm 1.96$ | $\underline{67.65} \pm 11.40$ | $\mathbf{57.74} \pm 12.10$ | $\underline{40.86} \pm 2.56$ | $\underline{25.42} \pm 1.77$ | **51.73** | **37.48** |
| | FedDAT + Post-train | $90.62 \pm 0.65$ | $41.88 \pm 0.75$ | $\mathbf{21.88} \pm 2.52$ | $66.97 \pm 10.98$ | $\underline{57.23} \pm 11.79$ | $41.13 \pm 1.94$ | $\mathbf{25.50} \pm 1.74$ | $\underline{51.62}$ | $\underline{37.47}$ |
| | **TAP (Ours)** | $\mathbf{92.66} \pm 0.34$ | $\mathbf{45.81} \pm 4.32$ | $19.80 \pm 3.83$ | $\mathbf{69.38} \pm 7.47$ | $47.60 \pm 14.92$ | $34.01 \pm 2.96$ | $17.17 \pm 4.56$ | 50.52 | 29.87 |
| ViLT | FedAvg | $79.88 \pm 6.30$ | $54.91 \pm 2.00$ | $\underline{41.08} \pm 3.53$ | $55.58 \pm 4.27$ | $39.73 \pm 11.87$ | $40.51 \pm 1.51$ | $\mathbf{25.96} \pm 1.67$ | 49.42 | 34.49 |
| | FedAvg + Post-train | $82.39 \pm 3.07$ | $55.16 \pm 0.97$ | $40.47 \pm 1.30$ | $55.08 \pm 3.51$ | $39.42 \pm 10.86$ | $40.62 \pm 0.99$ | $\underline{25.66} \pm 1.88$ | 49.31 | 34.12 |
| | Per-FedAvg | $68.38 \pm 11.39$ | $55.51 \pm 3.02$ | $39.01 \pm 1.52$ | $54.99 \pm 2.87$ | $31.29 \pm 8.29$ | $39.94 \pm 3.43$ | $23.53 \pm 4.66$ | 49.07 | 29.73 |
| | Per-FedAvg + Post-train | $81.68 \pm 3.88$ | $55.03 \pm 0.89$ | $\mathbf{41.73} \pm 1.85$ | $55.20 \pm 3.24$ | $30.02 \pm 6.45$ | $40.40 \pm 1.56$ | $24.25 \pm 3.03$ | 49.24 | 30.06 |
| | FedDAT | $\underline{85.95} \pm 0.95$ | $54.74 \pm 1.99$ | $40.35 \pm 2.94$ | $\underline{66.61} \pm 10.48$ | $\underline{60.54} \pm 6.41$ | $\mathbf{43.32} \pm 5.71$ | $25.58 \pm 1.89$ | $\underline{54.34}$ | $\underline{42.52}$ |
| | FedDAT + Post-train | $\mathbf{86.72} \pm 0.46$ | $\underline{55.30} \pm 1.09$ | $40.77 \pm 1.15$ | $65.92 \pm 10.32$ | $60.44 \pm 6.33$ | $\underline{41.77} \pm 0.96$ | $25.43 \pm 1.89$ | 54.13 | 42.50 |
| | **TAP (Ours)** | $77.44 \pm 2.63$ | $\mathbf{56.43} \pm 3.12$ | $40.79 \pm 1.61$ | $\mathbf{72.99} \pm 7.47$ | $\mathbf{62.52} \pm 9.09$ | $41.46 \pm 1.45$ | $25.35 \pm 1.85$ | **57.07** | **43.31** |

Here, we compare the usage of TAP with the model architecture adopted from Chen & Zhang (2024) against a more standard architecture with a non modality-task pair-aware server. In this scenario, each client employs a standard transformer layer structure, with no MoE routing taking place (i.e., each client shares the same backbone), which is sent to the server for vanilla FedAvg (not component-wise) aggregation (McMahan et al., 2017).

From Tables 14, 15, and 16, we note that TAP outperforms the baselines on a majority of the tasks evaluated. Even when TAP lags behind one of the baselines, such as with TAP vs. FedDAT on CommonGen, it is the most consistent method across all tasks. For example, TAP retains an average BS score and image accuracy of 50.52 and 51.66 respectively on FLAVA. By contrast, FedDAT + Post-train obtains 51.62 and 25.85, indicating that while FedDAT + Post-train achieves marginally better average BS scores, it significantly underperforms TAP on image classification. Overall, this demonstrates the merit of adopting a model architecture that is modality-task pair aware, to bet-

ter prevent conflicting tasks from competing with one another and ensuring better consistency in performance across all tasks.

## C.5 STATISTICAL SIGNIFICANCE OF TAP'S IMPROVEMENT VS. BASELINES

Table 17: Percentage of metrics for which TAP exhibits superior performance and is statistically significant with a significance threshold of $0.05$ in comparison to each baseline.

| Model | Baseline | % Metrics TAP Significantly Better |
|---|---|---|
| **FLAVA** | Local | 90.9% |
| | FedAvg | 90.9% |
| | FedAvg + Post-train | 63.6% |
| | Per-FedAvg | 90.9% |
| | Per-FedAvg + Post-train | 63.6% |
| | DisentAFL | 81.8% |
| | DisentAFL + Post-train | 72.7% |
| | FedDAT | 54.5% |
| | FedDAT + Post-train | 45.5% |
| **ViLT** | Local | 55.6% |
| | FedAvg | 100.0% |
| | FedAvg + Post-train | 66.7% |
| | Per-FedAvg | 100.0% |
| | Per-FedAvg + Post-train | 77.8% |
| | DisentAFL | 77.8% |
| | DisentAFL + Post-train | 66.7% |
| | FedDAT | 100.0% |
| | FedDAT + Post-train | 88.9% |

In this section, we seek to see if the improvements exhibited from TAP in Tables 1, 2, and 3 from Sec. 5.2 is statistically significant. In Table 17, we utilize the commonly used Welch $t$-test West (2021) with a standard significance threshold of $0.05$ to see the percentage of metrics for which TAP is both (1) the higher performing method and (2) statistically significant.

From Table 17, we note that across nearly all baselines and settings, TAP achieves statistically significant improvements on the majority of metrics. Even in cases where significance is comparatively lower (e.g., FedDAT vs. ViLT on FLAVA), TAP remains consistently stronger across architectures (e.g., FedDAT vs. TAP on ViLT). These findings give credence that TAP's gains are not only numerically stronger but statistically reliable.

## D  IMPLEMENTATION SPECIFICS

### D.1  DECODER ARCHITECTURE

Here, we outline the architectures utilized for the decoders $\mathbf{W}^{(D)}$ on differing tasks for both FLAVA and ViLT, presented in Tables 18 (classification heads), 19 (image generation head), and 20 (text generation head).

### D.2  HYPERPARAMETER SETUP

In Table 21, general hyperparameters utilized in running our experiments are outlined below. Table 22 details the margin values $m_{i,o}$ utilized in the TAP algorithm to determine whether replacement

Table 18: Settings of MLP-type classification heads (both text and image).

| Type | FLAVA (Singh et al., 2022) | ViLT (Kim et al., 2021) |
|---|---|---|
| Num. of Linear Layers | 2 | 3 |
| Activation Func. | ReLU | GELU |
| Hidden Dimension | embed_dim / 2 | [embed_dim * 4, embed_dim] |
| Dropout | 0.3 | 0.1 |
| LayerNorm Utilized | No | Yes |

Table 19: Architecture of the image generation head used in both FLAVA and ViLT.

| Component | Specification |
|---|---|
| Head Type | Transposed Convolutional Decoder |
| Input Embedding Dimension | 768 |
| Target Image Size | 64×64 |
| Output Channels | 3 (RGB) |
| **Initial Projection** | |
| Layer Type | Linear |
| Output Dimensions | $64 \times 8 \times 8$ |
| **Decoder Layers** | |
| Layer 1: ConvTranspose2d | |
| Input/Output Channels | $64 \rightarrow 32$ |
| Kernel/Stride/Padding | 4/2/1 |
| Output Resolution | 16×16 |
| Activation | ReLU |
| Layer 2: ConvTranspose2d | |
| Input/Output Channels | $32 \rightarrow 16$ |
| Kernel/Stride/Padding | 4/2/1 |
| Output Resolution | 32×32 |
| Activation | ReLU |
| Layer 3: ConvTranspose2d | |
| Input/Output Channels | $16 \rightarrow 3$ |
| Kernel/Stride/Padding | 4/2/1 |
| Output Resolution | 64×64 |

will take place (Stage 1 of Fig. 2). Unless stated otherwise, the following are the settings across all experiments.

### D.3 TEXT TEMPLATES

In Table 23, we specify the template format utilized for each text generation-based dataset (MMLU, VQA, and CommonGen). Portions surrounded with {} brackets indicate where certain data fields are to be inputted.

Table 20: Architecture of the text generation head for both FLAVA and ViLT.

| Component | Specification |
|---|---|
| Head Type | Conditional Autoregressive (GPT-2 based) |
| **Conditioning Mechanism** | |
| Layer Type | Multi-Head Attention |
| # of Attention Heads | 8 |
| Embedding Dimension | 768 |
| **Core Generator** | |
| Architecture | GPT-2 |
| # of Layers | 1 |
| # of Attention Heads | 4 |
| Hidden Dimension | 768 |
| Inner FFN Dimension | 1536 ($2 \times 768$) |
| **Output Projection** | |
| Layer Type | Linear (no bias) |

Table 21: Hyperparameter settings to run numerical experiments.

| Hyperparameters | FLAVA (Singh et al., 2022) | ViLT (Kim et al., 2021) |
|---|---|---|
| Num. of Clients Aggregated each Round | 2 | 2 |
| Batch Size | 128 | 128 |
| LoRA Encoder Attention Rank | 8 | 4 |
| LoRA Backbone Attention Rank | 16 | 8 |
| LoRA Backbone Expert Rank | 4 | 8 |
| LoRA Dropout Rate | 0.3 | 0.3 |
| FedDAT (Chen et al., 2024) Adapter Bottleneck Size | 32 | 16 |
| KD Temperature $\tilde{\tau}$ | 1 | 1 |
| Disent. Loss Weight | 0.5 | 0.5 |
| Image Gen. Distill. Weight | 2e-3 | 2e-3 |
| Text Gen. Distill. Weight | 1 | 1 |
| Image Classification Distill. Weight | 2e-3 | 2e-3 |
| Text Classification Distill. Weight | 2e-3 | 2e-3 |
| FedDAT KL Weight, $\forall o \in \mathcal{O}$ | 2e-3 | 2e-3 |
| Initial Learning Rate $\eta_0$ | 1e-4 | 1e-4 |
| Post-warmup Learning Rate $\eta_t$ | 3e-4 | 4e-4 |
| Per-FedAvg (Fallah et al., 2020) inner update initial learning rate | 1e-4 | 1e-4 |
| Per-FedAvg post-warmup inner update learning rate | 3e-4 | 4e-4 |
| AdamW weight decay | 0.01 | 0.01 |
| Num. of Warmup Rounds | 20 | 20 |
| Num. Local Iterations | 20 | 30 |
| Num. Distillation Iterations $P$ | 50 | 50 |
| Total Communication Rounds $T$ | 200 | 200 |

Table 22: Margin settings to run numerical experiments.

| Task Type | FLAVA (Singh et al., 2022) | ViLT (Kim et al., 2021) |
|---|---|---|
| Image Classification Task(s) | 0.01 | — |
| Image Generation Task(s) | 0.005 | 0.01 |
| Text Classification Task(s) | 0.005 | 0.01 |
| Text Generation Task(s) | 0.01 | 0.02 |

Table 23: Templates utilized for the text generation task-based datasets. The inputs are highlighted in blue, with the model's expected response marked in red.

| Dataset | Template |
|---|---|
| MMLU | The following is a multiple-choice question about {SUBJECT}.
Choose the correct answer from the options below.
Question: {QUESTION}
Options:
{CHOICES}

Correct answer on {SUBJECT}: {ANSWER} |
| VQA | Answer the question based on the image.
Question: {QUESTION}

Answer: {ANSWER} |
| CommonGen | Generate an appropriate description from the concepts below.
Concepts: {CONCEPTS}

Description: {ANSWER} |

