# OpenReview forum: "TAP: Two-Stage Adaptive Personalization of Multi-task and Multi-Modal Foundation Models in Federated Learning"
_ICLR.cc/2026/Conference — Submitted to ICLR 2026_

### Official Review · Reviewer_GYCq · 2025-10-16

**Soundness:** 2
**Presentation:** 2
**Contribution:** 2
**Rating:** 2
**Confidence:** 4

**Summary:**

TAP is a two-step adaptive Personalized Federated Learning algorithm designed to personalize heterogeneous multi-modal and multi-task foundation models. TAP effectively balances global collaboration with client-specific adaptation, supported by theoretical convergence analysis highlighting the limits of a shared server model. By integrating margin-based optimization during federated training, TAP achieves personalization across diverse datasets, tasks, and architectures.

**Strengths:**

1.	The proposed two-step adaptive mechanism (margin-based optimization followed by post-training knowledge distillation) is intuitively motivated and straightforward to implement, potentially allowing integration into existing FL pipelines.

2. The inclusion of convergence analysis provides some theoretical context for why pure global aggregation may be insufficient.

3. The experiments cover multiple datasets and architectures, which helps demonstrate the potential generality of the approach, even though deeper ablations and baselines are needed for stronger validation.

**Weaknesses:**

1.	**Lack of motivating examples**: While the paper highlights the limitations of prior uni-modal and uni-task PFL approaches, it does not provide concrete examples of realistic multi-modal, multi-task federated scenarios (e.g., clients working on paired image–text data with distinct downstream tasks). This omission weakens the motivation for TAP's design. What are the scenarios where TAP can be practically used?

2.	**Insufficient distinction from related work**: The related work section reads as a descriptive overview rather than a comparative positioning of TAP relative to existing PFL, multi-modal, or LoRA-based methods. It should explicitly articulate the novel contributions and technical differentiators that set TAP apart.

3. **Dependence on existing techniques**: The approach leverages LoRA for efficient fine-tuning but does not contribute novel adaptations or theoretical insights specific to LoRA in the PFL context. Hence, the claim of extending PFL to foundation models should be supported with more than existing parameter-efficient methods.

4. **Choice of baselines**: The paper does not include established PFL baselines, limiting the fairness and interpretability of the reported improvements. Comparisons with standard methods such as pFedMe [1], Per-FedAvg [2], or FedPer [3] would strengthen the evaluation.

5. **Unclear utility of knowledge distillation**: Tables 4 and 5 show marginal or no improvement when KD is applied, raising questions about its contribution to the method’s effectiveness.

[1] Personalized Federated Learning with Moreau Envelopes (Canh et al., NeurIPS 2020)

[2] Personalized federated learning with theoretical guarantees: A model-agnostic meta-learning approach (Fallah et al., NeurIPS 2020)

[3] Federated Learning with Personalization Layers (Arivazhagan et al., arXiv:1912.00818)

**Questions:**

1.	At initialization, do all clients receive the entire set of parameters (e.g., encoders, decoders, and mixture-of-experts modules) from the server, even for modalities or tasks they do not engage with? If so, this may introduce communication and storage inefficiencies.

2. When and how frequently is the personalized model trained? How long is each local training interval?

3. The paper defines historical terms $h_{i,o}(\ell)$​ and $h_{i,o}(p)$; what are their dimensions, update rules, or role in guiding the adaptive mechanism? Clarification on whether they represent moving averages, scalar accumulators, or vector-valued metrics would aid interpretability.

4. Why is historical loss tracking required to compute the indicator? Wouldn’t the current-round loss $\ell_t$ suffice for determining adaptation triggers?

5. When the indicator activates, are all personalized parameters replaced with global ones? If so, does this erase accumulated personalization, potentially discarding useful local adaptations that could generalize later?

6. Empirically, is the learning rate static or decayed over time?

7. Are the reported results based on personalized accuracies (evaluated on the same clients used during training) or generalized accuracies (on unseen clients)? This distinction is critical for understanding TAP’s true generalization and personalization trade-offs.

---

> ### Author Response · Authors · 2025-11-19
>
> We thank the reviewer for the in-depth and thorough response, and taking valuable time to provide insightful feedback.
>
> ## 1. Weaknesses
> **(W1) Lack of motivating examples**
>
> We thank the reviewer for raising this concern. We note that heterogeneous multi-modal and multi-task personalized FL is an emerging area, with DisentAFL (AAAI 2024) [3] being, to the best of our knowledge, the only prior work that explicitly studies heterogeneous architectures under this setting. To clarify the practical relevance, we include the following motivating scenarios, which we have also added to the introduction of the manuscript under “Challenges”:
> 1.  Healthcare: Different hospitals may collect differing modalities (e.g., image scans, text reports, etc) and pursue distinct tasks such as diagnosis prediction or report generation.
> 2.  Industrial IoT: Factories may have differing sensor systems (e.g., acoustic vs. temperature measurement sensors) that optimize for differing priorities such as safety or defective product detection.
>
> **(W2) Insufficient distinction from related work**
>
> We appreciate the reviewer’s observation. In line with the reviewer’s suggestion, in the “PFL” portion of the related work, we have made additions where necessary to explicitly differentiate TAP between existing works to better emphasize the novelty of the proposed method. With regards to LoRA, we have addressed this concern in the response below. For the reviewer's convenience, we have also attached the changes to the PFL portion in the manuscript below:
> - "While Lu et al. (2024) explores heterogeneous model architectures in a multi-task setting, it is specifically designed for image inputs and therefore is uni-modal, unlike TAP, which deals with both heterogeneous tasks and modalities. To deal with this limitation, Chen & Zhang (2024) propose to have separate encoders and decoders for differing modalities and tasks, with a shared transformer backbone utilizing a Mixture of Experts (MoE) (Yuksel et al., 2012) structure to route inputs based off modality-task pairs. However, while a disentanglement axillary loss is introduced to potentially create more separable latent spaces between pairs, this approach can still result in entanglement of local models that share common subsets of the server model during aggregation. By contrast, TAP seeks to decouple itself as much as possible from the FL training process by only interacting with the server-engaged parameters when they signal that they can benefit the personalization process, avoiding the issue of losing the granularity of specific tasks during aggregation. Similar to Chen & Zhang (2024), Panchal et al. (2022) also explores using a routing mechanism for personalization, but the approach is uni-modal and uni-task without consideration for foundation models. Moreover, it necessitates the training of additional routing parameters, which TAP does not require for replacement operations."
>
> **(W3) Dependence on existing techniques**
>
> We appreciate the reviewer’s observation. We agree that we do not introduce a new variant of LoRA specific to the PFL context, but we would like to clarify that the contribution of TAP is not centered on modifying LoRA or any PEFT method itself. Instead, TAP addresses a different problem: how to perform effective personalization of foundation models heterogeneous in modalities and tasks, when the models engaged in the FL process seek to engage in fine-tuning of their models via a PEFT method. While we are, to the best of our knowledge, the first to apply fine-tuning techniques to pre-trained foundation models under the specified architecture in Fig. 1, it’s not the defining contribution of our work.
>
> Our novelty lies in TAP’s two-stage personalization process, whereby the personalized parameters engage in replacements for specific modality-task pairs, allowing for surgical adaption without negatively affecting other modality-task pairs that are not in need of replacement. With the second stage, the use of KD allows for the distillation of more generalizable representations that are not available to the personalized parameters without engaging in intrusive operations such as replacement, which would remove personalization when it wasn’t requested by the client. This procedure does not depend on LoRA-specific properties, as it involves the replacement of parameters and the addition of a KL-divergence term between two differing output logits. We chose LoRA due to its prevalence and efficiency in large-model fine-tuning. In addition, one of the baselines we evaluate in Tables 1,2, and 3 is FedDAT [2], which uses global and personalized adapters after FFN layers, not LoRA, to conduct personalization. Through this, we see that augmenting LoRA with our selective modality-task pair module replacement scheme outperforms PEFT methods designed specifically for personalized fine-tuning of foundation models in an FL setting.

---

> > ### Author Response · Authors · 2025-11-19
> >
> > Table 3: Performance comparison across FLAVA and ViLT on text datasets.
> >
> > | Model | Method | AG News (Acc ↑) | MMLU (BS ↑) | MMLU (METEOR ↑) | VQA (BS ↑) | VQA (METEOR ↑) | CommonGen (BS ↑) | CommonGen (METEOR ↑) | Avg. Gen. (BS ↑) | Avg. Gen. (METEOR ↑) |
> > |--------|--------------------------|------------------|--------------|--------------------|------------|-------------------|------------------------|---------------------------|--------------------|-------------------------|
> > | FLAVA  | Local | 90.26 ± 0.55  | $\underline{41.63}$ ± 1.26  | **20.89 ± 2.83**    | 47.72 ± 1.69 | 10.44 ± 2.37      | 30.62 ± 1.52            | 13.17 ± 2.68              | 39.66             | 13.62  |
> > |   | FedAvg  | 92.88 ± 0.22      | 37.83 ± 2.25  | 7.71 ± 1.76  | 37.68 ± 6.88 | 10.84 ± 4.33      | 29.70 ± 4.05            | 12.46 ± 6.72              | 34.52             | 10.86   |
> > |   | FedAvg + Post-train | $\underline{92.97}$ ± 0.26      | 36.88 ± 1.97  | 8.66 ± 1.24  | 41.62 ± 4.56 | 12.85 ± 2.87      | 31.48 ± 1.84            | 11.92 ± 1.68              | 36.62  | 11.64   |
> > |   | Per-FedAvg  | 92.82 ± 0.20      | 39.74 ± 2.14  | 9.17 ± 1.27  | 41.79 ± 10.81 | 18.06 ± 11.09     | 31.06 ± 3.68  | 10.53 ± 5.05              | 37.09             | 13.27                  |
> > |   | Per-FedAvg + Post-train  | **92.99** ± 0.28      | 39.40 ± 3.84  | 10.92 ± 4.06        | 46.70 ± 8.42 | 22.09 ± 8.38      | 31.83 ± 2.90            | 13.76 ± 5.21              | 39.29             | 16.53                  |
> > |  | DisentAFL | 92.76 ± 0.28      | 40.10 ± 0.77  | 19.54 ± 0.85        | 30.77 ± 0.64 | 5.46 ± 0.11       | 26.67 ± 0.48            | 7.60 ± 0.38               | 30.99             | 9.13                   |
> > |  | DisentAFL + Post-train   | 92.69 ± 0.30      | 40.37 ± 0.37  | $\underline{20.39}$ ± 1.64  | 34.99 ± 0.74 | 7.68 ± 0.77       | 27.86 ± 1.93            | 8.51 ± 0.33               | 33.21             | 10.56                  |
> > |   | FedDAT  | 92.54 ± 0.72      | 41.01 ± 2.63  | 6.83 ± 1.65         | 50.72 ± 17.55 | 28.30 ± 22.72  | $\underline{37.02}$ ± 5.32            | 13.74 ± 11.69             | 43.30             | 18.18                  |
> > |   | FedDAT + Post-train | 92.81 ± 0.36      | 40.29 ± 2.86  | 11.13 ± 6.60        | $\underline{57.96}$ ± 14.97 | $\underline{39.32}$ ± 24.72 | **40.25** ± 1.99            | **17.91** ± 7.97  | $\underline{47.34}$  | $\underline{25.12}$                  |
> > |   | **TAP (Ours)**  | 92.66 ± 0.34  | **45.81** ± 4.32 | 19.80 ± 3.83  | **69.38** ± 7.47 | **47.60 ± 14.92** | 34.01 ± 2.96      | $\underline{17.17}$ ± 4.56         | **50.52**         | **29.87**               |
> > |        |                          |                   |                |                    |             |                   |                        |                           |                   |                        |
> > | ViLT   | Local | 54.98 ± 15.88      | 55.03 ± 1.99  | 40.59 ± 2.65        | 55.68 ± 4.04 | 40.90 ± 11.61     | $\underline{40.79}$ ± 1.19            | **25.63** ± 1.53          | 49.60             | $\underline{34.73}$                  |
> > |    | FedAvg  | 70.61 ± 4.00       | 41.89 ± 1.45  | 6.79 ± 1.40         | 49.01 ± 11.17 | 18.25 ± 8.81      | 37.06 ± 4.53 | 14.03 ± 9.69              | 42.80             | 14.27                  |
> > |    | FedAvg + Post-train | **77.71** ± 2.35       | 42.87 ± 2.53  | 9.60 ± 4.64 | 60.42 ± 11.15 | 34.70 ± 14.45      | 39.22 ± 3.43            | 17.19 ± 9.56              | 48.43             | 22.68                  |
> > |    | Per-FedAvg  | 25.00 ± 0.05       | 44.61 ± 3.40  | 8.26 ± 1.66  | 49.30 ± 10.46 | 29.07 ± 17.48     | 31.98 ± 2.27  | 4.51 ± 2.97               | 41.43             | 15.08                  |
> > |    | Per-FedAvg + Post-train  | 25.00 ± 0.05       | 44.81 ± 4.60  | 22.82 ± 10.01 | **75.01** ± 7.22 | **65.48** ± 3.27 | 31.98 ± 2.27  | 4.51 ± 2.97          | $\underline{51.76}$ | 32.56  |
> > |    | DisentAFL  | 65.50 ± 9.35       | 54.83 ± 1.22  | 40.25 ± 1.95 | 42.01 ± 0.50 | 21.72 ± 0.85 | 33.37 ± 0.51   | 9.69 ± 0.13              | 41.12             | 20.62  |
> > |    | DisentAFL + Post-train   | 74.69 ± 4.02       | $\underline{55.31}$ ± 0.94  | $\underline{40.64}$ ± 1.42         | 48.94 ± 3.06 | 31.53 ± 10.13     | 34.94 ± 2.87            | 11.92 ± 1.33              | 44.61             | 25.51                  |
> > |    | FedDAT  | 26.62 ± 0.96       | 43.10 ± 1.88  | 8.92 ± 2.62         | 52.04 ± 16.53 | 29.67 ± 22.28      | 37.92 ± 3.64  | 15.93 ± 9.55              | 44.61             | 20.02   |
> > |   | FedDAT + Post-train  | 27.07 ± 0.97       | 43.72 ± 3.72  | 9.71 ± 7.71         | 56.53 ± 14.86 | 33.61 ± 22.48 | 39.77 ± 2.37            | 17.06 ± 8.88              | 47.27             | 22.21  |
> > |   | **TAP (Ours)**  | $\underline{77.44}$ ± 2.63   | **56.43** ± 3.12 | **40.79** ± 1.61  | $\underline{72.99}$ ± 7.47 | $\underline{62.52}$ ± 9.09 | **41.46** ± 1.45      | $\underline{25.35}$ ± 1.85         | **57.07** | **43.31**   |

---

> ### Author Response · Authors · 2025-11-19
>
> **(W4) Choice of baselines**
> We thank the reviewer for the insightful comment. The choice of baselines outlined in the manuscript is based in part from our closest peer, DisentAFL (AAAI 2024) [3], which we outperform in our experiments, which adopt similar baselines as there is a lack of existing methods that deal with heterogeneity of modalities and tasks across clients in a federated learning. However, in line with the reviewer’s suggestion, we have included Per-FedAvg, which is a well-established PFL algorithm. Moreover, we have additionally included a more recent work that deals with personalized fine-tuning of vision-language foundation models that are heterogeneous in tasks called FedDAT (AAAI 2024) [2] as a baseline. Both baselines have been added to Tables 1,2, and 3 in the manuscript. We have also posted the updated tables below for convenience. In addition, we have updated our experimental discussion in the manuscript with the new baselines.
>
> Table 1: Performance comparison for ViLT on image datasets. — indicates that the algorithm has failed to converge for that specific task.
>
> | Model | Method               | FMNIST (MSE ↓)        | Caltech-256 (MSE ↓)    | Avg. Gen. (MSE ↓) |
> |--------|------------------------|--------------------------|-----------------------------|-----------------------|
> | ViLT   | Local                  | 0.6267 ± 0.0379          | 0.6518 ± 0.1302             | 0.6368                |
> | ViLT   | FedAvg                 | 0.6128 ± 0.0822          | 0.5011 ± 0.1496             | 0.5681                |
> | ViLT   | FedAvg + Post-train    | $\underline{0.5502}$ ± 0.0454          | $\underline{0.4099}$ ± 0.1168             | $\underline{0.4941}$                |
> | ViLT   | Per-FedAvg             | —                        | —                           | —                     |
> | ViLT   | Per-FedAvg + Post-train| —                        | —                           | —                     |
> | ViLT   | DisentAFL              | 0.7925 ± 0.0444          | 0.5137 ± 0.0787             | 0.6810                |
> | ViLT   | DisentAFL + Post-train | 0.6333 ± 0.0423          | 0.4347 ± 0.0651             | 0.5539                |
> | ViLT   | FedDAT                 | 1.0326 ± 0.1585          | 1.5726 ± 0.1497             | 1.2576                |
> | ViLT   | FedDAT + Post-train    | 64.4282 ± 360.6894       | 1.5497 ± 0.1876             | 38.2288               |
> | ViLT   | **TAP (Ours)**         | **0.5467** ± 0.0409      | **0.3949** ± 0.1148         | **0.4860**            |
>
> Table 2: Performance comparison for FLAVA on image datasets.
>
> | Model | Method                 | Tiny-ImageNet (Acc ↑) | CIFAR-100 (Acc ↑) | FMNIST (MSE ↓)        | Caltech-256 (MSE ↓)     | Avg. Class. (Acc ↑) | Avg. Gen. (MSE ↓) |
> |--------|--------------------------|--------------------------|------------------------|---------------------------|----------------------------|-------------------------|------------------------|
> | FLAVA  | Local                    | 18.78 ± 12.50            | 23.26 ± 13.13          | 0.6209 ± 0.0230           | 0.6331 ± 0.0752            | 21.02                  | 0.6290                |
> | FLAVA  | FedAvg                   | 33.39 ± 5.26             | 46.93 ± 7.41           | 0.6292 ± 0.0304           | 0.4915 ± 0.0650            | 40.16                  | 0.5374                |
> | FLAVA  | FedAvg + Post-train      | 42.27 ± 6.79             | $\underline{53.67}$ ± 5.46       | 0.5651 ± 0.0064           | 0.4345 ± 0.0141            | $\underline{47.97}$             | 0.4780                |
> | FLAVA  | Per-FedAvg               | 27.75 ± 7.34             | 43.19 ± 8.42           | 0.7404 ± 0.0833           | 0.6390 ± 0.1000            | 35.47                  | 0.6728                |
> | FLAVA  | Per-FedAvg + Post-train  | 41.27 ± 7.06             | 53.36 ± 5.01           | 0.5796 ± 0.0105           | $\underline{0.4051}$ ± 0.0244            | 47.31                  | $\underline{0.4633}$               |
> | FLAVA  | DisentAFL                | 36.99 ± 11.12            | 0.85 ± 0.22            | 0.9036 ± 0.0179           | 0.5890 ± 0.0203            | 18.92                  | 0.6939                |
> | FLAVA  | DisentAFL + Post-train   | $\underline{45.53}$ ± 8.57             | 4.15 ± 0.52            | 0.7714 ± 0.0144           | 0.5496 ± 0.0253            | 24.84                  | 0.6235                |
> | FLAVA  | FedDAT                   | 16.54 ± 9.04             | 30.53 ± 12.74          | 0.5890 ± 0.0639           | 0.4164 ± 0.0733            | 23.53                  | 0.4787                |
> | FLAVA  | FedDAT + Post-train      | 26.15 ± 13.96            | 39.63 ± 13.41          | **0.5491** ± 0.0026       | **0.3718** ± 0.0176        | 32.89                  | **0.4309**            |
> | FLAVA  | **TAP (Ours)**           | **47.06** ± 6.80         | **56.26** ± 5.44       | $\underline{0.5572}$ ± 0.0083       | 0.4252 ± 0.0258        | **51.66**              | 0.4692            |

---

> ### Author Response · Authors · 2025-11-19
>
> **(W5) Unclear utility of knowledge distillation**
>
> We thank the reviewer for raising this point. While the reviewer is correct to note no major impact on the application of KD on image datasets, we see that in Table 6, the application of KD can have very pronounced differences in the realm of text. We believe this to be the case due to two factors. Firstly, as shown in prior work [4], language models tend to produce more sharply peaked predictive distributions compared to their vision counterparts. Secondly, in FL settings, text data also can exhibit substantially higher cross-client heterogeneity. For example, clients may differ in vocabulary, topic, writing style, etc. In contrast, image datasets such as CIFAR-100 and Tiny-ImageNet share a relatively consistent semantic space: the set of object categories and visual concepts remains more consistent across clients. This means for text data, the global model captures a broader and more diverse distribution over words than any single client, and therefore KD provides a more breadth of knowledge to the client in comparison to image-based tasks. In line with the reviewer’s comment, we have added these points to the experimental section discussing KD usage for better clarity and have attached them below for convenience:
> - "In regards to the difference in performance of image vs. text based tasks with KD, we believe this discrepancy to occur due to (i) the overconfidence of language models in comparison to vision models (Li et al., 2025) and (ii) the higher cross-client heterogeneity of text data. With text data, clients may differ in aspects such as vocabulary, topic, or writing style, meaning for text data, the server model captures a broader and more diverse distribution over words than any single client. This provides KD more breadth of knowledge to the client in comparison to image-based tasks."
>
> ## 2. Questions
> **(Q1) At initialization, do all clients receive the entire set of parameters?**
>
> We thank the reviewer for the question. At initialization, each client will only receive a subset of the server model’s parameters dependent on the tasks and modalities that each client is responsible for, thereby limiting issues of communication and storage inefficiencies. Moreover, we have added numbers in Appendix A.3 that demonstrates that the additional number of parameters introduced by TAP is small when compared to the overall model size.
>
> **(Q2) When and how frequently is the personalized model trained? How long is each local training interval?**
>
> We thank the reviewer for the question. We follow the standard practice of many existing PFL works, where personalized model training is triggered when the device is selected for aggregation. In our experiments, we set this local training interval to 20 minibatch iterations for FLAVA and 30 for ViLT, which can be noted at Appendix D.2 in Table 20.
>
> **(Q3) The paper defines historical terms $h_{i,o}(\ell)$ ​and $h_{i,o}(p)$ ; what are their dimensions, update rules, or role in guiding the adaptive mechanism?**
>
> We thank the reviewer for the insightful question. We note that the average local training losses for $\mathbf{\widetilde{W}}\_{[i]}$ and $\mathbf{X}\_{[i]}$ for round $t$ correspond to $h_{i,o}^{(l)}$ and $h_{i,o}^{(p)}$. Their roles, dimensions, and update rules were not explicitly defined, which may have caused confusion about what they represented. We thank the reviewer for raising this point. To account for this, we have specified that $h_{i,o}^{(l)}$ and $h_{i,o}^{(p)}$ are updated based on $\ell_{i,o}^{(l)}$ and $\ell_{i,o}^{(p)}$ in Sec. 3.2 via the following:
> - "These values are used to update client $c_i$'s history values, defined as $h_{i, o}^{(l)}$ and $h_{i, o}^{(p)}$ via $h_{i, o}^{(l)} = \ell_{i, o}^{(l)}$ and $h_{i, o}^{(p)} = \ell_{i, o}^{(p)}$."
>
> Additionally, we would like to point the reviewer to Appendix A.1 for the pseudocode of the algorithm, which lies out in greater specificity on how $h_{i,o}^{(l)}$ and $h_{i,o}^{(p)}$ interact with the algorithm. Lastly, we also address greater specifics on the history values in response to the question below.

---

> ### Author Response · Authors · 2025-11-19
>
> **(Q4) Why is historical loss tracking required to compute the indicator? Wouldn’t the current-round loss $\ell_t$ suffice for determining adaptation triggers?**
>
> We thank the reviewer for raising this concern. As noted in our earlier response, we do use the round losses to determine when replacements should occur for task $o$. However, relying solely on the current round loss is insufficient for making correct adaptation decisions. Because TAP's workflow follows (1) training $\mathbf{\widetilde{W}}\_{[i]}$, (2) conducting replacements on $\mathbf{X}\_{[i]}$, and then (3) training $\mathbf{X}\_{[i]}$ (as seen in Appendix A.1), we need to know how $\mathbf{X}\_{[i]}$ performed the last time it was updated, which took place in a previous round before $t$. Without keeping track of this, we would have nothing to compare $\mathbf{\widetilde{W}}\_{[i]}$’s newly updated performance with the performance of $\mathbf{X}\_{[i]}$. Therefore, we track the loss from the previous round in which client $c_i$ was trained via the history values so that we can determine whether $\mathbf{X}\_{[i]}$ is still outperforming $\mathbf{\widetilde{W}}\_{[i]}$ and thus warrants replacement.
>
> **(Q5) When the indicator activates, are all personalized parameters replaced with global ones?**
>
> We thank the reviewer for raising this point. When looking at Eq. 4, we emphasize that replacement takes place for $\mathbf{X}\_{[i, o]}$, which is a subset of $\mathbf{X}\_{[i]}$, which means that TAP will conduct replacement only for tasks that receive the signal that they wish to use the server-engaged parameters. In this way, we prevent the discarding of useful local adaptations for tasks that do not need it.
>
> **(Q6) Empirically, is the learning rate static or decayed over time?**
>
> We thank the reviewer for the question. Empirically, we adopt linear warmup of the learning rate in the early stages of training, with it holding static for the remainder of training after the desired number of warmup rounds have been completed. Specifics on the set learning rates and the number of warmup rounds can be found in Appendix D.2 in Table 20.
>
> **(Q7) Are the reported results based on personalized accuracies (evaluated on the same clients used during training) or generalized accuracies (on unseen clients)?**
>
> We appreciate the reviewer’s concern and raising this point. The reported results use personalized accuracy, meaning we compute each client’s evaluation metrics on its corresponding test dataset, consistent with established practice in prior work [1-3]. Since TAP aims to provide per-client personalization aligned with each client’s specific tasks, personalized evaluation is a natural choice. We also emphasize that generalization to unseen clients constitutes a separate research direction from PFL.
>
> **References:**
>
> [1] Personalized federated learning with theoretical guarantees: A model-agnostic meta-learning approach (Fallah et al., NeurIPS 2020).
>
> [2] FedDAT: An approach for foundation model finetuning in multi-modal heterogeneous federated learning (Chen et al., AAAI 2024).
>
> [3] On disentanglement of asymmetrical knowledge transfer for modality-task agnostic federated learning (Chen & Zhang, AAAI 2024).
>
> [4] Bild: Bi-directional logits difference loss for large language model distillation (Li et al., International Conference on Computational Linguistics 2025).

---

> > ### Comment · Reviewer_GYCq · 2025-11-21
> >
> > Thank you to the authors for their detailed response.
> > ___
> > 1. The core techniques of TAP, the parameter-efficient fine-tuning (LoRA or adapters) and Knowledge Distillation, are well-established in the literature. Framing existing methods within a “modality-task pair replacement” mechanism represents more of an application or adaptation of known techniques rather than a fundamentally new algorithmic contribution.
> > ___
> > 2. In Tables 1 to 3, while TAP shows higher mean values in some metrics, a closer inspection of the reported standard deviations shows substantial overlap with the results of simpler baselines such as FedAvg + Post-train or Per-FedAvg + Post-train. For example, the Tiny-ImageNet accuracy on FLAVA (Table 2) of TAP is 47.06 $\pm$ 6.80, which overlaps with the upper range of Per-FedAvg + Post-train (41.27$\pm$7.06). Similar overlaps exist for the other models, datasets, and metrics too. This suggests that the apparent gains may not be statistically significant and could be within the variability of simpler methods. As such, the empirical improvement of TAP over established baselines is limited and does not clearly demonstrate a substantial advantage.
> >
> > Moreover, TAP introduces a considerably more complex procedure with two-stage personalization, selective parameter replacement, and knowledge distillation. Given the marginal or uncertain benefits, this added complexity unnecessarily increases computational cost and training overhead, offering little practical advantage over much simpler and more efficient baselines.
> > ___
> > 3. While the authors provide a rationale for why knowledge distillation (KD) may have more impact on text than image datasets, the argument relies largely on qualitative intuition rather than quantitative evidence. For instance, the claim that image datasets “share a relatively consistent semantic space” does not account for the high variability that can exist even within object categories, lighting conditions, or backgrounds; factors that can still produce heterogeneous client distributions.
> >
> > Similarly, the assertion that KD provides more “breadth of knowledge” for text assumes that the global model’s output distribution is meaningfully richer than any individual client’s, yet no direct analysis is provided to quantify how much information is actually transferred via KD in these FL settings.
> >
> > Additionally, even in the FL setting, aren’t the “clients” in these experiments essentially simulated by splitting the same dataset? If so, can we really conclude that KD is addressing true cross-client heterogeneity, or are the observed effects simply a result of intra-dataset variation rather than realistic differences in vocabulary, style, or topics across genuinely distinct clients?
> >
> > Therefore, the explanation remains largely speculative and would benefit from more concrete, controlled comparisons.
> > ___
> > 4. I understand that personalized accuracy is appropriate for evaluating per-client performance in PFL, it would still be informative to report generalized accuracy on unseen clients. In federated learning, both personalized and generalized performance are important: personalized accuracy reflects how well the model adapts to each client’s specific data, while generalized accuracy indicates how effectively knowledge is shared across clients and how well the model performs on new, unseen clients. This is particularly relevant given that the Contributions section (1.1) states that “while not tailored for personalization, its (Knowledge Distillation's) learned representations can still provide generalizable knowledge presented from other modality-task pairings without engaging in an aggressive operation such as replacement.” Reporting generalized accuracy would therefore offer a more complete picture of TAP’s effectiveness and help contextualize the trade-off between personalization and generalization.

---

> > > ### Author Response · Authors · 2025-11-24
> > >
> > > **2.  Statistical significance of TAP and computation/training overhead.**
> > > - ***Regarding the reviewer's question on the statistical significance of the improvements of TAP in comparison to the baselines, we have included new metrics in Appendix C.5 of the manuscript, where we apply a Welch t-test [4] to determine the statistical significance of TAP's performance improvement in comparison to the baselines with a p-value less than 0.05. We attach the new results at the end of this response for the reviewer's convenience.*** We note that across nearly all baselines and settings, TAP achieves statistically significant improvements on the majority of metrics. Even in cases where significance is comparatively lower (e.g., FedDAT vs. ViLT on FLAVA), TAP remains consistently stronger across architectures (e.g., FedDAT vs. TAP on ViLT). These findings give credence that TAP's gains are both numerically and statistically reliable.
> > >
> > > 	Moreover, we would like to emphasize the consistency of TAP across all tasks for both ViLT and FLAVA in Tables 1-3, in contrast with the baselines. For example, with Per-FedAvg [2] + Post-train on ViLT, while it might demonstrate high performance on VQA, it struggles to converge to a meaningful result on both FMNIST and Caltech-256, and lags significantly behind TAP on tasks such as MMLU (e.g., 44.81 vs. 56.43 BS scores) and CommonGen (31.98 vs. 41.46 BS scores). Another example would be FedDAT [3] + Post-train on FLAVA, where its gains in image generation tasks come at the cost of approximately 20% accuracy difference compared to TAP on Tiny-ImageNet and CIFAR-100.
> > >
> > > 	We would like to additionally note that although the first stage of TAP (selective modality-task pair module replacement scheme) involves the training of the regular and then personalized parameters, this is consistent with existing PFL literature and therefore does not exhibit significant computation training overhead in comparison to existing works. For example, for Per-FedAvg, it must conduct two forward passes and collect two gradients for an inner and outer step update [2]. Likewise, with FedDAT, it must update both its regular adapters and personalized DAT adapters during each training iteration [3]. We would also like to refer the reviewer to Table 7 in Appendix A.3 of the manuscript, where we show the average per-client training time for KD is approximately 200 seconds, indicating that training for long durations after FL is not required. This is especially true since based from our observations, the FL training phase takes on average 50-60 minutes of wall clock time on a per-client basis.
> > >
> > > Table: Percentage of metrics for which TAP exhibits both superior performance and statistical significance with a significance threshold of 0.05 in comparison to each baseline.
> > >
> > > |**Model**|**Baseline**|**% Metrics TAP Significantly Better**|
> > > |-----------|------------------------------|-----------------------------------------|
> > > | **FLAVA** | Local| 90.9%|
> > > || FedAvg| 90.9%|
> > > || FedAvg + Post-train| 63.6%|
> > > || Per-FedAvg| 90.9%|
> > > || Per-FedAvg + Post-train| 63.6%|
> > > || DisentAFL| 81.8%|
> > > || DisentAFL + Post-train| 72.7%|
> > > || FedDAT| 54.5%|
> > > || FedDAT + Post-train| 45.5%|
> > > | **ViLT**  | Local| 55.6%|
> > > || FedAvg| 100.0%|
> > > || FedAvg + Post-train| 66.7%|
> > > || Per-FedAvg| 100.0%|
> > > || Per-FedAvg + Post-train| 77.8%|
> > > || DisentAFL| 77.8%|
> > > || DisentAFL + Post-train| 66.7%|
> > > || FedDAT| 100.0%|
> > > || FedDAT + Post-train| 88.9% |
> > >
> > > **3.  Explanation on KD usage mostly speculative.**
> > > - While we acknowledge that there indeed could be cases by which image settings are more heterogeneous, we would like to note that this is not the case in our experiments, as there are more semantic similarities between Tiny-ImageNet and CIFAR-100, for example, than there is between CommonGen and MMLU.
> > >
> > > 	On the reviewer's point whether the global model's output distribution is meaningfully richer than any individual client, existing work [5] reveals that adding KD as an auxiliary term (similarly to our work) works as an anchor for the student model, injecting the aggregated distribution learned from all clients into each individual client. While these methods have been developed mainly for decreasing client drift from the global model, it demonstrates that such methods are an effective means of distilling meaningful information from the global teacher to the local student.
> > >
> > > 	Lastly, in regards to whether we can really conclude cross-client heterogeneity from our numerical experiments, we would like to note that our experiments deal with 8 datasets spread across clients for FLAVA and 6 for ViLT, with not all clients sharing the same datasets. 3 of these datasets, we note, deal with text generation which vary in their objectives. For example, MMLU might seek to answer an exam-style question on professional law, while CommonGen will seek to create a common-sense sentence based off a given list of words, meaning clients do exhibit realistic differences in used vocabulary, style, etc.

---

> ### Author Response · Authors · 2025-11-24
>
> Thank you for your response. Please find our further rebuttal below:
>
> **1. Core techniques of TAP, the parameter-efficient fine-tuning (LoRA or adapters) and Knowledge Distillation. Framing existing methods within a “modality-task pair replacement” mechanism represents more of an application or adaptation of known techniques.**
> - **We would like to re-emphasize that the defining contribution of our work does not center on PEFT methods, *but on a method by which we can personalize on a surgical level for individual modality-task pairs, when the FL process is conducting fine-tuning of an existing pre-trained foundation model.*** As iterated in the introduction of the manuscript, the problem is a challenging one due to a number of key challenges: **(1)** heterogeneity of modalities and tasks across clients, **(2)** heterogeneity of individual model architectures between clients, and **(3)** the need to target modality-task pairs on an individual level without influencing others. As there is no straightforward mechanism for personalizing on an individual modality-task-wise basis, ***we would like to emphasize that we are the first to apply selective modality-task pair module replacement in a federated multi-modal and multi-task setting.*** Due to the heterogeneity of model architectures, selective modality-task pair module replacement of TAP is particularly well-suited for the problem setup, as it allows for replacement of specific modality-task pairs without negatively impacting those pairs that do not require replacement. ***Existing works do not contend with these additional dimensions and therefore are not suited for knowing when certain modality-task pairs should be prioritized over others.***
>
> 	For knowledge distillation (KD), while its usage is well-known throughout FL literature, our use of it is ***motivated by challenges that are unique to our setting and has not been previously explored in the context of task and modality heterogeneous FL.*** In our framework, TAP is deliberately designed to minimize interaction with server-engaged parameters in the FL training phase, which allows each client to maintain a high degree of personalization. However, this decoupling means that personalized parameters may not fully benefit from the broad, cross-client representations that occur during aggregation at the server. These representations produced from the server-engaged parameters could encompass generalizable patterns derived from diverse client data, which a purely personalized model would otherwise not have access to. Unlike more aggressive operations, like the replacement that takes place in the FL stage, post-FL KD enables transfer of beneficial knowledge while preserving each client’s personalized nature. Existing work shows this to be the case, as [1] theoretically demonstrates that when the number of data samples are few (as is the case for a single client compared to the amount of data aggregation exposes the server model to), the student is pulled towards the teacher but will in general not coincide completely with the teacher’s logits.

---

> ### Author Response · Authors · 2025-11-24
>
> **4.  It would still be informative to report generalized accuracy on unseen clients.**
> - From our understanding, we believe that the reviewer implies to leave a subset of clients separate from FL training, which will receive the final global model upon FL completion and will evaluate on those clients that have not participated in the FL protocol. ***We would like to emphasize that this does not apply to TAP or PFL generally due to the fact that personalization is triggered when a client is selected for aggregation [2,3,6,7], and therefore holding out a subset of unseen clients results in no personalization on said clients. Moreover, it is common convention in PFL that the end-goal is achieving low local loss on each client-specific dataset (as we state for TAP near the conclusion of Sec. 3.1.) [2,6,7]. Because of these factors, the reporting of only per-client average performance on their individual test sets is the common setup [2,3,6,7].*** In addition, due to the fact that TAP separates the personalized and FL-engaged parameters from one another during training, ***a client can merely utilize the aforementioned FL-engaged parameters for a more general output. Moreover, due to this decoupling that TAP conducts between the two groups, the FL-engaged parameters will always have the same performance as FedAvg.***
>
> 	We would also like to note that the server architecture adopted from [6] requires the server to know modality-task pairings to route specific inputs. If some clients are left out in this process, the server will have no means of taking input from modality-task pairings it's not trained for. We state
> 	- “While not tailored for personalization, its (Knowledge Distillation’s) learned representations can still provide generalizable knowledge presented from other modality-task pairings without engaging in an aggressive operation such as replacement.”
>
> 	in the contribution of the manuscript since there could be instances where there is some overlap between differing pairs in the parameter space (e.g., image generation and image classification with image modality).
>
> Please let us know if you have any additional comments. We hope you can reconsider your score in light of our clarifications.
>
> **References**
>
> [1] Towards understanding knowledge distillation (Phuong & Lampert, ICML 2019).
>
> [2] Personalized federated learning with theoretical guarantees: A model-agnostic meta-learning approach (Fallah et al., NeurIPS 2020).
>
> [3] FedDAT: An approach for foundation model finetuning in multi-modal heterogeneous federated learning (Chen et al., AAAI 2024).
>
> [4] Best practice in statistics: Use the Welch t-test when testing the difference between two groups (West, Robert M., Annals of Clinical Biochemistry 2021)
>
> [5] Knowledge Distillation in Federated Learning: A Practical Guide (Mora et al., IJCAI 2024).
>
> [6] On Disentanglement of Asymmetrical Knowledge Transfer for Modality-Task Agnostic Federated Learning (Chen & Zhang, AAAI 2024).
>
> [7] Mixture of Experts Made Personalized: Federated Prompt Learning for Vision-Language Models (Luo et al., ICLR 2025).

---

### Official Review · Reviewer_AVRh · 2025-11-01

**Soundness:** 3
**Presentation:** 3
**Contribution:** 3
**Rating:** 4
**Confidence:** 4

**Summary:**

The paper proposes TAP for FL with heterogeneous multi-modal, multi-task foundation models. In Stage 1, each client maintains a local model alongside the global model. In Stage 2, the method performs knowledge distillation from the global model into the local one using a KL.

**Strengths:**

1. Addresses a challenging and underexplored problem in PFL where clients differ in tasks and modalities.
2. A proof of convergence has been provided. The convergence bound for component-wise FedAvg over blocks supports the claim.

**Weaknesses:**

1. The convergence bound grows with the number of modality–task pairs $K^2$.
2. Experiments involve only a small number of clients (≈10) offering little evidence of scalability.

**Questions:**

1. Please emphasize the differences between the method with previous ones. My understanding is that the most significant innovation lies in the personalized approach in stage 2?
2. How does TAP perform when the number of clients increases, like to 100 clients?
3. How can the issues of modality–task pairs and scalability be addressed?
4. Please provide details of the computation overhead in Stage 2, such as training time and resource consumption.

---

> ### Author Response · Authors · 2025-11-19
>
> We thank the reviewer for the time and effort to provide valuable feedback.
>
> ## 1. Weaknesses
> **(W1) Convergence bound grows with the number of modality–task pairs  $K^2$.**
>
> We thank the reviewer for the comment. We would like to note that the convergence bound is not for the personalized model, but instead for the server model. The second and third RHS terms of the bound growing in $\zeta_r^2$ demonstrates that with the increase in modality-task pairs, the ability of the server model to cater to each individual client decreases. Therefore, the bound demonstrates the need for more refined personalized techniques that can decouple from this bound, which TAP seeks to achieve by separating itself from the regular FL process as much as possible, only engaging with the server-based parameters when offering benefit for specific modality-task pairs.
>
> **(W2) Experiments involve only a small number of clients (≈10) offering little evidence of scalability.**
>
> We thank the reviewer for the comment. We would like to direct the reviewer to the “Datasets” portion of Sec. 5.1 in the manuscript, where we have stated that we consider 30 clients, not 10 in our experiments. For convenience, we have attached the sentence that is being referred to from the manuscript below:
> - "We adopt the usage of eight common datasets for FLAVA and six for ViLT, spread across 30 clients."

---

> ### Author Response · Authors · 2025-11-19
>
> ## 2. Questions
> **(Q1) Emphasizing TAP's difference over previous works**
>
> We thank the reviewer for the question. Our most significant innovation lies in Stage 1, with replacement operations during the FL training phase. Due to the heterogeneity of modalities and tasks between clients, a challenge is knowing how to maximize personalization to each client when the server is juggling numerous modalities and tasks. Our approach allows for selective modality-task pair module replacement of the personal model’s parameters with the server-engaged ones when benefit is indicated for task $o$, as outlined in Eq. 3. A novelty of this strategy is that TAP is surgical in how it carries out replacements–it is able to conduct replacements on an individual modality-task pair level, so that it doesn’t degrade performance on tasks that do not need replacement. Stage 2 is the post-FL KD phase, whereby the server-engaged model serves as a teacher model to the personalized model, allowing it to learn beneficial and generalizable representations derived from diverse client data that it does not have access to without engaging in aggressive operations such as replacement, which is what is utilized in Stage 1. We would also like to note our work is the first, to the best of our knowledge, that deals specifically with the fine-tuning personalization of existing pre-trained foundation models in a setting where modalities and tasks are heterogeneous across clients.
>
> **(Q2) TAP's performance with more clients**
>
> We thank the reviewer for the question. As mentioned earlier when addressing the reviewer’s question on **(W2)**, we clarify that we utilize 30 clients, not 10 in our numerical results.
>
> **(Q3) How can the issues of modality–task pairs and scalability be addressed?**
>
> We would like to direct the reviewer to our response to **(W1)**, whereby our method seeks to decouple itself as much as possible from the vanilla FL training protocol, thereby limiting interference with other modality-task pairs.
>
> **(Q4) Details of the computation overhead**
>
> We thank the reviewer for raising this point. We have reported the average per-client number of parameters, trainable percentage of parameters, and memory footprint of TAP in comparison to not utilizing the TAP algorithm in Appendix A.3. Additionally, we have also reported the average per-client wall clock training time of Stage 2 in Appendix A.3. We have attached those numbers and observations written in the updated manuscript below for the reviewer's convenience:
>
> Table 7: Memory footprint of utilizing TAP vs. no usage of TAP and the wall clock time in seconds of the KD post-FL phase on average for each client.
>
> | Model | Setting | Avg. Total Params. | Avg. % Trainable | Avg. Memory | Avg. KD Training Time (sec.) |
> |-------|---------|----------------------|-------------------|-------------|-------------------------------|
> | **FLAVA** | No TAP | 310,559,996.40 | 10.3347% | 1.1569 GB | — |
> | **FLAVA** | TAP | 347,422,994.40 | 18.1086% | 1.2943 GB | 202.65 |
> | **ViLT** | No TAP | 585,627,611.80 | 9.9788% | 2.1816 GB | — |
> | **ViLT** | TAP | 646,296,580.40 | 18.0053% | 2.4076 GB | 190.58 |
> - "Due to the necessity of TAP requiring a post-FL knowledge distillation (Stage 2 of Fig. 2) phase and additional personalized parameters, we seek to characterize how these aspects affects the memory footprint and total extra training time per-client, which can be found in Table 7.
>
> 	We firstly note that although TAP adds additional parameters, this expansion remains small relative to the full model size. For example, with FLAVA, the total parameter count increases from 310.6M to 347.4M, with ViLT being 585.6M to 646.3M. For TAP, this accounts for about 18% of total parameters, showing that TAP still remains faithful to the idea that fine-tuning foundation models should only involve a small number of the total parameters. Moreover, the memory overhead introduced by TAP is minimal. FLAVA’s memory usage increases from 1.16GB to 1.29GB, and ViLT’s from 2.18GB to 2.41GB, amounting to a roughly 10% increase. This indicates that the additional personalized parameters do not substantially affect the memory footprint.
>
> 	In terms of average per-client knowledge distillation (KD) time, we note that the post-FL KD step adds only a modest amount of computation. The average per-client KD time is approximately 200 seconds, meaning it does not require each client to train for long durations after FL has completed."

---

> ### Author Response · Authors · 2025-11-27
>
> We once again thank the reviewer for the careful evaluation of our paper. We would appreciate any additional comments in light of our latest responses.

---

### Official Review · Reviewer_KtKr · 2025-11-01

**Soundness:** 2
**Presentation:** 3
**Contribution:** 2
**Rating:** 4
**Confidence:** 2

**Summary:**

This work proposes the TAP strategy, which integrates an adaptive parameter replacement mechanism during federated training with a knowledge distillation phase after training. This design enables clients to perform efficient personalized modeling in multimodal and multi-task heterogeneous environments. Furthermore, the convergence of the multimodal–multi-task server model is theoretically analyzed, and the method demonstrates promising results across multiple benchmarks.

**Strengths:**

1. The writing is clear and easy to follow;

2. The paper provides a well-articulated theoretical framework and convergence analysis;

3. It achieves state-of-the-art performance on eight common datasets for FLAVA and six for ViLT, distributed across 30 clients;

**Weaknesses:**

In fact, I believe this work mainly combines existing ideas and presents an incremental improvement rather than a fundamentally novel contribution.

Specifically, in multi-task learning, there already exist numerous studies that employ Mixture-of-Experts (MoE) for unified modeling (e.g., [1,2,3]). Similarly, for multimodal unification, related approaches have been explored (e.g., [4]), and even some works [5] have attempted to jointly unify both aspects. The contribution of this paper primarily lies in extending these ideas to the federated learning setting, which, in essence, appears relatively trivial.

Moreover, in the federated learning community, the use of knowledge distillation (KD) is far from new. Incorporating KD here further adds to the overall complexity of TAP without introducing substantial conceptual novelty. In other words, this work can be viewed largely as a combination of existing components and engineering-oriented refinements rather than a breakthrough innovation.

[1]. Modeling task relationships in multi-task learning with multi-gate mixture-of-experts, KDD 2018

[2]. Mod-squad: Designing mixtures of experts as modular multi-task learners, CVPR 2023

[3]. M³vit: Mixture-of-experts vision transformer for efficient multi-task learning with model-accelerator co-design, NeurIPS 2022

[4]. Multi-modal gated mixture of local-to-global experts for dynamic image fusion, ICCV 2023

[5]. Dynamic modeling of patients, modalities and tasks via multi-modal multi-task mixture of experts, ICLR 2025

**Questions:**

Please see the Weaknesses.

---

> ### Author Response · Authors · 2025-11-19
>
> We thank the reviewer for the time and effort to provide in-depth feedback and insight.
>
> ## 1. Weaknesses
> **(W) Incremental improvement rather than a fundamentally novel contribution**
>
> We thank the reviewer for the thorough response and the issues raised. Firstly, with regards to [1-5], we would like to note the focus of TAP is not the use of Mixture of Experts (MoE) in its training process. The MoE process is an artifact from [DisentAFL] (AAAI 2024), whose model architecture we adopt for allowing training of models varying in both modalities and tasks, which [1-4] does not consider (e.g., [4] deals with multi-modal models but is uni-task, and [2, 3] is multi-task but uni-modal). The goal of TAP is to engage in personalization of clients heterogeneous in modalities and tasks by (1) selective modality-task pair module replacement and (2) post-FL KD. For (1), we emphasize that the replacement operation conducts replacement on parameters that are responsible for certain modality-task pairs via floating-point margin hyperparameters, and therefore does not inherently rely on an MoE being present within the system. For (2), the KD process relies on measuring the KL divergence between the two differing logit outputs, and therefore likewise does not rely on there being an MoE present. Moreover, we would like to emphasize that as [1-5] focuses on the centralized setting, there is no straightforward mechanism to personalize specific modality-task pairs that vary across clients when naively extended to a federated environment. By contrast, TAP is able to engage in selective modality-task pair module replacement operations for individual modality-task pairs, allowing for more surgical levels of personalization without negatively affecting pairs that do not need a signal from the server-engaged parameters. To better emphasize these points, we have added a section to the related work with regards to MoEs and specifying their usage in relation to TAP. For the reviewer's convenience, we attach the aforementioned related work section below:
> - "Mixture of Experts (MoE): Mixture of Experts (MoE) are a class of architectures that seek to specialize models by introducing “expert” sub-networks with routing mechanisms to activate certain experts over others depending on the input. A common example is the top-$k$ selection (Zhou et al., 2022), whereby the top-$k$ experts selected by the router are utilized. Existing work has shown that MoEs can be utilized for multi-task objectives (Ma et al., 2018). Moreover, MoEs have been developed for multi-task (Chen et al., 2023; Fan et al., 2022) and multi-modal (Cao et al., 2023) foundation models, with some considering both dimensions (Wu et al., 2025). However, these works focus on a centralized setting and offer no straightforward mechanism to personalize specific modality-task pairs that vary across clients when naively extended to a federated environment. While our method adopts the MoE components from Chen & Zhang (2024), this choice is architectural rather than conceptual as TAP itself does not target MoE-specific routing or expert selection strategies."
>
> In terms of the addition of KD into the post-FL training process, although post-FL KD has been studied in prior work, our use of KD is motivated by challenges that are unique to our setting and has not been previously explored in the context of task and modality heterogeneous FL. In our framework, TAP is deliberately designed to minimize interaction with server-engaged parameters in the FL training phase, which allows each client to maintain a high degree of personalization. However, this decoupling means that personalized parameters may not fully benefit from the broad, cross-client representations that occur during aggregation at the server. These representations produced from the server-engaged parameters could encompass generalizable patterns derived from diverse client data, which a purely personalized model would otherwise not have access to. Unlike more aggressive operations, like the replacement that takes place in the FL stage, post-FL KD enables transfer of beneficial knowledge while preserving each client’s personalized nature. We also emphasize that, to our knowledge, we are the first to demonstrate that this strategy remains effective even under heterogeneity of modality and task scenarios.
>
> **References:**
>
> [DisentAFL] On disentanglement of asymmetrical knowledge transfer for modality-task agnostic federated learning (Chen & Zhang, AAAI 2024).

---

> > ### Comment · Reviewer_KtKr · 2025-11-26
> >
> > Thank you for the authors’ clarification. In fact, I still tend to view this work as incremental, although it may represent a novel attempt within the federated learning community. From a methodological perspective, I find it difficult to justify a more positive score. In addition, as I am not an expert in FL, I keep my original rating.

---

> > > ### Author Response · Authors · 2025-11-26
> > >
> > > Thank you to the reviewer for the response.
> > >
> > > In regards to the concern about the novelty of our work, we would like to emphasize with the exception of DisentAFL (whose method we exceed in numerical experiments), we are, to the best of our knowledge, ***the only work that deals with heterogeneous modality and task based personalized FL.*** The problem is a challenging one in comparison to personalization in traditional FL due to a number of key challenges: **(1)** heterogeneity of modalities and tasks across clients, **(2)** heterogeneity of individual model architectures between clients, and **(3)** the need to target modality-task pairs on an individual level without influencing others. ***We would like to additionally emphasize that we are the first to apply selective modality-task pair module replacement in a federated multi-modal and multi-task setting.*** Due to the heterogeneity of model architectures, selective modality-task pair module replacement of TAP is particularly well-suited for the problem setup, as it allows for replacement of specific modality-task pairs without negatively impacting those pairs that do not require replacement.  ***Existing works do not contend with these additional dimensions and therefore are not suited for knowing when certain modality-task pairs should be prioritized over others.***
> > >
> > > Please let us know if you have any additional comments. We hope you can reconsider your score in light of our clarifications.

---

### Official Review · Reviewer_Luv2 · 2025-11-01

**Soundness:** 3
**Presentation:** 2
**Contribution:** 3
**Rating:** 6
**Confidence:** 3

**Summary:**

This paper proposes a new personalized federated learning method, called TAP, which enables the personalization of heterogeneous multi-modal and multi-task foundation models. Specifically, it designs a replacement operation to selectively update parameters across mismatched architectures, providing significant benefits to the personalization process. Moreover, a knowledge distillation (KD)-based post-training procedure is incorporated at the client side. In addition, the paper provides a convergence analysis of the server model under its modality–task pair architecture. Experimental results demonstrate the effectiveness of the proposed method.

**Strengths:**

1. The proposed PFL algorithm supports heterogeneous multi-modal and multi-task personalization, and is the first to offer a convergence analysis specifically designed for the modality–task pair architecture.
2. In the federated training phase, each client adaptively determines whether to update portions of its local parameters with the global ones from the server according to its task performance, enabling selective absorption of knowledge that benefits its specific task. This is a particularly interesting and thoughtful design.
3. The authors conduct extensive experiments across multiple modalities, including image generation, text generation, image classification, and text classification, to comprehensively evaluate the proposed framework.

**Weaknesses:**

1. The font size used in Figures 1 and 2 is too small, which affects the overall readability and clarity of the illustrations.
2. Each task requires a manually specified margin hyperparameter, making the approach sensitive to hyperparameter selection.
3. The motivation behind employing knowledge distillation in the post-training stage is not well-grounded. Apart from empirical performance improvements, the framework lacks a clear theoretical justification or necessity for incorporating this additional step.

**Questions:**

1. As $\mathbf{X}_{[i]}$ is updated only on the tasks encountered in each local training round, how does the method address potential catastrophic forgetting when a client is associated with multiple tasks?
2. After the completion of FL training, does the additional knowledge distillation stage risk compromising the personalized adaptations of client models by imposing generalized representations from the global teacher? What is the technical motivation for introducing this step, considering the lack of a clear theoretical justification in the paper?
3. In Table 5, the improvement brought by knowledge distillation on image-based tasks appears marginal compared with that on text datasets. What factors contribute to this discrepancy? A more thorough analysis would help clarify this point.
4. In the experimental section, does each client correspond to a single task or multiple tasks?

---

> ### Author Response · Authors · 2025-11-19
>
> We thank the reviewer for taking the time to provide thorough and in-depth feedback.
>
> ## 1. Weaknesses
> **(W1) Font size used in Figures 1 and 2 is too small.**
>
> We thank the reviewer for raising the following point. In line with the suggestion, we have increased the font sizes of both Figures 1 and 2 to improve readability.
>
> **(W2) Sensitive to margin hyperparameter selection.**
>
> We thank the reviewer for raising the following concern. In regards to the sensitivity of the margin hyperparameters, we would like to point the reviewer to an ablation study on the margins in Appendix C.2. There we demonstrate that differing margin hyperparameters do not result in significant performance differences for a majority of tasks, meaning that while the margins might need tuning for some tasks, it is not a highly sensitive hyperparameter.
>
> **(W3) Motivation behind employing knowledge distillation not well-grounded.**
>
> We thank the reviewer for the insightful comment. Our use of KD is motivated by challenges that are unique to our setting and has not been previously explored in the context of task and modality heterogeneous FL. In our framework, TAP is deliberately designed to minimize interaction with server-engaged parameters in the FL training phase, which allows each client to maintain a high degree of personalization. However, this decoupling means that personalized parameters may not fully benefit from the broad, cross-client representations that occur during aggregation at the server. These representations produced from the server-engaged parameters could encompass generalizable patterns derived from diverse client data, which a purely personalized model would otherwise not have access to. Unlike more aggressive operations, like the replacement that takes place in the FL stage, post-FL KD enables transfer of beneficial knowledge while preserving each client’s personalized nature. Existing work shows this to be the case, as [1] theoretically demonstrates that when the number of data samples are few (as is the case for a single client compared to the amount of data aggregation exposes the server model to), the student is pulled towards the teacher but will in general not coincide completely with the teacher’s logits. To emphasize this, we offer a more concise justification above Eq. 5 in Sec. 3.3 summarizing these points. For the reviewer's convenience, we also attach this newly-added justification from the manuscript below:
> - "The FL-trained base may help learn generalizable representations present across differing tasks, which a purely personalized model would otherwise not have access to. Unlike more aggressive operations such as replacement, post-FL KD enables transfer of beneficial knowledge while preserving each client’s personalized nature. Existing work (Phuong & Lampert, 2019) theoretically demonstrates this to be the case, whereby the student is pulled towards the teacher but in general will not coincide completely with the teacher’s logits."

---

> ### Author Response · Authors · 2025-11-19
>
> ## 2. Questions
> **(Q1) How does the method address potential catastrophic forgetting with multiple tasks?**
>
> We thank the reviewer for the question. By splitting the server model’s architecture along modality-task pairs, the model ensures that each task operates with differing parameter subsets. Therefore, training on new tasks does not overwrite completely what was learned earlier, thereby mitigating catastrophic forgetting. Moreover, should a client desire to do so, it can purposefully select samples such that all tasks are represented during each local training round, thereby ensuring that all tasks are consistently updated.
>
> **(Q2) KD risk of compromising personalized adaptions and technical motivation of KD.**
>
> We thank the reviewer for the insightful question. As mentioned in an earlier response, the rationale of utilizing KD is due to the fact it’s not as an aggressive operation as replacement, where we can balance both personalization and learning generalizable representations from the server-engaged parameters. Moreover, should KD overpowering personalization be a matter of concern, one can simply decrease the weight given to the KL-divergence term during the post-FL training phase, allowing flexibility in how much distillation is desired from each task. We also re-iterate that [1] theoretically shows that in general the student’s logits will not coincide completely with the teacher’s.
>
> **(Q3) Image task benefit of KD marginal compared to text datasets.**
>
> We thank the reviewer for the insightful question. As shown in prior work [2], language models tend to produce more overconfident and sharply peaked predictive distributions compared to their vision counterparts. In FL settings, text data also can exhibit substantially higher cross-client heterogeneity. For example, clients may differ in vocabulary, topic, writing style, etc. In contrast, image datasets such as CIFAR-100 and Tiny-ImageNet share a relatively consistent semantic space: the set of object categories and visual concepts remains more stable across clients. This means for text data, the global model captures a broader and more diverse distribution over words than any single client, and therefore KD provides a more breadth of knowledge to the client in comparison to image-based tasks. In line with the reviewer’s comment, we added a more thorough analysis of this phenomenon to the experimental section discussing KD usage, which we also attach below:
> - "In regards to the difference in performance of image vs. text based tasks with KD, we believe this discrepancy to occur due to (i) the overconfidence of language models in comparison to vision models (Li et al., 2025) and (ii) the higher cross-client heterogeneity of text data. With text data, clients may differ in aspects such as vocabulary, topic, or writing style, meaning for text data, the server model captures a broader and more diverse distribution over words than any single client. This provides KD more breadth of knowledge to the client in comparison to image-based tasks."
>
> **(Q4) In the experimental section, does each client correspond to a single task or multiple tasks?**
>
> We thank the reviewer for the question. For our experimental results, each client’s specific number of tasks and modalities are varied–uni-modal and uni-task, multi-task and uni-modal, uni-task and multi-modal, or both multi-task and multi-modal are the potential combinations that a client can exhibit.
>
> **References:**
>
> [1] Towards understanding knowledge distillation (Phuong & Lampert, ICML 2019).
>
> [2] Bild: Bi-directional logits difference loss for large language model distillation (Li et al., International Conference on Computational Linguistics 2025)

---

> ### Author Response · Authors · 2025-11-27
>
> We once again thank the reviewer for the careful evaluation of our paper. We would appreciate any additional comments you may have in light of our latest responses.

---

### Official Review · Reviewer_8mFv · 2025-11-01

**Soundness:** 3
**Presentation:** 2
**Contribution:** 3
**Rating:** 4
**Confidence:** 2

**Summary:**

In this paper, the authors propose TAP, a two-stage personalized federated learning framework for handling multi-task and multi-modal foundation models. Specifically, TAP introduces adaptive parameter replacement during federated training based on task-specific performance margins, and post-FL knowledge distillation to incorporate general knowledge without compromising personalization.

**Strengths:**

1. The proposed TAP addresses an underexplored but important setting: personalized federated learning (PFL) with both multi-task and multi-modal heterogeneity.

2. Provides the convergence analysis of a server model under a modality–task pair architecture.

3. The code is attached, making the method reproducible.

**Weaknesses:**

1. There is no analysis of how sensitive TAP is to margin hyperparameters. This is a hard threshold, which may cause optimization instability.

2. Figure 1-2 do not clearly illustrate which parameters are replaced or distilled. In particular, in Figure 2 stage(1) , the model appears visually identical before and after the replacement step, making it difficult to understand what changes occur or how the adaptive replacement actually operates. The figures are schematic rather than explanatory.

3. Communication efficiency is a key factor in personalized federated learning, yet the paper does not analyze or report the communication cost of TAP.

4. The proposed two-stage design (adaptive replacement and knowledge distillation) largely combines existing personalization and KD techniques in a straightforward way. Adaptive replacement is conceptually similar to selective parameter updates or client-specific gating, and post-FL KD is already common.

5.  The paper does not provide a clear discussion of its limitations or potential failure scenarios. While TAP shows consistent improvements across benchmarks, the authors do not analyze under what conditions the method might underperform.

6. The overall organization of the paper could be improved, especially in the Method section, which is dense and sometimes difficult to follow due to lengthy descriptions and intertwined explanations of architecture and training procedures. In addition, there are several presentation issues and minor errors that affect readability. For example, line 161 (“with it being being fine-tuned with LoRA...”) contains a duplicated word, and the section title 3.1 appears to include a typo (“Multi-Mask” should be “Multi-Task”).

**Questions:**

How does the proposed method handle aggregation across heterogeneous client architectures? Specifically, when clients possess different subsets of $\widetilde{W}$ due to varying tasks and modalities, how are these mismatched components aligned or mapped to enable FedAvg-style aggregation without a shared latent space?

---

> ### Author Response · Authors · 2025-11-19
>
> We thank the reviewer for taking the time and effort to give valuable feedback.
>
> ## 1. Weaknesses
> **(W1) Potential sensitivity of margin hyperparameters**:
>
> We thank the reviewer for the insightful question. In regards to the sensitivity of the margin hyperparameters, we would like to point the reviewer to an ablation study on this in Appendix C.2. Here, we demonstrate that differing margin hyperparameters do not result in significant performance differences for a majority of tasks, meaning that while the margins might need tuning for some tasks, it is not overly sensitive.
>
> **(W2) Figures more schematic than explanatory**
>
> We thank the reviewer for this comment. To clarify Figure 1, we have added arrows and increased font size to better emphasize differing parts of the server model architecture. In terms of Figure 2, besides increasing font size, in Stage 1 we have introduced a dotted-box scheme to better delineate the replacement operation. For the distillation stage (Stage 2), we add text to specify that the loss from the logits output are used as an auxiliary loss, which backpropagates on the personalized model to avoid confusion.
>
> **(W3) Communication cost of TAP**
>
> We thank the reviewer for the insightful comment. We would like to point the reviewer to the following remark near the conclusion of Sec. 3.2, which in part states:
> - "We note that the replacement process, which occurs in conjunction with the FL training protocol, does not induce additional communication costs, as the personalized parameters $\mathbf{X}_{[i]}$ *are not transmitted to server $S$.*"
>
> **(W4) Proposed method largely combines existing personalization and KD techniques in a straightforward way.**
>
> We thank the reviewer for bringing up this point. We would like to emphasize that we are the first to apply selective modality-task pair module replacement in a federated multi-modal and multi-task setting. Moreover, due to the heterogeneity of modalities and tasks between clients, and therefore their model architectures, selective modality-task pair module replacement of TAP is particularly well-suited for the problem setup, as it allows for replacement of specific modality-task pairs without negatively impacting those pairs that do not require replacement. In contrast, existing works do not contend with this additional dimension and therefore are not suited for knowing when certain modality-task pairs should be prioritized over others. Moreover, gating-based works often require the training of additional routing parameters, as seen in [1], which TAP does not require, as it utilizes floating point margin hyperparameters.
>
> In addition, though post-FL KD has been studied in prior work, our use of KD is motivated by challenges that are unique to our setting and has not been previously explored in the context of task and modality heterogeneous FL. In our framework, TAP is deliberately designed to minimize interaction with server-engaged parameters in the FL training phase, which allows each client to maintain a high degree of personalization. However, this decoupling means that personalized parameters may not fully benefit from the broad, cross-client representations that occur during aggregation at the server. These representations produced from the server-engaged parameters could encompass generalizable patterns derived from diverse client data, which a purely personalized model would otherwise not have access to. Unlike more aggressive operations, like the replacement that takes place in the FL stage, post-FL KD enables transfer of beneficial knowledge while preserving each client’s personalized nature. We also emphasize that, to our knowledge, we are the first to demonstrate that this strategy remains effective even under heterogeneity of modality and task scenarios.
>
> **(W5) No discussion on limitations.**
>
> We thank the reviewer for the comment. To better address the potential shortcomings of TAP, we have extended the conclusion to include discussions on limitations and future directions upcoming works can follow. We have attached the exact wording from the updated manuscript below for the reviewer's convenience:
> - "A potential limitation of our work is the reliance of the training loss to conduct replacements, which can produce issues should the training loss not be an accurate representation of the true difference in performance between the personalized and regular parameters on a specific task. Future work can explore differing methods of engaging in replacement to mitigate this pitfall."
>
> **(W6) Overly lengthy descriptions and minor readability errors**
>
> We thank the reviewer for the concise comment. For organizational purposes, we have moved many of the granular specifics about the architecture to the appendix (Appendix A.2). Moreover, we have addressed the typos on duplicate wording and misspelling of task for section title 3.1.

---

> ### Author Response · Authors · 2025-11-19
>
> ## 2. Questions
> **(Q) How does the proposed method handle aggregation across heterogeneous client architectures?**
>
> We thank the reviewer for the question. As outlined near the conclusion of Sec. 3.1 (beneath Eq. 2), aggregation is handled on a component-wise basis, whereby it averages common components that exist between clients chosen for an aggregation round. For example, if during aggregation two clients have a common task dealing with ImageNet image classification, the subset of the server model responsible for ImageNet image classification will update based on those two clients. The aforementioned sentence in the manuscript near the conclusion of Sec. 3.1 (beneath Eq. 2) states:
> - "Due to heterogeneous model architectures amongst clients in the current setup, aggregation is performed by utilizing FedAvg (McMahan et al., 2017) on each component individually, combining common components based off the clients chosen for a specific aggregation round. Therefore, it is a possibility that the entirety of $\widetilde{\mathbf{W}}$ is not updated within a single communication round."
>
> **References:**
> [1] Flow: per-instance personalized federated learning (Panchal et al., NeurIPS 2023).

---

> ### Author Response · Authors · 2025-11-27
>
> We once again thank the reviewer for the careful evaluation of our paper. If your schedule allows, we would appreciate any additional comments in light of our latest responses.

---

### Official Review · Reviewer_nfPM · 2025-11-02

**Soundness:** 3
**Presentation:** 3
**Contribution:** 3
**Rating:** 6
**Confidence:** 4

**Summary:**

The paper proposes an architecture for personalized federated learning dealing with multiple modalities, where different clients focus on potentially different tasks and different modalities, so only part of the full model is shared with a given client. Using a special architecture, building on the idea of Mixture of Experts, all client models get integrated in the global model, which is distributed again at the end of learning to post-train the local models via knowledge distillation.

**Strengths:**

The paper addresses the interesting yet challenging setting where different clients share only part of a global model, caused by different modalities and/or different tasks (mostly output modalities/formats).

**Weaknesses:**

1. The setup still seems to make quite a few assumptions about clients 'overlapping' and sharing a common goal, and it would be good to make those explicit. How similar or aligned should the tasks be to make sharing better than splitting ?

2. The architecture of the global model is chosen specifically for this setup. All baselines and competing methods are compared against the proposed method using this same architecture, while they were not specifically designed for it (unlike the proposed method).
The results should be compared against more standard architectures, to be able to appreciate the results.

3. There's little discussion in the paper about limitations of the proposed scheme.

4. The presentation of the paper could be improved. In particular the figures showing the overall architecture are not very clear.

**Questions:**

See above.

---

> ### Author Response · Authors · 2025-11-19
>
> We thank the reviewer for taking the time and effort to give valuable feedback.
>
> ## 1. Weaknesses
> **(W1) Assumptions on overlapping components and sharing of common goal**
>
> We do indeed assume that despite there being no unified architecture amongst clients, some clients may share some modality-task pairs. However, this hard-coded 0/1 definition of task similarity is similar to existing works, such as with [1, 2].
>
> **(W2) Comparison against more standard architecture**
>
> We thank the reviewer for bringing up this concern. In line with the suggestion, we have added an experimental section exploring the use of a standard transformer backbone architecture vs. TAP in Appendix C.4. We have added the results and observations from Appendix C.4 below for the reviewer's convenience. ***We would like to emphasize that these are only preliminary results. A separate update will be posted once all experimental results are available.***
>
> Table 14: Usage of standard transformer backbone with FedAvg vs. TAP on image datasets for FLAVA.
>
> | Model | Method              | Tiny-ImageNet (Acc ↑) | CIFAR-100 (Acc ↑) | FMNIST (MSE ↓)        | Caltech-256 (MSE ↓)   | Avg. Class. (Acc ↑) | Avg. Class. (MSE ↓) |
> |--------|----------------------|------------------------|---------------------|------------------------|-------------------------|------------------------|-------------------------|
> | FLAVA  | FedAvg               | 27.93 ± 12.69          | 35.25 ± 12.57        | 0.5996 ± 0.0163         | 0.6117 ± 0.0555          | 31.59                  | 0.6077                  |
> | FLAVA  | FedAvg + Post-train  | 29.55 ± 12.49          | 37.08 ± 12.52        | 0.5947 ± 0.0150         | 0.5959 ± 0.0533          | 33.32                  | 0.5955                  |
> | FLAVA  | TAP (Ours)           | **47.06** ± 6.80        | **56.26** ± 5.44      | **0.5572** ± 0.0083     | **0.4252** ± 0.0258       | **51.66**              | **0.4692**              |
>
> Table 15: Usage of standard transformer backbone with FedAvg vs. TAP on text datasets for FLAVA.
>
> | Model | Method              | AG News (Acc ↑) | MMLU (BS ↑) | MMLU ( METEOR ↑) | VQA (BS ↑) | VQA (METEOR ↑) | CommonGen (BS ↑) | CommonGen (METEOR ↑) | Avg. Gen. (BS ↑) | Avg. Gen. (METEOR ↑) |
> |--------|----------------------|------------------|------------------|------------------|---------------|-------------------|--------------|---------------------------|----------------------|-----------------------------|
> | FLAVA  | FedAvg               | 90.48 ± 0.60      | 41.67 ± 1.24      | 21.39 ± 2.61      | 48.32 ± 1.91   | 10.88 ± 2.52       | 30.37 ± 1.65  | 12.51 ± 2.17             | 39.81          | 13.63                      |
> | FLAVA  | FedAvg + Post-train  | 90.68 ± 0.54      | 41.90 ± 0.96      | **21.68** ± 2.37      | 47.88 ± 1.91   | 10.48 ± 2.24       | 30.39 ± 1.47  | 13.17 ± 2.97             | 39.69          | 13.80                      |
> | FLAVA  | TAP (Ours)           | **92.66** ± 0.34  | **45.81** ± 4.32  | 19.80 ± 3.83  | **69.38** ± 7.47 | **47.60** ± 14.92 | **34.01** ± 2.96 | **17.17** ± 4.56        | **50.52**             | **29.87**                  |
>
> - "From Tables 14 and 15, we note that TAP outperforms FedAvg on all metrics with the exception of one metric in MMLU METEOR. Moreover, we note for tasks that TAP previously lagged behind marginally on, such as AG News, TAP now exhibits superior performance in comparison to the baselines. Overall, this demonstrates the merit of adopting a model architecture that is modality-task pair aware, to better prevent conflicting tasks from competing with one another."
>
> **(W3) No discussion on limitations**
>
> In line with the reviewer’s suggestion, we have extended the conclusion to discuss potential limitations of our methodology and potential avenues for future work, whose exact wording we attach below for the reviewer's convenience:
> - "A potential limitation of our work is the reliance of the training loss to conduct replacements, which can produce issues should the training loss not be an accurate representation of the true difference in performance between the personalized and regular parameters on a specific task. Future work can explore differing methods of engaging in replacement to mitigate this pitfall."
>
> **(W4) Presentation of Figures and Organization**
>
> We thank the reviewer for raising this point. We have added more text and arrows to Fig. 1 to better direct attention where needed. Similarly, we have increased the size of the text in Fig. 2 and added more descriptive indicators to highlight the steps of the TAP algorithm. To improve clarity and readability of Sec. 3.1, we have additionally moved part of the description of server model $S$ to the appendix (Appendix A.2).
>
> **References:**
>
> [1] FedHCA: Towards Hetero-Client Federated Multi-Task Learning (Lu et al., CVPR 2024).
>
> [2] On disentanglement of asymmetrical knowledge transfer for modality-task agnostic federated learning (Chen & Zhang, AAAI 2024).

---

> ### Author Response · Authors · 2025-11-21
>
> **(W2 Updated) Comparison against more standard architecture**
>
> We thank the reviewer on waiting for our updated response. As indicated in a previous post, we now present more comprehensive results on the usage of TAP against relevant baselines on a more standard backbone architecture.  The updated tables and observations from the manuscript have been attached below for the reviewer's convenience.
>
> Table 14: Standard transformer backbone with baselines vs. TAP on image datasets for ViLT.
>
> | **Model** | **Method** | **FMNIST (MSE ↓)** | **Caltech-256 (MSE ↓)** | **Avg. Gen. (MSE ↓)** |
> |----------|------------|--------------------|--------------------------|------------------------|
> | **ViLT** | FedAvg | 0.6274 ± 0.0243 | 0.6192 ± 0.0873 | 0.6089 |
> || FedAvg + Post-train | 0.6097 ± 0.0201 | 0.5701 ± 0.0816 | 0.5939 |
> || Per-FedAvg | 0.7476 ± 0.0726 | 1.0180 ± 0.2944 | 0.8558 |
> || Per-FedAvg + Post-train | 0.6367 ± 0.0310 | 0.6436 ± 0.0703 | 0.6395 |
> || FedDAT | 0.5648 ± 0.0188 | 0.4332 ± 0.0571 | 0.5121 |
> || FedDAT + Post-train | $\underline{0.5612} \pm 0.0179$ | $\underline{0.4235} \pm 0.0566$ | $\underline{0.5061}$ |
> ||**TAP (Ours)**| **0.5467** ± 0.0409 | **0.3949** ± 0.1148 | **0.4860** |
>
> Table 15: Standard transformer backbone with baselines vs. TAP on image datasets for FLAVA.
>
> | **Model** | **Method** | Tiny-ImageNet (Acc ↑) | CIFAR-100 (Acc ↑) | FMNIST (MSE ↓) | Caltech-256 (MSE ↓) | Avg. Class. (Acc ↑) | Avg. Gen. (MSE ↓) |
> |----------|------------|-----------------------|-------------------|----------------|----------------------|----------------------|--------------------|
> | **FLAVA** | FedAvg | 27.93 ± 12.69 | 35.25 ± 12.57 | 0.5996 ± 0.0163 | 0.6117 ± 0.0555 | 31.59 | 0.6077 |
> || FedAvg + Post-train | $\underline{29.55} \pm 12.49$ | 37.08 ± 12.52 | 0.5947 ± 0.0150 | 0.5959 ± 0.0533 | $\underline{33.32}$ | 0.5955 |
> || Per-FedAvg | 26.93 ± 13.76 | 35.10 ± 12.45 | 0.7397 ± 0.0475 | 0.8654 ± 0.0932 | 31.02 | 0.8204 |
> || Per-FedAvg + Post-train | 28.70 ± 13.80 | $\underline{37.59} \pm 11.93$ | 0.6081 ± 0.0169 | 0.6151 ± 0.0420 | 33.14 | 0.6128 |
> || FedDAT | 17.65 ± 16.77 | 30.45 ± 18.06 | 0.5815 ± 0.0125 | 0.5655 ± 0.0530 | 24.05 | 0.5709 |
> || FedDAT + Post-train | 19.11 ± 17.29 | 32.60 ± 18.88 | $\underline{0.5782} \pm 0.0122$ | $\underline{0.5550} \pm 0.0513$ | 25.85 | $\underline{0.5628}$ |
> || **TAP (Ours)** | **47.06** ± 6.80 | **56.26** ± 5.44 | **0.5572** ± 0.0083 | **0.4252** ± 0.0258 | **51.66** | **0.4692** |

---

> ### Author Response · Authors · 2025-11-21
>
> Table 16: Standard transformer backbone with baselines vs. TAP on text datasets.
>
> | **Model** | **Method** | AG News (Acc ↑) | MMLU (BS ↑) | MMLU (METEOR ↑) | VQA (BS ↑) | VQA (METEOR ↑) | CommonGen (BS ↑) | CommonGen (METEOR ↑) | Avg. Gen. (BS ↑) | Avg. Gen. (METEOR ↑) |
> |----------|------------|------------------|------------|----------------|-----------|---------------|---------------------|--------------------|--------------------|-----------------------|
> | **FLAVA** | FedAvg | 90.48 ± 0.60 | 41.67 ± 1.24 | 21.39 ± 2.61 | 48.32 ± 1.91 | 10.88 ± 2.52 | 30.37 ± 1.65 | 12.51 ± 2.17 | 39.81 | 13.63 |
> || FedAvg + Post-train | 90.68 ± 0.54 | $\underline{41.90} \pm 0.96$ | $\underline{21.68} \pm 2.37$ | 47.88 ± 1.91 | 10.48 ± 2.24 | 30.39 ± 1.47 | 13.17 ± 2.97 | 39.69 | 13.80 |
> || Per-FedAvg | 89.64 ± 1.45 | 41.46 ± 1.99 | 20.31 ± 3.93 | 47.12 ± 2.49 | 10.20 ± 2.10 | 30.60 ± 1.40 | 13.19 ± 2.72 | 39.38 | 13.42 |
> || Per-FedAvg + Post-train | $\underline{90.72} \pm 0.58$ | 41.87 ± 1.14 | 20.85 ± 2.19 | 47.31 ± 2.24 | 10.01 ± 1.34 | 30.42 ± 1.30 | 13.57 ± 2.70 | 39.47 | 13.60 |
> || FedDAT | 90.21 ± 0.68 | 41.63 ± 1.05 | 21.06 ± 1.96 | $\underline{67.65} \pm 11.40$ | **57.74** ± 12.10 | $\underline{40.86} \pm 2.56$ | $\underline{25.42} \pm 1.77$ | **51.73** | **37.48** |
> || FedDAT + Post-train | 90.62 ± 0.65 | 41.88 ± 0.75 | **21.88** ± 2.52 | 66.97 ± 10.98 | $\underline{57.23} \pm 11.79$ | **41.13** ± 1.94 | **25.50** ± 1.74 | $\underline{51.62}$ | $\underline{37.47}$ |
> || **TAP (Ours)** | **92.66** ± 0.34 | **45.81** ± 4.32 | 19.80 ± 3.83 | **69.38** ± 7.47 | 47.60 ± 14.92 | 34.01 ± 2.96 | 17.17 ± 4.56 | 50.52 | 29.87 |
> | **ViLT** | FedAvg | 79.88 ± 6.30 | 54.91 ± 2.00 | $\underline{41.08} \pm 3.53$ | 55.58 ± 4.27 | 39.73 ± 11.87 | 40.51 ± 1.51 | **25.96** ± 1.67 | 49.42 | 34.49 |
> || FedAvg + Post-train | 82.39 ± 3.07 | 55.16 ± 0.97 | 40.47 ± 1.30 | 55.08 ± 3.51 | 39.42 ± 10.86 | 40.62 ± 0.99 | $\underline{25.66} \pm 1.88$ | 49.31 | 34.12 |
> || Per-FedAvg | 68.38 ± 11.39 | 55.51 ± 3.02 | 39.01 ± 1.52 | 54.99 ± 2.87 | 31.29 ± 8.29 | 39.94 ± 3.43 | 23.53 ± 4.66 | 49.07 | 29.73 |
> || Per-FedAvg + Post-train | 81.68 ± 3.88 | 55.03 ± 0.89 | **41.73** ± 1.85 | 55.20 ± 3.24 | 30.02 ± 6.45 | 40.40 ± 1.56 | 24.25 ± 3.03 | 49.24 | 30.06 |
> || FedDAT | $\underline{85.95} \pm 0.95$ | 54.74 ± 1.99 | 40.35 ± 2.94 | $\underline{66.61} \pm 10.48$ | $\underline{60.54} \pm 6.41$ | **43.32** ± 5.71 | 25.58 ± 1.89 | $\underline{54.34}$ | $\underline{42.52}$ |
> || FedDAT + Post-train | **86.72** ± 0.46 | $\underline{55.30} \pm 1.09$ | 40.77 ± 1.15 | 65.92 ± 10.32 | 60.44 ± 6.33 | $\underline{41.77} \pm 0.96$ | 25.43 ± 1.89 | 54.13 | 42.50 |
> || **TAP (Ours)** | 77.44 ± 2.63 | **56.43** ± 3.12 | 40.79 ± 1.61 | **72.99** ± 7.47 | **62.52** ± 9.09 | 41.46 ± 1.45 | 25.35 ± 1.85 | **57.07** | **43.31** |
>
>
> - "Here, we compare the usage of TAP with the model architecture adopted from Chen & Zhang (2024) against a more standard architecture with a non modality-task pair-aware server. In this scenario, each client employs a standard transformer layer structure, with no MoE routing taking place (i.e., each client shares the same backbone), which is sent to the server for vanilla FedAvg (not component-wise) aggregation (McMahan et al., 2017).
>
> 	From Tables 14, 15, and 16, we note that TAP outperforms the baselines on a majority of the tasks evaluated. Even when TAP lags behind one of the baselines, such as with TAP vs. FedDAT on CommonGen, it is the most consistent method across all tasks. For example, TAP retains an average BS score and image accuracy of 50.52 and 51.66 respectively on FLAVA. By contrast, FedDAT + Post-train obtains 51.62 and 25.85, indicating that while FedDAT + Post-train achieves marginally better average BS scores, it significantly underperforms TAP on image classification. Overall, this demonstrates the merit of adopting a model architecture that is modality-task pair aware, to better prevent conflicting tasks from competing with one another and ensuring better consistency in performance across all tasks."

---

> ### Author Response · Authors · 2025-11-27
>
> We once again thank the reviewer for the careful evaluation of our paper. We would appreciate any additional comments you may have in light of our latest responses.

---

### Official Review · Reviewer_4iKf · 2025-11-10

**Soundness:** 3
**Presentation:** 3
**Contribution:** 3
**Rating:** 6
**Confidence:** 3

**Summary:**

The paper studies the problem of personalized fine-tuning of foundation models (with multi-task and multi-modal properties) in a Federated Learning (FL) setup. In this setting, the nature of heterogeneity in FL is expanded to include not just data heterogeneity, but also task and modality heterogeneity across clients. The paper develops a new learning algorithm called TAP for this scenario that is empirically shown to do well on accuracy metrics under several model architectures and datasets. TAP is further supported by a theoretical convergence analysis of the server model.

**Strengths:**

(S1) The problem at hand (pFL fine-tuning of foundation models with task, modality, data heterogeneity) is realistic and very important in practice. While the components used to build up TAP have appeared previously in published literature, putting them together to design a solution has originality.

(S2) The paper is well-written, and the presentation is clear. The problem is well motivated, and contributions of the paper are largely well contextualized w.r.t. cited prior work. TAP's design is well explained with adequate intuitions provided throughout.

(S3) Theoretical convergence analysis is presented for TAP under assumptions that are standard in FL literature. The theorem statement and proof look correct to me. The analysis also suggests a non-trivial insight that convergence rate could slow down as a larger number of modality-task pairs are catered to by the server model. The paper also presents good (could be improved upon) experimental support for the performance of TAP.

**Weaknesses:**

(W1) [Flow] has previously introduced the idea of dynamic routing that resembles the replacement mechanism discussed in section 3.2. Also, [FedHCA] presents several experiments which do cover multi-modal and multi-task FL setup. These should be cited and the differences briefly stated in the paper.

[Flow] Panchal, K., Choudhary, S., Parikh, N., Zhang, L., & Guan, H. (2023). Flow: Per-instance personalized federated learning through dynamic routing. 37th Annual Conference on Neural Information Processing Systems (NeurIPS 2023)

[FedHCA] Lu, Y., Huang, S., Yang, Y., Sirejiding, S., Ding, Y., & Lu, H. (2024). FedHCA$^2$: Towards Hetero-Client Federated Multi-Task Learning. In Proceedings of the IEEE/CVF Conference on Computer Vision and Pattern Recognition (CVPR), pp. 4106-4115.

(W2) Lines 345-347: While this is theoretically indicated by equation (7), does it actually show up in practice? I couldn't find which experiment in the paper would support/refute this possibility.

(W3) In the main body of the paper, there is no concise supporting statistics/discussion on the necessity of $R_i\[o\]$ based selection for the problem at hand. As best I could tell, supporting evidence for some tasks is only available in Appendix C.0.3 through Fig. 3 & 4.

Things to improve the paper that did not impact the score:
- Lines 151, 155 - $\mathcal M$ and $\mathcal O$ are undefined.

**Questions:**

(Q1) Based on details of DisentAFL and TAP, I don't understand why TAP should outperform DisentAFL/DisentAFL + Post-train. I couldn't find any targetted discussion around this in the paper. Could the authors explain?

(Q2) Lines 200-202: Does the server need to know the tasks and modalities at each client? Are there privacy/security implications?

(Q3) Section 5.2 Tables 1-3: Should there be different numbers to judge generalization and personalization separately?

---

> ### Author Response · Authors · 2025-11-19
>
> We thank the reviewer for taking the time and effort to give valuable feedback.
>
> ## 1. Weaknesses
> **(W1) Differences from Flow and FedHCA**
>
> We thank the reviewer for raising these points. To address the differences of our proposed method and the aforementioned works of Flow and FedHCA, we emphasize the following points: for FedHCA, we note that while it does exhibit heterogeneity in tasks and model architectures amongst clients, it only works with the image modality, and therefore is not multi-modal. In addition, with Flow, while its routing mechanism allows for the usage of either the personal or server-based components, our method does not require the training of additional routing parameters to switch between personal or server-engaged parameters as we utilize floating point margin hyperparameters. We further note that both works do not work with fine-tuning already existing pre-trained foundation models. In terms of our setup, which exhibits heterogeneity in modalities and tasks across clients, we are one of the only few to our knowledge that explores this setting. DisentAFL [1] is our closest peer, which we demonstrate that TAP outperforms in our numerical results. As per the reviewer’s point, we have added discussion and references to Flow and FedHCA in the related work section under “Personalized Federated Learning” to emphasize these differences, which we have also attached in this response for convenience:
> - "While Lu et al. (2024) explores heterogeneous model architectures in a multi-task setting, it is specifically designed for image inputs and therefore is uni-modal, unlike TAP, which deals with both heterogeneous tasks and modalities. To deal with this limitation, Chen & Zhang (2024) propose to have separate encoders and decoders for differing modalities and tasks, with a shared transformer backbone utilizing a Mixture of Experts (MoE) (Yuksel et al., 2012) structure to route inputs based off modality-task pairs. However, while a disentanglement axillary loss is introduced to potentially create more separable latent spaces between pairs, this approach can still result in entanglement of local models that share common subsets of the server model during aggregation. By contrast, TAP seeks to decouple itself as much as possible from the FL training process by only interacting with the server-engaged parameters when they signal that they can benefit the personalization process, avoiding the issue of losing the granularity of specific tasks during aggregation. Similar to Chen & Zhang (2024), Panchal et al. (2022) also explores using a routing mechanism for personalization, but the approach is uni-modal and uni-task without consideration for foundation models. Moreover, it necessitates the training of additional routing parameters, which TAP does not require for replacement operations."
>
> **(W2) Demonstrating Support for Eq. 7.**
>
> We thank the reviewer for the insightful question. We see that for Tables 1,2, and 3, FedAvg, DisentAFL, and other baselines consistently underperform compared to TAP on a vast majority of tasks. This is because by limiting interaction with the server, TAP is able to focus better on the tasks prevalent to its own local data. The server, conversely, has to consider more modality-task pairs when doing aggregation of common components, leading to some of the information for a specific modality-task pair to be lost during averaging. The more modality-task pairs there are, the more prevalent this becomes. We would also like to state that Lines 345-347 in the original submission now correspond to Lines 367-369.
>
> **(W3) Necessity of $\mathbf{R}\_i[o]$ based selection for the problem.**
>
> We thank the reviewer for the question. At the heart of TAP during the FL training phase is the following: knowing when to conduct replacements. $\mathbf{R}\_i[o]$ is necessary not only because it tells us that replacement should take place for a task due to the superior performance of the FL-engaged parameters at that timestep, but additionally due to the fact it prevents replacement operations where undesirable. This is because through $\mathbf{R}\_i[o]$ in Eq. 3, we ensure that $\mathbf{X}$’s training loss is no worse than $\mathbf{\widetilde{W}}$’s for task $o$. To clarify this point further, we have added a brief sentence after Eq. 3 explaining this necessity, which we attach below for the reviewer's convenience:
> - "In addition, through equation 3, we ensure $\mathbf{X}\_{[i]}$’s training is no worse than $\mathbf{\widetilde{W}}\_{[i]}$’s on any task $o$, preventing replacements on tasks that do not require it."
>
> **(Definition of $\mathcal{M}_i$ and $\mathcal{O}_i$) Lines 151, 155 - $\mathcal{M}$ and $\mathcal{O}$ are undefined.**
>
> We thank the reviewer for pointing out this discrepancy. We have added definitions to $\mathcal{M}$, $\mathcal{M}_i$, $\mathcal{O}$, and $\mathcal{O}_i$ at lines 199-201 in the newly updated manuscript.

---

> ### Author Response · Authors · 2025-11-19
>
> ## 2. Questions
> **(Q1) Explanation of TAP's improvement in comparison to DisentAFL**
>
> We thank the reviewer for the insightful question. DisentAFL relies primarily on the architecture of the server for its personalization. The intuition of DisentAFL is the fact that separating parts of the model based on modality-task pairs makes it easier for the server to learn a larger, all-encompassing model while also better catering to each client’s needs. Besides this, an orthogonality auxiliary loss is introduced, whereby the client attempts to make differing modality-task pairs orthogonal to each other in the latent space. TAP exhibits superior performance because it attempts to decouple itself from the server-client interaction as much as possible. With DisentAFL, due to no explicit decoupling process in place, it can potentially exhibit lower performance when many task(s) from multiple clients utilize common components. As a quick example, if there are many clients working with differing text generation tasks, they will all overlap to varying degrees in terms of the parameters they activate. This means during aggregation, some of the granularity for a specific task will be lost in the averaging, affecting personalization. To be distinct about this point, we have placed greater emphasis on the shortcomings of DisentAFL compared to our method in the related work under the “Personalized Federated Learning” section, whose exact wording the reviewer can find in **(W1)**.
>
> **(Q2) Server knowledge of modalities and tasks on each client**
>
> We thank the reviewer for bringing up this insightful question. For the aggregation process to work, the server will have to know what tasks and modalities each client has. However, this setup is identical to the privacy guarantees as existing works, such as DisentAFL [1].
>
> **(Q3) Judging generalization vs. personalization**
>
> We thank the reviewer for the question. The reported results use personalized accuracy, meaning we compute each client’s evaluation metrics on its corresponding test dataset, consistent with established practice in prior work [1-3]. Since TAP aims to provide per-client personalization aligned with each client’s specific tasks, personalized evaluation is a natural choice.
>
> **References:**
>
> [1] On disentanglement of asymmetrical knowledge transfer for modality-task agnostic federated learning (Chen & Zhang, AAAI 2024).
>
> [2] Personalized federated learning with theoretical guarantees: A model-agnostic meta-learning approach (Fallah et al., NeurIPS 2020).
>
> [3] FedDAT: An approach for foundation model finetuning in multi-modal heterogeneous federated learning (Chen et al., AAAI 2024).

---

> ### Author Response · Authors · 2025-11-27
>
> We once again thank the reviewer for the careful evaluation of our paper. We would appreciate any additional comments you may have in light of our latest responses.

---

### Comment · Area_Chair_qLRC · 2025-11-24

Dear Reviewers,

**We kindly encourage you to review and respond to the authors’ rebuttals**. Your timely feedback is important for ensuring a fair and thorough review process. Thank you for your contributions to ICLR 2026.

AC

---

### Author Response · Authors · 2025-12-03
**Author Final Comments**

We would like to thank the reviewers for their valuable feedback, helping to improve the contribution of our work. We would also like to thank the AC for taking the time to consider our work in light of the shortened discussion period.

We believe through our responses, the reviewer's questions have been addressed and considered, which has helped to highlight the novelty of TAP, a method by which personalized federated learning can take place when dealing with the additional dimensions of (1) heterogeneity in modalities and tasks, (2) heterogeneity in model architectures, and (3) need to personalize on an individual modality-task basis.

In our response to the reviewer comments, we have incorporated the following to the manuscript and made the following clarifications:
- **Novelty of TAP**: We have clarified misunderstandings on the novelty of our method, whereby we emphasized that with the exception of DisentAFL (whose method we exceed), we are, to the best of our knowledge,  ***the only work that deals with heterogeneous modality and task based personalized FL.*** ***We are also the first to apply selective modality-task pair module replacement in a federated multi-modal and multi-task setting, allowing for personalization on an individual modality-task pair level, which existing works do not consider.*** In regards to the use of KD, we have clarified that its usage is motivated by challenges unique to our setting, driven by the decoupling of personalized and FL-engaged parameters. **(Reviewers 8mFv, KtKr, GYCq)**
- **Additional Baselines**: We have added four additional baselines, two derived from a more recent approach and two from a classical PFL approach, to demonstrate that TAP still maintains superior and consistent performance in comparison. We also added an analysis on the statistical significance of the improvement in the Appendix. **(Reviewer GYCq)**
-  **Additional experimental analysis on TAP's setup**: We have acknowledged that while the approach adopted by TAP introduces additional computational costs, it is consistent with those introduced by existing works, with the training of personalized and regular parameters **(Reviewers AVRh, GYCq)**. We additionally demonstrate that the wall clock time of the KD post-FL stage is short, indicating that TAP does not significantly increase training time compared to existing works **(Reviewers AVRh, GYCq)**. Lastly, we have also shown adopting the modality-task pair aware architecture from DisentAFL in conjunction with TAP outperforms the baselines on a standard transformer backbone **(Reviewer nfPM)**.
- **Organization of Presentation**: We have moved more granular detailed specifics into the appendix **(Reviewer 8mFv)**, added relevant specifics where needed to the proposed methodology section **(Reviewers GYCq, 4iKf, Luv2)**, and made the figures more explanatory, detailed, and readable in response to the reviewer's confusions **(Reviewers nfPM, 8mFv, Luv2)**.
- **Additional Common Comments**: Outside of the previously mentioned comments, we have also addressed additional points that have arisen from multiple reviewers.
	- We have clarified that through Appendix C.2, we demonstrate that TAP is not highly sensitive to margin hyperparameter setting **(Reviewers 8mFv, Luv2)**.
	- We have justified the use of our evaluation metrics as appropriate in the context of personalized FL, emphasizing that our approach for evaluation is the standard in the literature **(Reviewers 4iKf, GYCq)**.
	- We have better differentiated our work from existing methods by emphasizing their limitations in a heterogeneous modality-task setup in comparison to TAP in the related work section of the manuscript **(Reviewers 4iKf, KtKr, GYCq)**.
	- We include discussion of potential limitations of TAP and how future work can work to deal with these pitfalls **(Reviewers nfPM, 8mFv)**.
- **Misunderstandings**: We have additionally addressed misunderstandings that have arisen from our work by emphasizing the focus of TAP is not on modifications to MoEs **(Reviewer KtKr)** or PEFT methods **(Reviewer GYCq)**. Additionally, the convergence analysis demonstrates that TAP is necessary as it is the server model, not the personalized model that suffers due to a large amount of modality-task pairs **(Reviewer AVRh)**. We also demonstrate that TAP is scalable through experiments with 30 clients, not 10 **(Reviewer AVRh)**.

We believe that the additional clarifications and experimental results gives strong support to the novelty of TAP and demonstrates its superiority in comparison to a wide breadth of baselines in an emerging area of research in the FL community.

---

### Meta-Review · Area_Chair_FErL · 2026-01-07

**Summary:**

While the paper addresses a complex and timely problem, several significant concerns shaped the discourse:


* Novelty and Technical Contribution: Multiple reviewers (KtKr, GYCq, 8mFv) argued that the method is an incremental combination of existing techniques like Mixture of Experts (MoE), LoRA, and Knowledge Distillation (KD).


* Baselines and Evaluation: Initial reviews noted a lack of standard personalized federated learning baselines (e.g., Per-FedAvg) and questioned the statistical significance of the results given overlapping standard deviations.


* Clarity and Presentation: Reviewers found the figures confusing and noted that several key terms and hyperparameters were poorly defined or unexplained.


* Motivation for KD: Reviewers questioned the necessity and theoretical grounding of the second stage (knowledge distillation), especially where improvements appeared marginal.


* Scalability: Concerns were raised regarding the small number of clients used in experiments.

**Reviewer Concerns:**

Addressed by Rebuttal

* Scalability and Setup: The authors clarified that experiments used 30 clients, not 10. They also provided time and memory overhead data to show that TAP is computationally feasible.


* Baselines and Statistical Significance: Authors added four baselines (including Per-FedAvg and FedDAT) and performed a Welch t-test, proving that TAP’s improvements are statistically significant in the majority of metrics.


* Agnostic Backbone Support: In response to nfPM, the authors provided new results showing TAP outperforms baselines even on a standard transformer backbone without MoE components.

* Clarity: Figures were updated with better indicators, and granular architectural details were moved to the Appendix to improve flow.

Outstanding Concerns

* Differences from prior work: Reviewers KtKr and GYCq remained skeptical, maintaining that the "surgical" replacement is just a specific application of known concepts.


* Hyperparameter Sensitivity: While authors provided an ablation study for the margin hyperparameter, the need for manual tuning per task remains a potential practical hurdle.


* Qualitative vs. Quantitative Reasoning for KD: Reviewer GYCq felt the explanation for why KD helps text more than images was more "speculative" than proven with direct information-transfer analysis.

**Reviewer Scores:**

* 4iKf:	Already positive; authors addressed all technical clarifications and defined missing terms.


* nfPM: Strongest positive shift expected due to the authors providing comprehensive results on a "standard architecture," which seems to be one of this reviewer's main requests.

* 8mFv: Authors addressed the communication cost, figure clarity, and scalability concerns. However, the "incremental" concern likely persists.


* Luv2: Authors provided the requested motivation for KD and clarified multi-task catastrophic forgetting. Unlikely to drop, but may not rise due to hyperparameter concerns.


* KtKr: In a follow-up, the reviewer stated they still view the work as "incremental" and kept their original rating.

* GYCq: Authors addressed baseline and significance concerns directly. The reviewer was very critical.

* AVRh: The reviewer's primary objection regarding scalability would likely remain (30 is still a small number).

---

### Decision · Program_Chairs · 2026-01-26

Reject